# Petrological and geochemical characterisation of the sarsen stones at Stonehenge

**David J. Nash**[1,2]*, **T. Jake R. Ciborowski**[1], **Timothy Darvill**[3], **Mike Parker Pearson**[4], **J. Stewart Ullyott**[1], **Magret Damaschke**[5], **Jane A. Evans**[6], **Steven Goderis**[7], **Susan Greaney**[8], **Jennifer M. Huggett**[9,10], **Robert A. Ixer**[4], **Duncan Pirrie**[11], **Matthew R. Power**[12], **Tobias Salge**[10], **Neil Wilkinson**[13]

**1** School of Environment and Technology, University of Brighton, Brighton, United Kingdom, **2** School of Geography, Archaeology and Environmental Studies, University of the Witwatersrand, Johannesburg, South Africa, **3** Department of Archaeology and Anthropology, Bournemouth University, Poole, Dorset, United Kingdom, **4** Institute of Archaeology, University College London, London, United Kingdom, **5** Environmental Science Centre, British Geological Survey, Keyworth, Nottinghamshire, United Kingdom, **6** National Environmental Isotope Facility, British Geological Survey, Keyworth, Nottinghamshire, United Kingdom, **7** Analytical-, Environmental-, and Geo-Chemistry, Vrije Universiteit Brussel, Brussels, Belgium, **8** English Heritage, Bristol, United Kingdom, **9** Petroclays Ltd, Heathfield, United Kingdom, **10** Imaging and Analysis Centre, Core Research Laboratories, The Natural History Museum, London, United Kingdom, **11** School of Applied Sciences, University of South Wales, Pontypridd, United Kingdom, **12** Vidence Inc., Burnaby, British Columbia, Canada, **13** Gatan UK, Abingdon, Oxfordshire, United Kingdom

* d.j.nash@brighton.ac.uk

**Data Availability Statement:** All geochemical data are contained within the manuscript and its S1–S4 Data, S1 Movie files. Additional thin-section, SEM-EDS, BSE, micro-XRF and CT scan images are

## Abstract

Little is known of the properties of the sarsen stones (or silcretes) that comprise the main architecture of Stonehenge. The only studies of rock struck from the monument date from the 19th century, while 20th century investigations have focussed on excavated debris without demonstrating a link to specific megaliths. Here, we present the first comprehensive analysis of sarsen samples taken directly from a Stonehenge megalith (Stone 58, in the centrally placed trilithon horseshoe). We apply state-of-the-art petrographic, mineralogical and geochemical techniques to two cores drilled from the stone during conservation work in 1958. Petrographic analyses demonstrate that Stone 58 is a highly indurated, grain-supported, structureless and texturally mature groundwater silcrete, comprising fine-to-medium grained quartz sand cemented by optically-continuous syntaxial quartz overgrowths. In addition to detrital quartz, trace quantities of silica-rich rock fragments, Fe-oxides/hydroxides and other minerals are present. Cathodoluminescence analyses show that the quartz cement developed as an initial <10 μm thick zone of non-luminescing quartz followed by ~16 separate quartz cement growth zones. Late-stage Fe-oxides/hydroxides and Ti-oxides line and/or infill some pores. Automated mineralogical analyses indicate that the sarsen preserves 7.2 to 9.2 area % porosity as a moderately-connected intergranular network. Geochemical data show that the sarsen is chemically pure, comprising 99.7 wt. % $SiO_2$. The major and trace element chemistry is highly consistent within the stone, with the only magnitude variations being observed in Fe content. Non-quartz accessory minerals within the silcrete host sediments impart a trace element signature distinct from standard sedimentary and other crustal materials. $^{143}Nd/^{144}Nd$ isotope analyses suggest that these host sediments were likely derived from eroded Mesozoic rocks, and that these Mesozoic rocks

available via the Archaeology Data Service at https://doi.org/10.5284/1084808.

**Funding:** DJN, TJRC, JSU, MPP, TD were awarded British Academy / Leverhulme Trust Small Research Grant SG170610 https://www.thebritishacademy.ac.uk/. DP received additional funding support from the University of South Wales CESRIS grant (no number) https://www.southwales.ac.uk/. These funders had no role in study design, data collection and analysis, decision to publish, or preparation of the manuscript. JMH, MRP and NW are employed by commercial companies (Petroclays Ltd, Vidence Inc. and Gatan UK, respectively). These companies provided support in the form of salaries but did not have any additional role in the study design, data collection and analysis, decision to publish, or preparation of the manuscript. The specific roles of these authors are articulated in the 'author contributions' section.

**Competing interests:** Three of the authors (JMH, MRP and NW) are employed by commercial companies (Petroclays Ltd, Vidence Inc. and Gatan UK, respectively). This does not alter our adherence to PLOS ONE policies on sharing data and materials.

incorporated much older Mesoproterozoic material. The chemistry of Stone 58 has been identified recently as representative of 50 of the 52 remaining sarsens at Stonehenge. These results are therefore representative of the main stone type used to build what is arguably the most important Late Neolithic monument in Europe.

## 1. Introduction

In 2015, Mr Robin Phillips from Bath (UK) contacted Historic England reporting that his father, Mr Robert Phillips of Aventura (Florida, USA), owned a core extracted from one of the large 'sarsen stones' at Stonehenge [sarsen is a vernacular term for the geochemical sediment *silcrete*; see 1, 2]. The family had had the core in their possession for over 40 years and wanted to repatriate it. A wooden case was sent to Florida for its return, and the core was handed over to English Heritage (the current custodians of Stonehenge) in May 2018. Following publicity surrounding the return of the core, part of a second core was discovered at Salisbury Museum in July 2019. Archive research confirmed that both cores were extracted from Stone 58, one of the large upright sarsen megaliths that form part of the centrally placed trilithon horseshoe at the monument (Fig 1). These megaliths were erected during Stage 2 of the development of Stonehenge at *2585–2400 cal BC* [3, 4]. The cores from Stone 58 are scientifically and culturally important in that they are the only known examples of sarsen stone that can be definitively linked to a specific megalith at the monument. Recent work has also shown that Stone 58 is representative geochemically of 50 of the 52 sarsens remaining at Stonehenge [5]. The cores are, therefore, of international significance in providing a unique window into the physical and chemical properties of the main stone type used in the construction of Stonehenge.

Little is known of the geology of the sarsen stones at Stonehenge. Historically, sarsens in the UK have been divided into two classes, namely 'hard' and 'saccharoid' [on the basis of its appearance resembling "that of a broken loaf-sugar"; 7] types—both are present at Stonehenge. Fragments of both hard and saccharoid sarsen have been excavated during archaeological investigations. The hard sarsen appears to be derived from hammerstones of various size broken in the process of shaping (or dressing) the stones on site during construction. Saccharoid sarsen is used exclusively to form the uprights and lintel stones at the monument.

The first descriptions of the lithology of the Stonehenge sarsens date from the 19th century. The earliest, likely made by James Sowerby from specimens of sarsen from the monument sent to him by Sir Richard Colt Hoare [8], appears in the 1812 monograph *The Ancient History of South Wiltshire* [9]. Here, the material is described as "a fine-grained species of siliceous sandstone" (Vol. 1, p.149). An 1877 essay by Nevil Maskelyne [10] is the first to include a thin-section sketch of a sarsen, cut from stone struck from the monument; however, no further details are given of the petrography of the sample. Similarly, despite providing rich details about other rock types, the descriptions of 172 chips of stone excavated at Stonehenge published in 1884 by William Cunnington [11] contain no details of the nine sarsen fragments included in the collection. Seemingly, the sarsens were considered less worthy of detailed description than the exotic 'bluestones' [a colloquial term used to describe a range of igneous and sedimentary rock types whose origins lie mainly in southwest Wales], the other component of the Stonehenge architecture (Fig 1).

The first detailed research on sarsens at Stonehenge was by John Wesley Judd, who carried out analyses of lithological variations in sarsen hammerstones excavated by William Gowland in 1901 [8]. He described them (pp.109-110) as comprising silica-cemented coarse-grained to

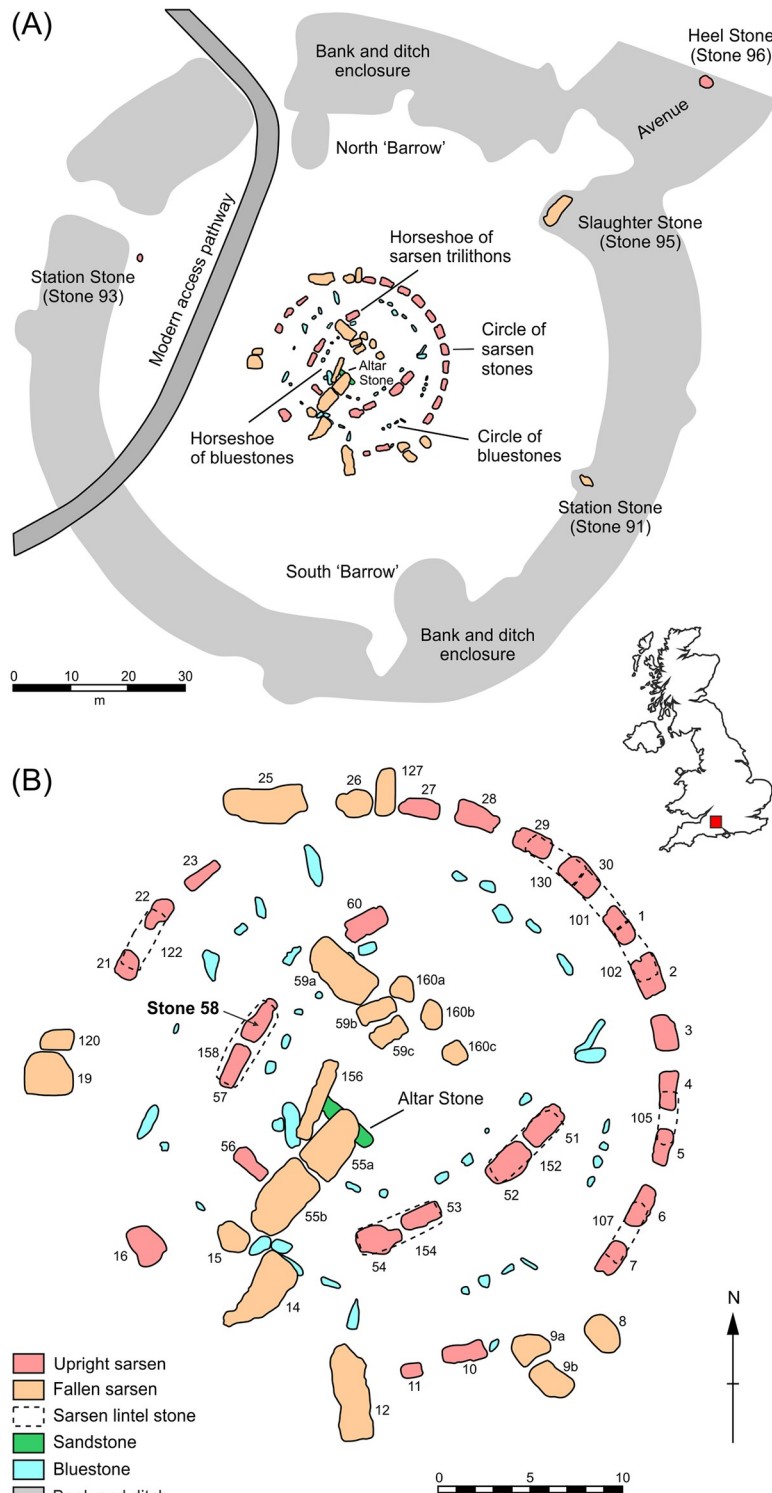

**Fig 1. Plans of Stonehenge showing (A) the area of the monument enclosed by earthworks and (B) detail of the stone circle.** Sarsen stones are numbered following the system devised by W.M. Flinders Petrie in the late 19th century [6].

very fine-grained well-rounded to angular quartz plus feldspars, mica and glauconite, with chips of flint occasionally present (but see critique in section 5.2). The only other 20th century study was by Hilary Howard, who used thin-section and heavy mineral analyses to characterise sarsen fragments recovered during Mike Pitts' 1979 and 1980 excavations [12–14]. Howard [in 12] identified that the majority of the excavated fragments (170 kg in total) were of saccharoid sarsen, with smaller quantities (~8 kg) of hard sarsen. The saccharoid sarsen could be distinguished in thin-section from the hard variety by the size, shape and packing density of cemented quartz grains and the nature of the silica cement. Quartz grains were generally larger and more well-rounded in the saccharoid sarsen, with average grain sizes ranging from 0.2 to 0.5 mm in the 20 fragments analysed. Howard also suggested that there might be variability within the sarsens at Stonehenge; three samples of saccharoid sarsen from a working floor adjacent to the Heel Stone (Stone 96 in Fig 1) excavated by Pitts were found to be consistent in terms of petrology but different from a sample of saccharoid sarsen from the fill in the ditch surrounding the megalith. Heavy mineral analyses indicated that the excavated saccharoid sarsens differed from samples collected from sarsen boulders at nearby Piggledene, particularly in the proportions of zircon, rutile, kyanite, staurolite, andalusite, tourmaline and garnet present [see Table 3 in 12, and further discussion in section 5.2].

The aim of this paper is to document and provide a detailed characterisation of the cores extracted from Stone 58, the first study of its kind for a sarsen megalith at Stonehenge. We do this using a range of techniques (Table 1), selected to include both standard sedimentological approaches and state-of-the-art mineralogical and geochemical methods. Both cores—referred to hereafter as the 'Phillips' Core' and 'Salisbury Museum Core'—are first visually logged. A suite of analyses is then applied to the Phillips' Core: Portable X-ray fluorescence spectrometry (pXRF) of the entire core; X-ray computed tomography (CT) imaging and XRF-scanning of a segment of the core; optical petrography, μXRF, high resolution scanning electron microscopy (SEM) with cathodoluminescence (CL) and energy-dispersive spectrometry (EDS), and automated SEM-EDS mineral analysis (QEMSCAN) of petrographic thin-sections; and whole-rock inductively coupled plasma-mass spectrometry (ICP-MS), ICP-atomic emission spectrometry (ICP-AES) and whole-rock isotope analyses of powdered samples. To avoid potential

**Table 1. Analytical techniques applied to the Phillips' Core and Salisbury Museum Core from Stone 58 at Stonehenge (see main text for details of abbreviations and Fig 6 for details of sampling of the Phillips' Core).**

| Analytical technique | Analyst(s) | Salisbury Museum Core | Phillips' Core | | | | | |
|---|---|---|---|---|---|---|---|---|
| | | | Whole core | Section 2–3 of core (before sampling) | 3 × larger rock fragments | 2 × smaller rock fragments | Polished thin-section Set A | Polished thin-section Set B |
| Rock description and sedimentary logging | DJN, TJRC | x | x | | | | | |
| X-ray CT imaging | MD | | | x | | | | |
| Optical petrography | RAI, DJN, TS, JSU | | | | | | x | x |
| QEMSCAN | DP, MRP | | | | | | | x |
| High-resolution SEM/CL/EDS | TS, DJN, TJRC, JMH, NW | | | | | | | x |
| Portable XRF | TJRC | | x | | | | | |
| XRF core scanner | MD | | | x | | | | |
| Micro-XRF | SGo | | | | | | | x |
| ICP-MS and ICP-AES | DJN, TJRC | | | | x | | | |
| Whole-rock isotope analyses | JAE | | | | | x | | |

confusion between automated and non-automated SEM-EDS mineral analyses, we refer to the former by the trademarked name QEMSCAN (an abbreviation standing for quantitative evaluation of minerals by scanning electron microscopy). Too few zircon grains of sufficiently large size were identified in samples to permit statistically reliable zircon dating.

The paper unfolds by first detailing the history of the Phillips' and Salisbury Museum cores. We then review the results of each technique in turn before drawing the key findings together in section 5. We adopt this approach so that each set of results can be evaluated in isolation and—recognising that not all future investigations will have access to the same suite of techniques—that subcomponents of the dataset can be easily compared in follow-on studies. We recognise that some techniques offer better image, spatial, elemental or spectral resolution than others, but by including all here we provide methodological insights that future studies may want to pick up on when evaluating the relative merits of different approaches. In characterising the cores from Stone 58 –situated in the context of wider investigations into the nature and variability of sarsens at Stonehenge and elsewhere in the UK [e.g. 3, 5, 15–17]–we aim to develop a full geological picture of the main stone type used to build what is arguably the most important Late Neolithic monument in Europe.

## 2. History of the cores from Stone 58

The following account was compiled through research into Ministry of Works (MoW) archive files, held by the National Archives (Kew, London) and at the Historic England Archives in Swindon, UK, and via communication with the Phillips family.

### 2.1. Drilling of the cores in 1958

During a restoration programme at Stonehenge in 1958, three sarsen stones that had fallen in 1797 [uprights 57/58 and lintel 158, forming Trilithon 4 in the trilithon horseshoe; 3] were re-erected. This was done with the express intention of making the monument more intelligible to visitors, but also to protect supposed markings and carvings on Stone 57 that were thought to be under threat by people climbing on the stone.

Site preparations began in February 1958. Fractures in Stone 58 had been noted in a report by the Chief Architect T.A. Bailey, who had recommended that metal dowels be inserted once the stone was erected. However, a press release dated 11 June 1958 from the MoW states that Stone 58 "was checked for cracks by Harwell scientists using radioactive sodium. Their conclusion was that surface fissures do not extend through the width of the massive slab, but very great care will have to be taken in getting it upright" (MoW Registry File AA 71786/2R Part 1: Note from Mr Bailey relating to the drilling of Stone 58).

On 12 June 1958, when the work of restoring the trilithon was nearing completion, the Ancient Monuments Board visited Stonehenge to inspect the operations. According to a draft of their annual report for that year:

"We noted that the known longitudinal fracture of Stone 58, one of the uprights of the trilithon, had proved, on being raised, to be more extensive than had been suspected and we endorsed the Ministry's proposal to reinforce the stone. This was done by drilling through the stone and inserting metal bars tied by plates; the bars and plates were then hidden by small plugs of stone cut from sarsen fragments found during excavations associated with the main operation"

(MoW Registry File AA 71786/2R Part 2, 16. Ancient Monuments Board for England, First Draft of Report for 1958).

A note from Mr Bailey dated 15 August 1958 stated that the 'very badly fractured' Stone 58 needed to be drilled in three places with 1¼ inch (31.8 mm) diameter holes and tied together:

"The bolt heads and 3" [76.2 mm] diameter washer plates would be recessed 2½" [63.5 mm] into the surface on this stone and the holes plugged with 3½" [88.9 mm] diameter stone discs which after a few years should be almost unnoticeable. I am now therefore going ahead and will be drilling Stone 58 on Tuesday next. The work is being done by Messrs. L. M. Van Moppes (Diamond Tools) Ltd. of Basingstoke and the Chief Inspector will be present"

(MoW Registry File AA 71786/2R Part 2, 16, as above).

Three cores were drilled by Van Moppes on 20 August 1958 horizontally through the upright stone to the specifications noted above (Fig 2). Plugs were placed in the countersunk drill holes following the installation of the metal ties on 30 September 1958 (MoW Registry File AA 71786/2R Part 2, 9. Note from Mr Bailey relating to costs of re-erection of trilithon). Mr Robert Phillips, an employee of Van Moppes, recalled that he attended the site during the operations and that the drill operator was a Mr R. Berridge (information from the Phillips family by e-mail to Abigail Coppins, English Heritage Curator, 8 June 2018). The Van Moppes company was given permission by the Ministry of Works to retain one of the cores, which was entrusted to Mr Phillips.

## 2.2. Return of the Phillips' Core to the UK

Van Moppes were clearly proud of their involvement in restoration work at Stonehenge, writing about it in an unpublished company report ("Diamonds in the Service of Industry", LM Van Moppes & Sons Ltd) and commissioning a watercolour of the works (Fig 3). Both the painting and images in the report suggest that coring proceeded from the exterior-facing (i.e. northwest-facing) surface of Stone 58. The Phillips' Core hung in a protective Perspex tube in Robert Phillips' office in Basingstoke, together with the watercolour, until 1976 when he retired, taking the core and painting with him (with permission of Van Moppes). These accompanied him when he emigrated to the USA in 1977, and subsequently travelled with him from Rochester (New York) to Chicago (Illinois), Ventura (California) and finally to Aventura (Florida). Approaching 90 years of age, Robert was keen that this important artefact should be returned to the UK. Following his son Robin's contact with Historic England and English Heritage, arrangements were made for a Gander & White packing case to be delivered to Robert's home during a visit by Lewis Phillips in 2018. Lewis returned to the UK with the core on 2 May 2018, and it was delivered into the care of English Heritage on 31 May 2018 (Fig 4). The Phillips' Core is now archived in the English Heritage Collections Store at Temple Cloud near Bath, UK (English Heritage accession numbers 88371912.1 to 88371912.6).

## 2.3. Documenting and subsampling the Phillips' Core

All necessary permissions were obtained for the described study, which complied with all relevant regulations. The Phillips' Core from Stone 58 was inspected at the English Heritage Collections Store, Temple Cloud, on 27 March 2019. The core had broken into six sections at some point in its history, each with a diameter of 25 mm but ranging in length from ~7 cm to ~29 cm (Fig 5A). Each section had been labelled on its ends with a marker pen to indicate the correct sequence of pieces. Refitting the sections showed that the core was complete, with the presence of dead lichen at either end indicating that the core had penetrated the full thickness of the sarsen upright. The refitted sarsen core measured 108 cm in length. Inspection of the

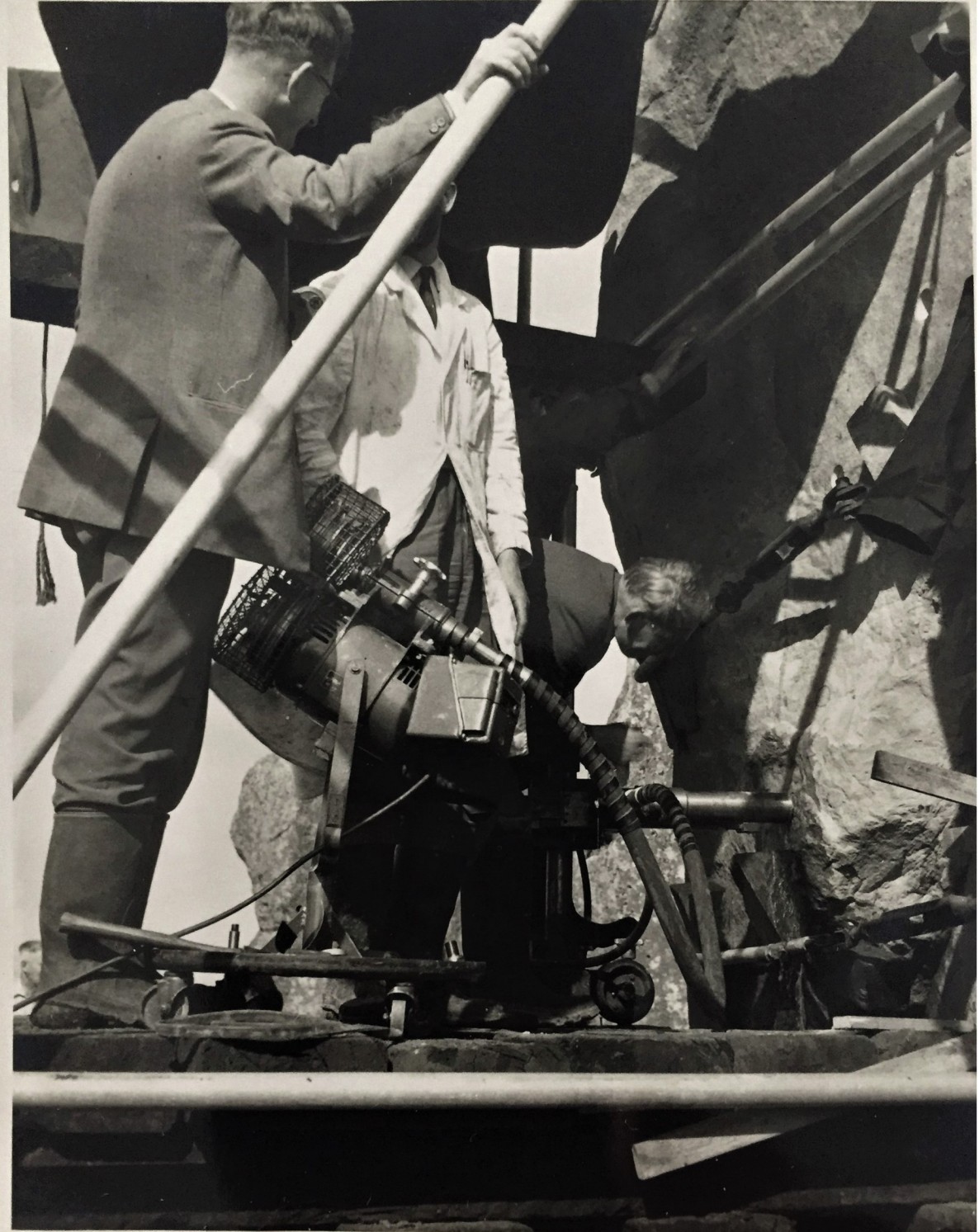

**Fig 2. Drilling work on Stone 58 at Stonehenge by Van Moppes Ltd in August 1958, with Mr Robert Phillips pictured left.** Permission was obtained from Mr Lewis Phillips for the image of his late father to appear in this picture and for him to be identified by name. This image is reproduced under a CC BY 4.0 license, with permission from Lewis Phillips, original copyright (2020).

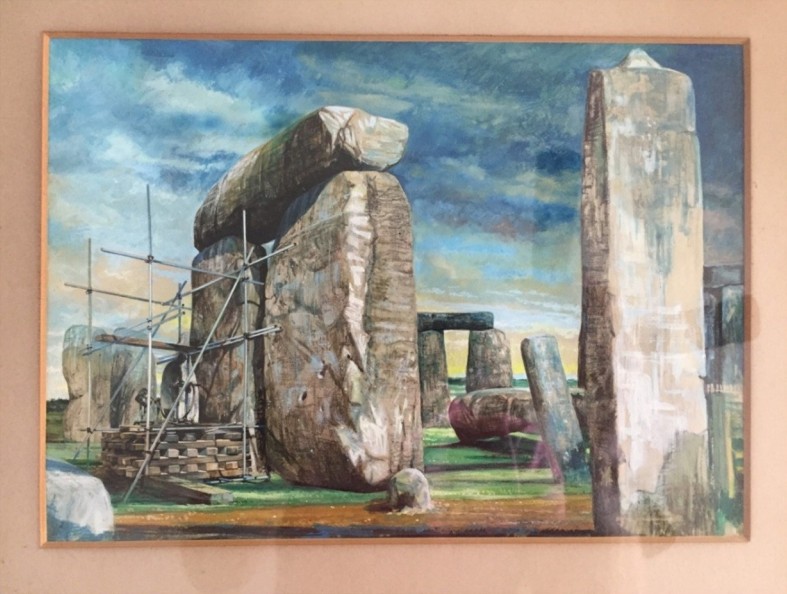

**Fig 3. Watercolour painting commissioned by Messrs.** L.M. Van Moppes (Diamond Tools) Ltd., now in the possession of the Phillips family, showing coring operations on Stone 58 of Stonehenge in 1958. This image is reproduced under a CC BY 4.0 license, with permission from Lewis Phillips, original copyright (2018).

surfaces of the five fracture planes within the core revealed that three (marked by pale brown staining; Fig 5A) had most likely followed lines of inherent weakness in the original sarsen. The other two, unstained, fractures had occurred either during the process of core extraction or later handling/transport.

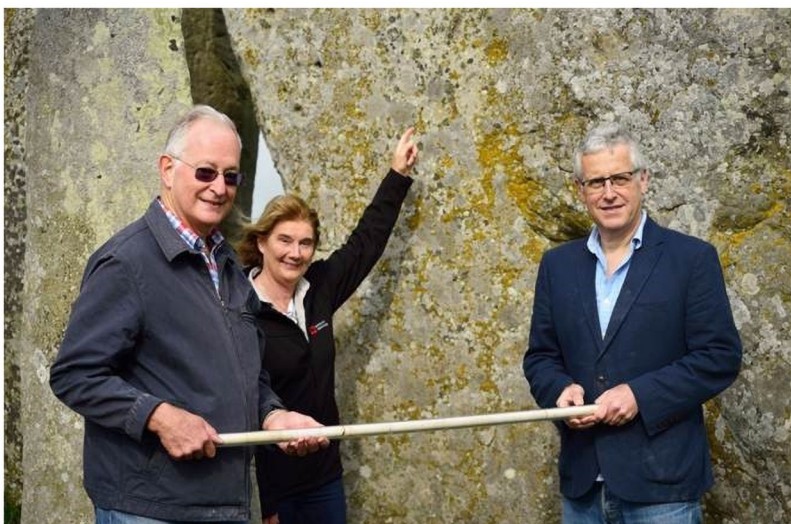

**Fig 4. Lewis (left) and Robin Phillips (right) at Stonehenge, handing over the 'Phillips' Core' from Stone 58 to Senior Property Curator, Stonehenge, Heather Sebire (pictured pointing at the position from which the core was drilled.** Permission was obtained from the individuals pictured to appear in this image and to be identified by name. This image is reproduced under a CC BY 4.0 license, with permission from English Heritage, original copyright (2018).

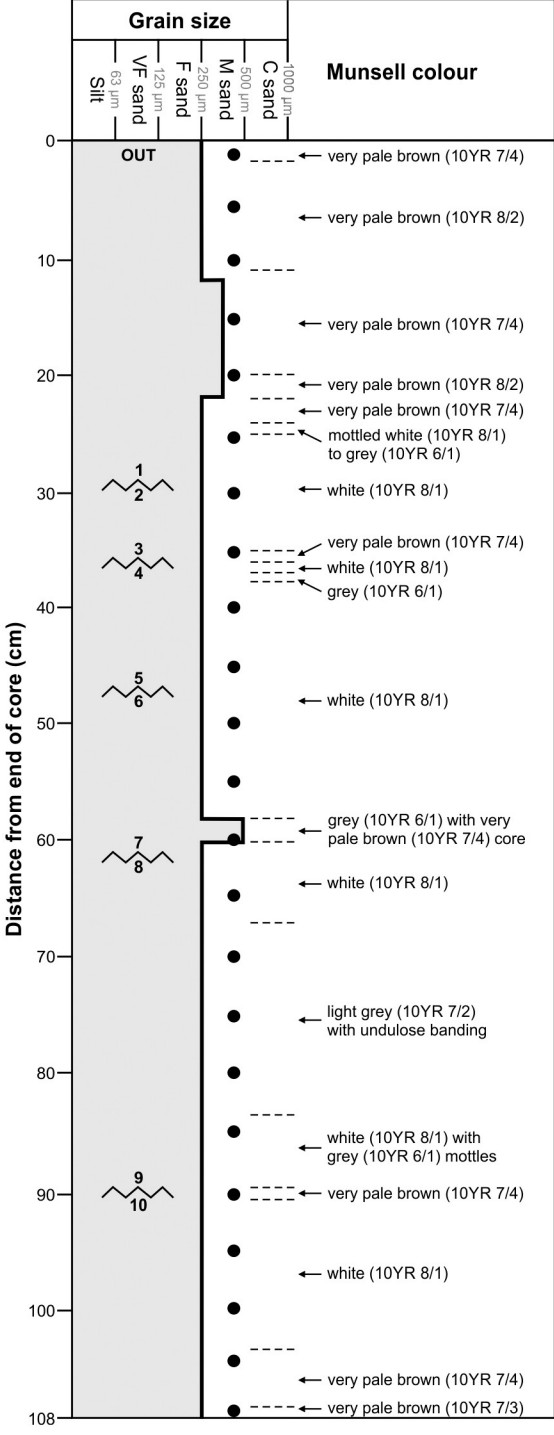

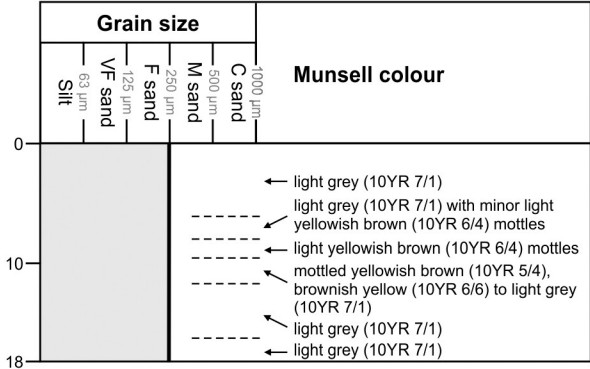

**Fig 5. Sedimentary logs of (A) the Phillips' Core and (B) Salisbury Museum Core from Stone 58 at Stonehenge.** Grain size and Munsell colour are plotted with distance from the end of each core. Letter and numbers shown at the end of each section of the Phillips' Core (i.e. OUT to 10) are those written with marker pen on the original core (see text). Section 2–3 of the Phillips' Core from 29–36 cm was subject to further detailed petrographical, mineralogical and geochemical analyses.

With permission of Martin Allfrey, Senior Curator of Collections (West) at English Heritage, section 2–3 of the core between 29 and 36 cm (English Heritage accession number 88371912.2) was subject to further analyses. To sample the section, the 67 mm-long piece was first cut along its long axis at the University of Bristol on 13 May 2019 using a diamond saw to form two semi-cylinders (Fig 6A), one of which was retained by English Heritage. The remaining semi-cylinder was transported first to the British Geological Survey (Keyworth) for CT imaging and geochemical scanning and then to the Open University (Milton Keynes).

At the Open University, the semi-cylinder of sarsen was cut laterally into three equal-sized subsamples using a diamond saw (Fig 6B). The curved uppermost part of each subsample was sawn off and further subsampled (Fig 6C), with the three larger fragments sent to ALS Minerals (Seville, Spain) for major and trace element analysis and two of the three smaller fragments to British Geological Survey for whole-rock isotope analysis (Fig 6D). One rectangular face of each of the remaining three subsamples was lapped smooth and glued to a glass microscope slide. The three mounted subsamples were cut again using a diamond saw, this time parallel to the glass slide. The offcut sarsen slices were then lapped smooth and mounted onto a second set of glass slides (Fig 6E).

The glass-mounted subsample fragments were processed at the Open University to produce two parallel sets of three polished thin sections, each set covering the full rectangular surface area of the original sarsen semi-cylinder. One set (SH1A, SH2A, SH3A) was used for standard optical-microscopic petrography and the other (SH1B, SH2B, SH3B) primarily for electron-microscopic analyses. Glass cover slips were not applied to either set of thin-sections. Optical microscopy datasets were also obtained from the second set of sections at the Natural History Museum. Further preparation of the entire sample set (Fig 6F) is described in sections 3 and 4.

## 2.4. Discovery and documentation of the Salisbury Museum Core

The return of the Phillips' Core to the UK was announced in a press release via the English Heritage website on 8 May 2019 [18]. This included a general request from Heather Sebire (Senior Property Curator, Stonehenge): "The other two Stonehenge cores may still be out there somewhere and if anyone has any information, we'd love to hear from them." On 30 July 2019, Martin Allfrey (English Heritage) received an email from Adrian Green, Director of Salisbury Museum, stating that museum staff had discovered a small section of a second sarsen core during detailed cataloguing of their Stonehenge collection. The core was found in a box marked *3x Stonehenge Stones from 'Treasure Box'*, alongside a fragment of 'Altar Stone' and a piece of polished bluestone (a spotted dolerite) from the monument. The core had a retrospective accession number (2010R.240) given in 2010 during an audit of collections, but how and when it came to be at the museum was unknown (information given to DJN in an e-mail from Adrian Green, 29 January 2020). The whereabouts of the rest of the Salisbury Museum Core, and of the third core drilled from Stone 58, is similarly unknown.

The Salisbury Museum Core was inspected in the collection storeroom of the museum on 24 January 2020. The core consisted of a single cylindrical piece of sarsen (177 mm length, 25 mm diameter) with the label *Stonehenge Sarsen– 1958* written along its long-axis. The cylinder was clearly a section of a much longer original core—both ends exhibited fresh fractures, and, unlike the Phillips' Core, lichen was absent. It was not possible to ascertain the position of the Salisbury Museum Core relative to the overall thickness of Stone 58, nor to establish the correct orientation of the fragment in relation to the Phillips' Core. Permission was not given to sub-sample.

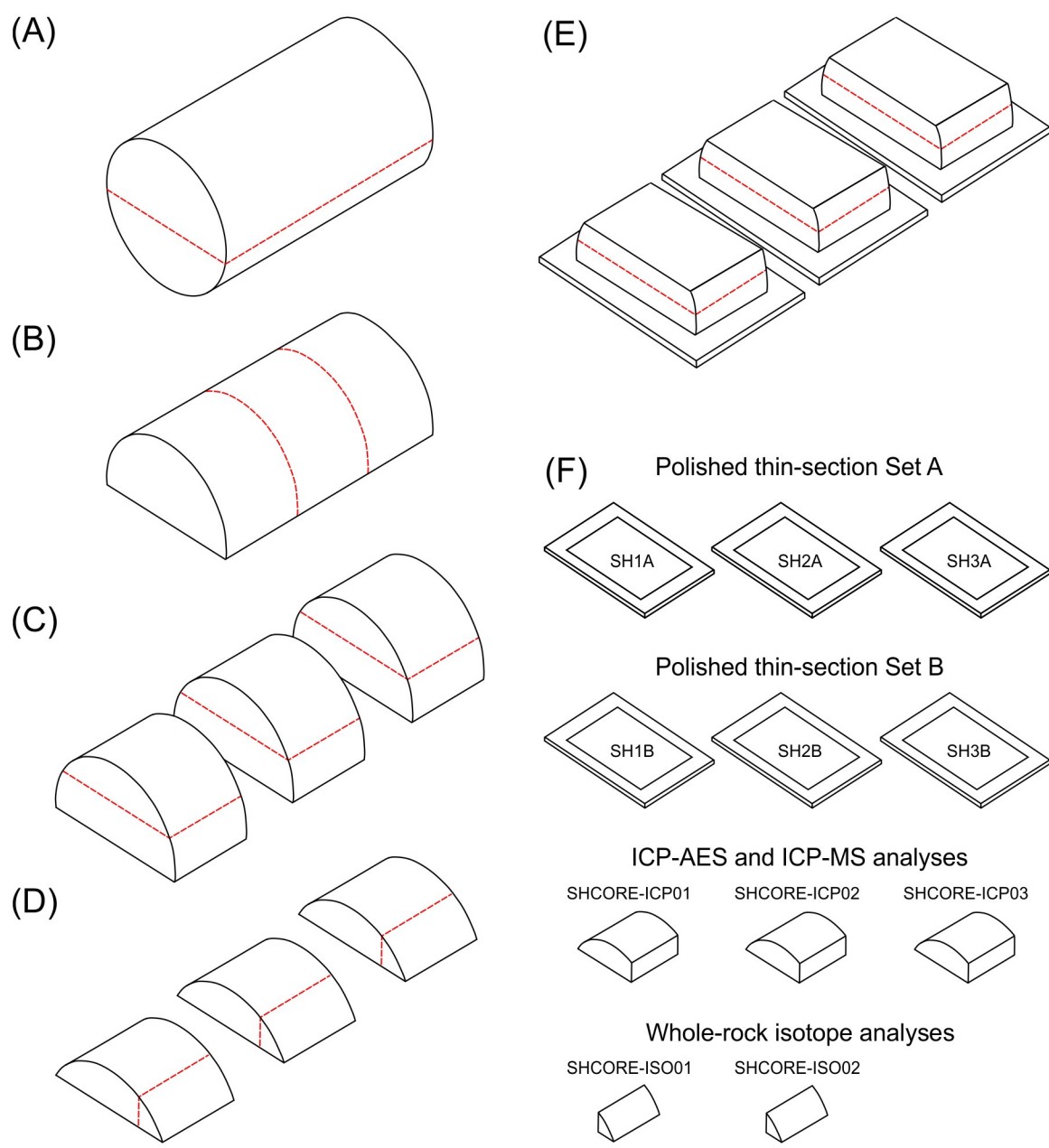

**Fig 6. Schematic representation showing how the 67 mm long section 2–3 of the Phillips' Core from Stone 58 at Stonehenge was (A) cut (dashed lines) and prepared (B-E) in order to produce two sets of three polished thin sections, three samples for whole-rock major and trace element analysis, and two samples for whole-rock isotopic analysis (F).** See text for full description.

## 3. Petrography of the sarsen cores

The petrography of the two sarsen cores was assessed using five complementary approaches. Standard rock description and sedimentary logging was carried out on both cores (prior to subsampling in the case of the Phillips' Core; see section 2.3). CT imaging was undertaken of the 67 mm long semi-cylinder of sarsen cut from section 2–3 of the Phillips' Core (Fig 6A). Analysis of the polished thin-sections from section 2–3 of the core included: (i) standard petrographic description (thin-section set A), (ii) quantitative analysis of sarsen mineralogy using

QEMSCAN, and (iii) SEM, CL and EDS analysis of host mineral grains and cements (thin-section set B).

### 3.1. Rock description

**3.1.1. Methodology.** Description of the two sarsen cores from Stone 58 at Stonehenge followed standard sedimentological procedures designed to provide a macroscale record of rock properties. Each core was logged visually to identify changes in grain size, mineralogy and colour along its length. Grain size was estimated using a Geo Supplies Ltd grain size card. Mineralogy was determined visually using a ×20 hand lens. Colour variation was assessed using a Munsell colour chart, having first moistened the rock surface with water.

**3.1.2. Rock description results.** The Phillips' Core from Stone 58 is a well-indurated, grain-supported, texturally mature and predominantly fine-grained (i.e. 0.125–0.250 mm grain size range) silcrete—in geological terms it would be described as an orthoquartzite or quartz arenite. The results of sedimentary logging are summarised in Fig 5A, with a high-resolution image of section 2–3 of the core shown in Fig 7 to illustrate the typical rock properties. The majority of the 108 cm core comprised structureless quartz-cemented fine-to-medium grained sand, the exception being slightly coarser laminae of medium grained sand (0.25–0.50 mm range) from 12–22 cm and one thin lamina of medium-to-coarse (0.5–1.0 mm range) grained sand from 58–60 cm. The mineralogy of the sand-sized grains within the core showed little variability, being dominated by quartz. Minor opaque minerals were distributed throughout the core. The main variability was in terms of colour. The majority of the core exhibited wet Munsell colours ranging from white (10YR 8/1) to grey (10YR 6/1). There was, however, variability in the degree of iron hydroxide staining present. The section of the core from 0–22 cm was very pale brown in colour with some mottling and banding, with the colour varying between 10YR 7/4 and 10YR 8/2. As noted in section 2.3, other zones of very pale brown (10YR 7/4) iron hydroxide staining were present adjacent to three of the five fracture surfaces at around 36, 60 and 90 cm.

The Salisbury Museum Core (Fig 5B) exhibits very similar properties to the Phillips' Core, again being a well-indurated, structureless, grain-supported and texturally mature silcrete comprising predominantly fine-to-medium grained quartz in a quartz cement. Minor, fine sand-sized, dark-coloured minerals are also present along with minor pyrite. Again, the main variability along the core was in terms of colour. The background colour of the Salisbury Museum Core was slightly darker than the Phillips' Core (light grey; 10YR 7/1). Where present, mottles had a more intense chroma, ranging from yellowish brown (10YR 5/4) to brownish yellow (10YR 6/6), with the most intense-coloured mottling evident between 8 and 11 cm along the core. Fig 8 illustrates the typical rock properties.

### 3.2. X-ray computed tomography (CT) imaging

**3.2.1. Methodology.** X-ray computed tomography (CT) imaging was used to determine the presence/absence of sedimentary structures within section 2–3 of the Phillips' Core and to assess the distribution of fractures and pore spaces. CT images were collected using a Geotek Ltd MSCL-RXCT core scanner at the British Geological Survey, Keyworth, UK. CT data were acquired as a stack of 2D radiographic images with a pixel resolution of approximately 30 μm. The 2D images were reconstructed into a 3D volume using the Geotek Ltd Image Reconstructor software version 1.0.0.0, with corrections applied for beam-hardening and ring artefacts—phenomena produced by the selective attenuation of lower energy photons as X-rays pass through an object [19]–to improve the signal to noise ratio. The 3D image dataset was

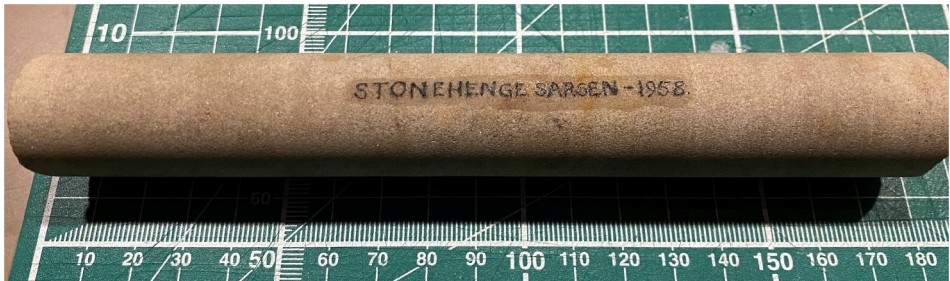

**Fig 7. Image of section 2–3 of the Phillips' Core from Stone 58 at Stonehenge.** The right-hand end of the core segment represents a natural fracture in the original sarsen, with the thin band of iron hydroxide staining running diagonally from ~40 to ~50 mm mirroring the fracture surface. The left-hand end of the core represents a break developed either during or after drilling. The grey diagonal band running from ~10 to ~0 mm is residual metal from the diamond saw blade smeared onto the surface of the sarsen during cutting. This image is reproduced under a CC BY 4.0 license, with permission from British Geological Survey, original copyright (2019).

simulated and analysed using the digital rock and core analysis software PerGeos version 2019.4 from Thermo Scientific.

**3.2.2. CT imaging results.** Grey-scale CT images (B-C) are shown alongside an equivalent optical image (A) in Fig 9. These illustrate the sample matrix and mineral constituents; the

**Fig 8. The Salisbury Museum Core from Stone 58 at Stonehenge.** This image is reproduced under a CC BY 4.0 license, with permission from David J. Nash, original copyright (2020).

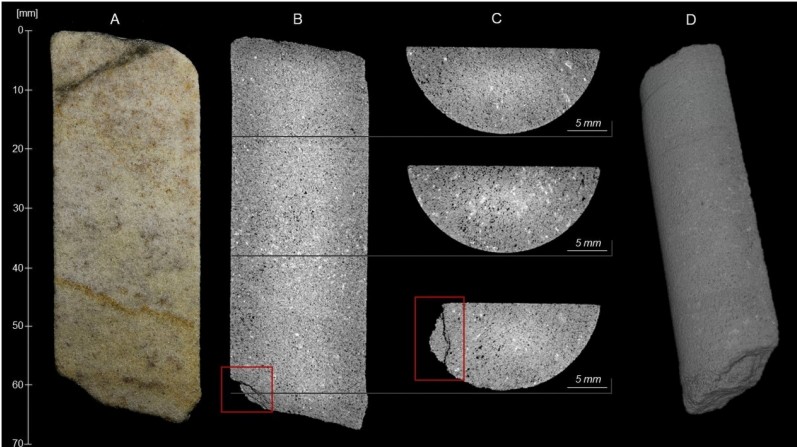

**Fig 9. Optical (A) and Computed Tomography (B-C) images of section 2–3 of the Phillips' Core from Stone 58 at Stonehenge.** Distance along the sample is measured relative to the fracture plane between sections 1 and 2 of the Phillips' Core (see Fig 5). Dark grey to black tones in the CT images indicate low density areas (e.g. pores, fractures), while light grey to white tones indicate high density areas (e.g. mineral constituents). A 3D reconstruction of sample (D) with full simulation is provided in the S1 Movie. These images are reproduced under a CC BY 4.0 license, with permission from British Geological Survey, original copyright (2019).

porosity appears black. The CT scan confirms the results of the visual rock description (section 3.1.2), indicating that the sarsen in section 2–3 is homogenous, displaying no bedding, lamination, nor any other fabrics at the scale of imaging. The dispersed band of slightly coarser minerals running across the core from ~35 mm to ~40 mm may be a zone containing larger quartz grains. An open fracture is observed at ~60 mm along the sample (Fig 9B and 9C). The thin band of iron hydroxide staining running diagonally from ~40 to ~50 mm in image A is visible as a thin zone of denser cementation in image B. The 3D image dataset was simulated (Fig 9D) using the digital rock and core analysis software PerGeos. A 3D simulation of the sample can be found in the S1 Movie. The full image dataset is archived at the Archaeology Data Service.

## 3.3. Optical petrography

**3.3.1. Methodology.** Optical microscopy was used to provide a qualitative assessment of the petrography and mineralogy of Stone 58. Polished thin-sections SH1A, SH2A and SH3A from the Phillips' Core were investigated initially using a ×20 hand lens and Geo Supplies Ltd grain size card. The petrography of each section was analysed under plane- and cross-polarised transmitted light using a Carl-Zeiss Amplival pol u dual-purpose microscope (×6.3 and ×12.5 objectives with ×12.5 eye pieces giving overall magnifications of ×80 and ×155 respectively). Each section was then investigated in reflected light using a Zeiss Universal reflected light microscope (with ×4.5 air, ×16 oil and ×40 oil immersion lens). Mineral identification in transmitted and reflected light was made following standard optical mineralogy texts [e.g. 20, 21] and atlases [22], with petrographic descriptions also following standard protocols for sedimentary rocks [23] and silcretes [1]. All mineral phases greater than 2 μm diameter were identified. Note, however, that the fine-grained nature of the $TiO_2$ phases present sometimes prevented further discrimination. Where a $TiO_2$ phase could be identified with certainty it is given a mineral name in section 3.3.2; where not, it is simply referred to as a '$TiO_2$ mineral'. Polished thin-sections SH1B, SH2B and SH3B were also analysed by automated polarised light microscopy at the Natural History Museum, London, to produce whole-section mosaic images for archiving with the Archaeology Data Service. A ZEISS Axio Imager.M2m light microscope with

motorised stage was used to obtain these images with a resolution of ~14,000×12,000 pixels and a pixel size of ~2 μm.

**3.3.2. Optical petrography results.** Microscopic analyses of thin sections SH1A, SH2A and SH3A demonstrate that the sarsen is a grain-supported, very well-sorted groundwater silcrete that displays no bedding or any other fabric. It comprises silica-cemented sub-rounded to rounded, detrital quartz grains with a mean diameter of 187 μm (corresponding to fine sand, consistent with rock descriptions; section 3.1.2) and minor proportions of other minerals set in an optically-continuous syntaxial quartz overgrowth cement (see Fig 10). The silcrete is very well cemented but with primary pore space present where voids are not completely infilled with quartz overgrowth cement. We identify the sarsen as a groundwater silcrete (as opposed to a pedogenic silcrete) based on its simple micromorphology, textural homogeneity and lack of pedogenic features such as geopetal and colloform structures (see discriminating criteria in [24]). The tabular morphology of most sarsen boulders at Stonehenge is also consistent with this interpretation.

*Host sediments*. The quartz grains that make up most of the sarsen host sediment are monocrystalline with a restricted size range. Two populations of quartz are present. The overwhelming majority of quartz grains are clear and exhibit undulose extinction (i.e. different parts of the quartz crystal go into extinction as the microscope stage is rotated under cross-polarised light; Fig 10A and 10B). Most quartz grains are inclusion-free but rare grains enclose very fine-grained carbonate or very small (10 μm diameter) euhedral zircon, pale-coloured $TiO_2$ minerals, muscovite, biotite, tourmaline, $20 \times 1$ μm size graphite, 2–10 μm diameter hematite, 2–5 μm diameter magnetite and $40 \times 5$ μm size ilmenite. The most common inclusion is fine-grained (2–20 μm diameter), unaltered, framboidal pyrite or single cubic crystals, within small aggregates; some of the pyrite is altered to limonite. Untwinned feldspar may be present in section SH1A, as two rectangular grains exhibiting slight alteration along their cleavage planes were recognised. No alteration products of feldspar—specifically fine-grained mica or kaolinite—were observed.

Minor rock clasts (80–200 μm diameter) are present in all three thin-sections. All clasts are highly siliceous and comprise: (i) microquartz (most likely chert or flint, including one clast in SH1A with spherulitic quartz); (ii) slightly coarser-grained polycrystalline quartz (likely a fine-grained sandstone); and (iii) quartz with metamorphic textures (ribbon quartz). Clasts contain 1–5 μm long, pale-coloured $TiO_2$ grains or very fine-grained (<1–2 μm long) hematite, as well as rare, 5–15 μm diameter zircon and 20 μm diameter $TiO_2$ phases. An example of rock clast type (iii) is visible bottom centre right of Fig 10D.

Few detrital accessory (<0.1% area) minerals are present in the thin-sections. Of these, zircon is the most common and occurs as rare euhedral to rounded, 10–60 μm (but up to $200 \times 60$ μm), highly zoned grains. Smaller (10–30 μm) high-relief, high-birefringent minerals that may be unzoned zircon are also present. Small zircons are commonly enclosed within single quartz grains. Other accessory minerals include: (i) rare, up to 180 μm diameter, green or green-brown, zoned, subhedral tourmaline (with one grain enclosing zircon); (ii) very rare, 20–60 μm diameter, hexagonal apatite; (iii) very rare, 20–50 μm diameter, euhedral spinel (present in thin-sections from SH1A and SH2A only), most likely chrome-rich magnetite/spinel or chromite (based on optical properties in reflected light and supported by QEMSCAN and SEM-EDS analyses; see section 3.4.2); (iv) rare, $100–200 \times 60$ μm kyanite laths; and (v) rare, 160 μm diameter, yellow staurolite, some grains enclosing very fine-grained graphite or small quartz inclusions.

*Titanium mineral phases*. Detrital and authigenic titanium minerals are the most abundant opaque phases within all three thin-sections. Detrital grains are rounded to sub-rounded and more common than authigenic $TiO_2$ phases. However, discrete, detrital rutile grains were not

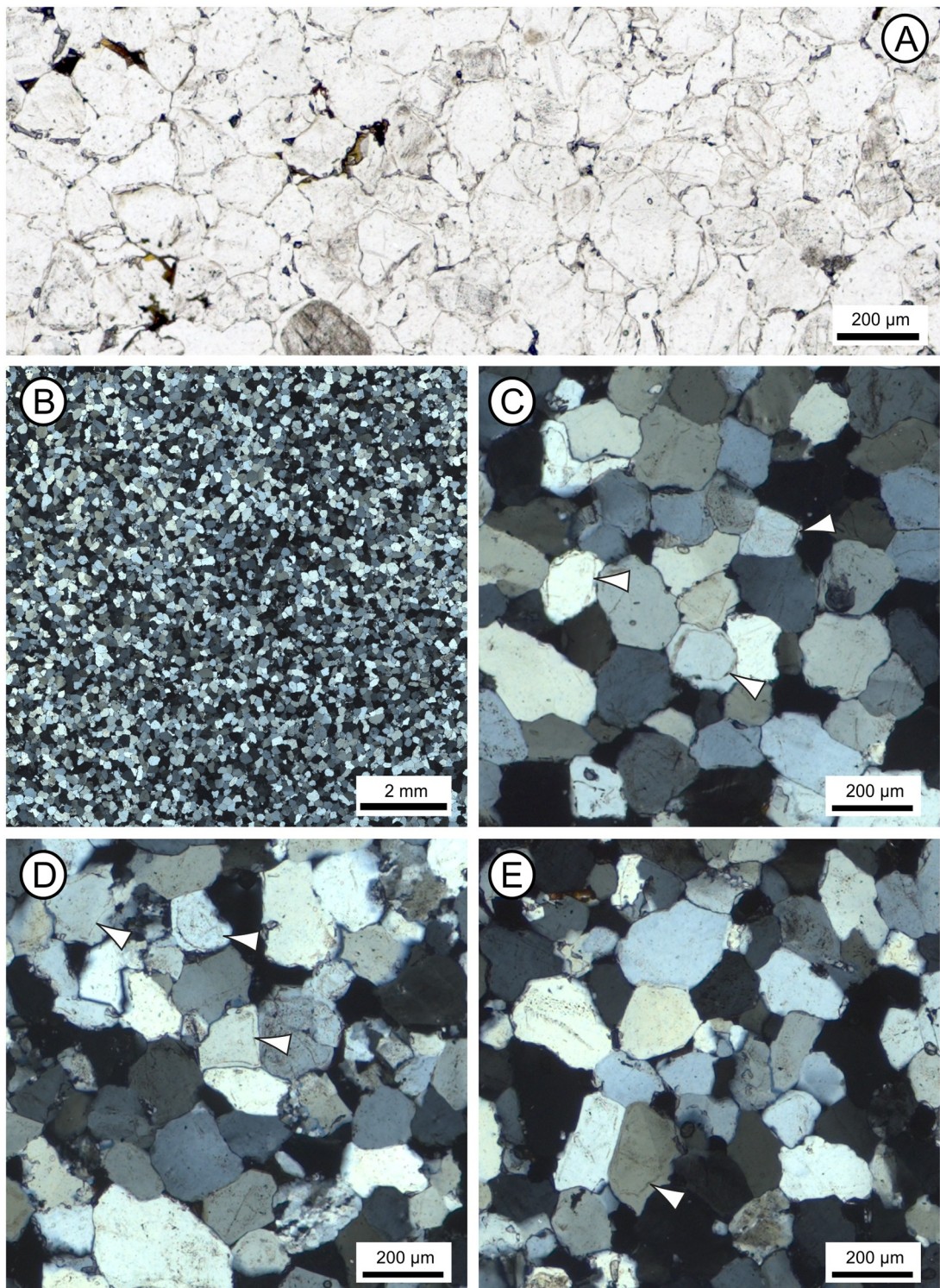

**Fig 10. Optical images from polished thin-sections taken under plane- (A) and cross-polarised (B-E) light, illustrating the petrography of section 2–3 of the Phillips' Core from Stone 58 at Stonehenge.** (A) Detail of thin-section SH1B showing the typical sarsen fabric comprising quartz grains cemented by quartz overgrowths, with late-stage Fe-Ti minerals lining and/or infilling some void spaces. (B) Overview of thin-section SH2B showing the pervasive nature and uniformity of syntaxial optically-continuous quartz overgrowth cements. (C-E) Details of thin-sections SH1B (C), SH2B (D) and SH3B (E), showing host quartz grains, some of which enclose accessory minerals; dust lines (arrowed) mark the margin between some quartz grains and the quartz overgrowth cement in images C-E. These images are reproduced under a CC BY 4.0 license, with permission from The Trustees of the Natural History Museum, original copyright (2019).

identified. Sparse, 40–200 μm diameter, rounded, poorly crystalline, fine-grained $TiO_2$ pseudomorphs (after iron titanium oxide mineral grains) are present. These exhibit yellow to orange internal reflections and some show signs of recrystallisation. Some grains are zoned—with darker, less altered, cores lacking internal reflections set within $TiO_2$ rims—or have orange-red cores within paler rims and thin authigenic $TiO_2$ mineral overgrowths. Most iron titanium oxides are too altered to be identified but some with a cubic habit may be former magnetite and those that are more tabular, former ilmenite. Very rare, altered ilmenite, 180–200 μm in length, exhibits a 'speckled' appearance and a $TiO_2$ rim or has altered to box-work leucoxene; magnetite-ilmenite intergrowths have been altered to well- and poorly-crystalline $TiO_2$. Little authigenic $TiO_2$ is present in the sections. Where it does occur, it is visible as 5–20 μm but up to 40 μm long, single crystals with white to pale yellow internal reflections, or as thin rims on earlier $TiO_2$ or quartz grains. The characteristics of titanium mineral phases are discussed further in section 3.5.2, with examples visible in Fig 14.

*Silica cements*. The host sediment grains within all three thin-sections are cemented by pervasive authigenic, syntaxial quartz overgrowths (occupying >99% of the cement area) that include euhedral terminations into void spaces. Some rounded quartz grains show a dust line between the detrital grain and its overgrowth (e.g. Fig 10C–10E), but many do not. A number of grains have cloudy or pale brown cores, probably fine-grained fluid inclusions rather than opaque mineral inclusions, within their clear overgrowths and some are partially surrounded by microquartz mosaics. Further details of these cements are provided in section 3.5.2 where the results of cathodoluminescence analyses are presented.

*Late-stage void linings and fills*. Late-stage void-lining minerals are present in all three thin-sections (e.g. Fig 10A). However, complete or almost complete void-fills mainly occur in thin-sections SH2A and SH3A. In terms of mineralogy, limonite forms thin, locally botryoidal, 2–5 μm but up to 10 μm wide linings to angular void spaces, post-dating authigenic quartz overgrowth cements. Some voids contain 20–60 μm diameter botryoidal limonite, which has probably replaced precursor sulfide including framboidal pyrite. Elsewhere, limonite forms $40 \times 5$ μm size laths within void spaces. Limonite also encloses altered iron titanium oxides in some voids but does not appear to be associated with their alteration. The relationship between Fe- and Ti-minerals in voids is considered further in section 3.5.

## 3.4. Automated SEM-EDS mineralogy (QEMSCAN)

**3.4.1. Methodology.**   The mineralogy of polished thin-sections SH1B, SH2B and SH3B from the Phillips' Core was analysed quantitatively via automated SEM-EDS using an FEI Quanta 650 QEMSCAN platform running at 20 kV with a 10 nA beam current and fitted with $3 \times 30$ $mm^2$ Bruker XFlash EDS detectors. QEMSCAN has been used widely to determine the mineralogy of sedimentary rocks [e.g. 25–27]. The method is a powerful and versatile tool in provenancing based on mineralogy [28, 29] and has been successfully applied to studies on the origin of Stonehenge debitage [including the Altar Stone; 30–32].

Each thin-section was measured using the QEMSCAN fieldscan mode [see 33 for a description of measurement modes] with energy dispersive spectra acquired at a 10 μm grid spacing. As the samples are virtually monomineralic, the area was optimised to scan as much of each polished section as possible (approximately $15 \times 24$ mm), with a total of between 2,708,539 and 3,170,220 individual ED spectra acquired per thin-section. The data were then processed using the iDiscover 5.4 software suite and each spectrum assigned to a defined mineralogical grouping based on its compositional signature [see 34 for a detailed method summary]. Output data includes modal mineralogy, false colour mineralogical maps and a range of other mineral and textural properties including mean mineral size (a textural index) and mineral

association (which minerals are in contact with each other). Mean mineral size data represent the average horizontal intercept lengths of each mineral in the image. Where, for instance, quartz grains are compacted and/or surrounded by quartz overgrowth cements, individual grains appear merged together and the true quartz grain size cannot be derived. However, for more widely separated phases, such as heavy minerals, the mineral size data represent an approximation of actual grain size.

**3.4.2. Automated mineralogy results.** Modal mineralogical data, expressed as area %, are presented in Table 2. Because approximately 3 million data points were acquired, data are reported to three decimal places; this is necessary to highlight the presence of ultra-trace (<0.01 area %) mineral phases within the samples that would otherwise be masked. The automated mineralogy results are entirely consistent with the petrographic descriptions in section 3.3.2, whilst providing fully quantitative data; note, however, that it is not possible to differentiate the various types of quartz (e.g. mono- or polycrystalline quartz) or between detrital or diagenetic quartz using this method. Measured quartz content is between 99.568 and 99.679 area %, while all other mineral phases comprise between 0.432 and 0.321 area % (note that these values exclude porosity). Hence, the mineralogy of the three thin sections is essentially invariant. Despite the almost monomineralic nature of the sarsens, textural features coupled with the trace mineral assemblage identified can provide important detail both in terms of the provenance and the diagenesis of the rock.

Quartz occurs as a framework of medium sand sized grains that appear partially merged in the mineralogical images (Figs 11 and 12). Many of the grain boundaries and pore walls are planar reflecting the presence of quartz overgrowth cement. Trace quantities of K-feldspar, plagioclase, muscovite and biotite are present. Rather than occurring as discrete detrital grains, examination of the images shows that these phases typically occur as tiny inclusions (<20 μm) enclosed within the detrital quartz. Similarly, oversized pores occur across all samples suggesting that labile phases may have dissolved and that only particles enclosed within quartz grains or surrounded by quartz cement (i.e. separated from the pore fluids) remain.

Overall, the sarsen samples are extremely clean. Aside from the quartz cement, a moderately connected intergranular pore network with between 7.2 (SH1B) and 9.2 area % (SH3B) porosity is preserved (Table 2). Small quantities of Fe-oxides/hydroxides and clay occur within the pore network. In line with the petrographic descriptions of thin-section set A (section 3.3.2), the Fe-oxides/hydroxides locally partially to completely line pore walls, including those cemented by quartz i.e. they post-date quartz cementation. Kaolinite, chlorite, and, to a lesser extent, illitic clays occur as sparse <10 μm particles that are finely disseminated throughout the pore network or enclosed within quartz overgrowths. Given this close spatial association with pore space, these clay minerals may be diagenetic in origin and possibly contemporaneous with quartz cementation. However, occasionally clay minerals (primarily chlorite) occur within rock fragments indicating that a proportion of the clay is detrital in origin. Other diagenetic phases identified within the pore network include scattered carbonates (calcite, dolomite, ferroan dolomite), pyrite and (in SH3B) baryte, but these minerals are represented by very sporadic, small crystals.

Detrital heavy minerals are isolated and sparse, but the assemblage is relatively diverse and includes small grains of Fe-oxide, Ti-oxide (some of which may be diagenetic), and Fe-bearing Ti-oxide, along with lesser amounts of tourmaline (both Fe- and Mg-rich compositions), kyanite and zircon. While the grains reported as kyanite in section 3.3.2 could potentially be andalusite or sillimanite (these minerals are polymorphs so cannot be separated based on SEM-EDS), the grain shape coupled with an optical assessment confirmed the identification as kyanite. Also identified, but not present in all samples, are sporadic grains of chromite (SH1B), apatite (SH1B and SH3B) and staurolite (SH2B and SH3B). Notably, the Fe-bearing Ti-oxides

**Table 2. Results of automated SEM-EDS (QEMSCAN) mineralogy.**

| Modal Mineralogy (area %) | SH1B | SH2B | SH3B |
|---|---|---|---|
| Quartz | 99.679 | 99.582 | 99.568 |
| K-feldspar | 0.001 | 0.001 | 0.001 |
| Plagioclase | 0.004 | 0.006 | 0.011 |
| Muscovite | 0.001 | 0.001 | 0.002 |
| Biotite | 0.001 | 0.001 | 0.001 |
| Kaolinite | 0.043 | 0.055 | 0.082 |
| Chlorite | 0.039 | 0.041 | 0.084 |
| Illite & illite-smectite | 0.005 | 0.006 | 0.009 |
| Fe-Illite & illite-smectite | 0.001 | 0.003 | 0.013 |
| Calcite | 0.007 | 0.007 | 0.017 |
| Dolomite | 0.006 | 0.005 | 0.003 |
| Ferroan dolomite | 0.002 | 0.002 | 0.001 |
| Fe-oxides | 0.147 | 0.192 | 0.078 |
| Chromite | 0.002 | 0.000 | 0.000 |
| Pyrite | 0.005 | 0.006 | 0.013 |
| Baryte | 0.000 | 0.000 | 0.001 |
| Ti-oxides | 0.020 | 0.038 | 0.041 |
| Fe-bearing Ti-oxides | 0.016 | 0.025 | 0.032 |
| Apatite | 0.001 | 0.000 | 0.001 |
| Kyanite | 0.001 | 0.008 | 0.010 |
| Mg-tourmaline | 0.011 | 0.007 | 0.008 |
| Tourmaline | 0.005 | 0.008 | 0.013 |
| Staurolite | 0.000 | 0.003 | 0.003 |
| Zircon | 0.002 | 0.004 | 0.008 |
| Undifferentiated | 0.000 | 0.000 | 0.000 |
| **Porosity (area %)** | 7.2% | 7.7% | 9.2% |
| **Number of EDS analyses** | 3170220 | 3103710 | 2708539 |

are comparatively Fe-poor and not ilmenite *sensu stricto* but instead the pseudomineral "leucoxene", which typically occurs as an alteration product associated with extended oxidation and Fe leaching of ilmenite [35].

In summary, based on the analysis of the three polished thin-sections, mineral abundance after quartz, whether detrital or diagenetic in origin, is as follows: Fe-oxides/hydroxides, kaolinite, chlorite, Ti-oxides, Fe-bearing Ti-oxides, calcite, Mg-tourmaline, tourmaline, pyrite, plagioclase, kyanite, illite and illite-smectite, Fe-illite and Fe-illite-smectite, zircon, dolomite, staurolite, muscovite, ferroan dolomite, K-feldspar, biotite, chromite, apatite and baryte. Grain size data show that all the phases (other than quartz) interpreted as detrital in origin have a mean grain size of typically 15–45 μm, apart from staurolite, which falls within the very fine sand size fraction (96 μm).

## 3.5. High resolution SEM-CL and SEM-EDS

**3.5.1. Methodology.** Polished thin-sections SH1B, SH2B and SH3B from the Phillips' Core were analysed at the Natural History Museum, London, by automated SEM. High-resolution SEM-CL and SEM-EDS were used to examine (i) the internal characteristics of quartz host grains, (ii) the internal structure of quartz overgrowth cements and (iii) the development of late-stage void linings and fills. Prior to analysis, each thin-section was coated with 10 nm

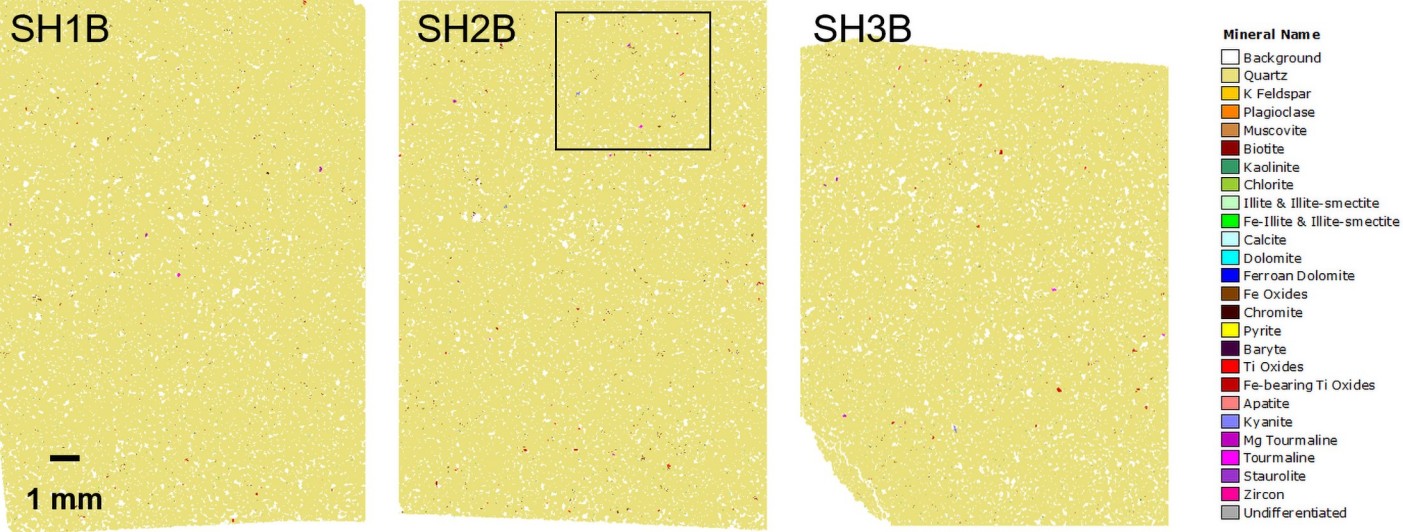

**Fig 11. Automated SEM-EDS (QEMSCAN) mineralogical maps for polished thin-sections SH1B, SH2B and SH3B from section 2–3 of the Phillips' Core.** The box indicates the location of the higher resolution mineralogical map shown in Fig 12.

carbon to prevent charging. CL and EDS analyses were performed using a Zeiss EVO LS15 SEM. Mosaic images on two representative areas for each sample were collected with a Gatan ChromaCL colour CL imaging system. To improve the spatial resolution for CL, the SEM was operated at an intermediate accelerating voltage of 10 kV, a beam current of 6 nA, and a working distance of 14.1 mm at a distance of ~1 mm from the CL detector. Four digital images (each 540×540 pixels, 320 nm pixel size) with three energy ranges, red (600–750 nm wavelength), green (475–580 nm) and blue (375–450 nm), and a composite red-green-blue (RGB) image were recorded for each field with a dwell time of 4 ms corresponding to a measurement time of ~20 min. Each area was analysed by 3×3 fields using automated stage control with an overlap of 10%. The nine individual RGB images were stitched into one mosaic using the Gatan DigitalMicrograph software option, DigitalMontage. During stitching, the best overlap fit for neighbouring images was calculated by cross correlation. Slight brightness adjustments were made to ensure that colours matched between neighbouring images to give a more balanced mosaic image and avoid unnecessary high contrast across the fields wherever possible.

Large area elemental maps were obtained with an Oxford Instruments Aztec EDS system with an XMax 80 mm² silicon drift detector. The SEM was operated at a working distance of 10 mm during elemental mapping. An accelerating voltage of 20 kV and a beam current of 3 nA resulted in an EDS pulse throughput of ~86,000 counts per second. Each thin section was analysed by 3317 to 4711 fields, with each field covering an area of 413×309 μm and analysed for 22 seconds. Secondary electron (SE) and backscattered electron (BSE) images were acquired at a resolution of 256×192 pixels, corresponding to a pixel size of 1.6 μm. EDS spectra were stored as hyperspectral imaging datasets at a resolution of 128×92 pixels, corresponding to a pixel size of 3.2 μm. Aztec software was used to stitch the individual fields into one hyperspectral imaging dataset with an EDS resolution up to 48 megapixels (Table 3). The distribution of elements is displayed as net intensity maps where the background has been subtracted and peaks with overlapping X-ray lines have been deconvolved.

High resolution BSE images of the thin sections were obtained using a Zeiss Ultra Plus field emission SEM with the large-area imaging software and hardware package ZEISS Atlas 5.

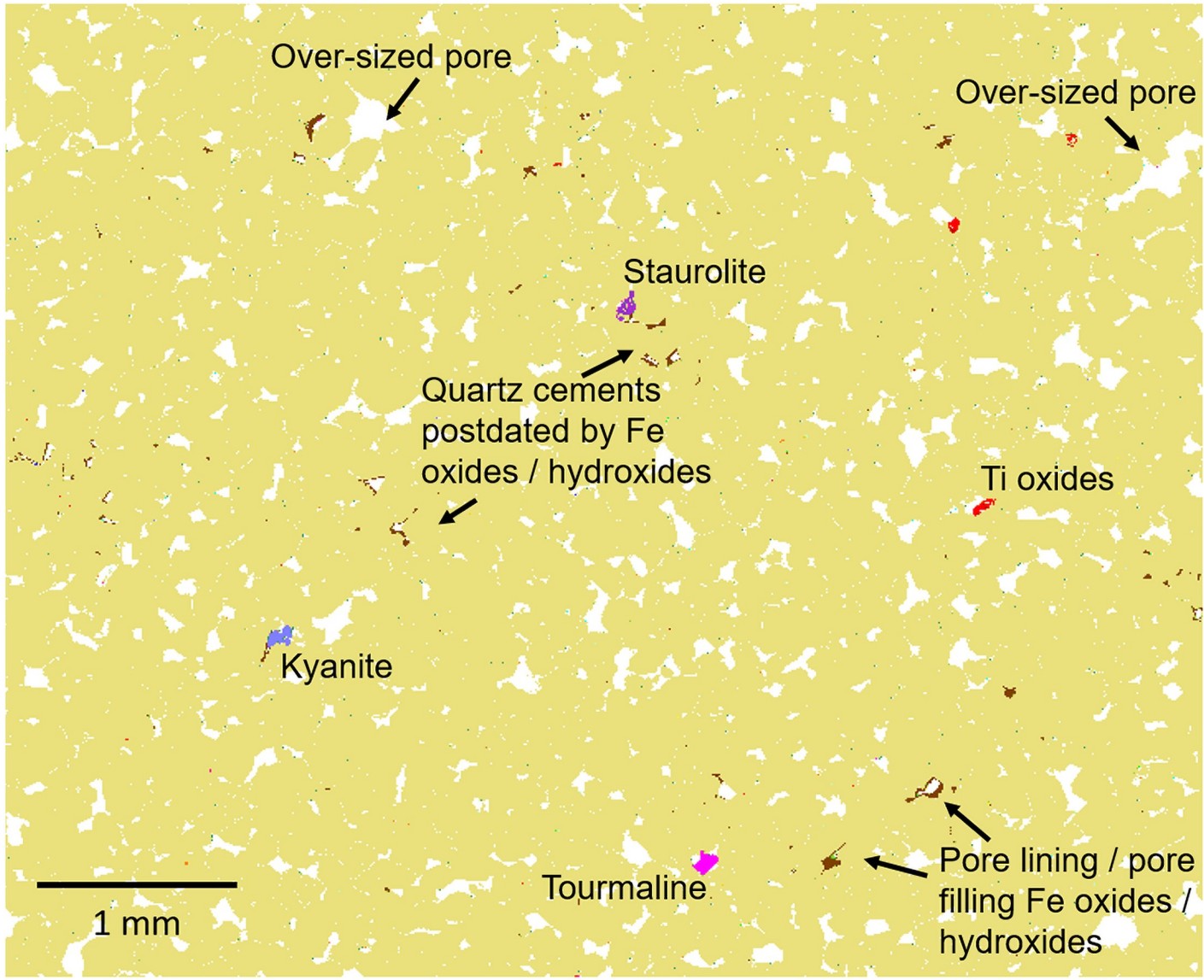

**Fig 12. Automated SEM-EDS (QEMSCAN) mineralogical map (for area of thin-section SH2B from the Phillips' Core detailed in Fig 11) highlighting textural features.**

Individual images were obtained with an accelerating voltage of 20 kV, a working distance of 8.0 mm, and a resolution of ~750 nm per pixel. The Atlas 5 software was then used to stitch the individual images from a thin section to produce a single mosaic image with a resolution of ~50,000×40,000 pixels.

**Table 3. Analysed area and number of spectra of hyperspectral SEM-EDS imaging datasets.**

| Thin-section | x (mm) | y (mm) | Area (mm²) | Spectra x | Spectra y | Total Spectra |
|---|---|---|---|---|---|---|
| SH1B | 25.9 | 13.9 | 359 | 8,032 | 4,300 | 34,537,600 |
| SH2B | 25.6 | 17.6 | 450 | 7,942 | 5,433 | 43,148,886 |
| SH3B | 25.1 | 20.0 | 503 | 7,796 | 6,210 | 48,413,160 |

**3.5.2. SEM-CL and SEM-EDS results.** SEM-CL analyses allow a clear distinction to be made between host sediment grains and macroscopic quartz overgrowth cements (Fig 13). SEM-CL results support the findings of optical petrographic (section 3.3) and automated mineralogical (section 3.4) analyses in demonstrating the dominance of quartz within the host sediment. Both red and blue luminescing quartz grains are present (Fig 13B and 13E), which confirms a mixed metamorphic and igneous origin for the detrital quartz [36]. We acknowledge, however, that CL colour shifts from blue to red with increasing radiation during analysis, and that this has almost certainly happened, to a limited extent, during the scanning of the polished sections. Given the difficulty in attributing quartz provenance on the basis of CL colour [36], and the effect of radiation, it is not possible to offer further detail about the proportions of volcanic, plutonic and metamorphic quartz that might be present.

Grain-cement contacts are imaged more clearly in CL than by optical (Fig 10) or back-scattered BSE microscopy (Fig 13A and 13D) and reveal that the majority of quartz grains are subrounded (Fig 13C and 13F). In some cases, grains show localised evidence of dissolution pitting/fretting. SEM-CL also offers unique insights into the internal structure of the quartz overgrowth cement within the sarsen. RGB images (Fig 13B and 13E) provide an overview of the total luminescence but the variability in cement luminescence is best observed at red wavelengths (Fig 13C and 13F).

Throughout the three polished sections, the initial cement comprises a thin (<10 μm) zone of non-luminescing quartz, which infills embayments and irregularities in quartz grain surfaces (e.g. see clast in bottom right on Fig 13F). This is overgrown by multiple concentric cement growth zones that become increasingly euhedral with distance from the quartz grain, culminating in crystal terminations in open pores (Fig 13B and 13E). Growth zones are typically on the scale of a few tens of microns in width and likely reflect changes in physico-chemical conditions during overgrowth precipitation and crystal growth [37–39]. Growth zones can be traced around individual host detrital quartz grains but may be crosscut and truncated by adjacent grain overgrowths. The most complete cement stratigraphy is observed adjacent to primary pore spaces, where up to 16 distinct growth generations of alternating luminescing and, typically thinner, non-luminescing quartz cement can be identified (e.g. see top left and centre of Fig 13C). Sector zonation is commonly superimposed on the concentric growth fabrics and ranges from relatively simple (reflecting differences in crystal orientation) to more complex (Fig 13C and 13F).

Analysis of the thin-sections using SEM-EDS (Fig 14) reveals additional insights into the development of late-stage void linings and fills within the sarsen. EDS net intensity composite elemental maps (Fig 14B–14D) indicate that most Fe-rich void linings occur as planar coatings on the faces of euhedral quartz overgrowths. These coatings can be seen to comprise crystallites in a botryoidal habit. Where larger (up to ~50 μm) botryoidal Fe-rich void fills occur, they postdate and sometimes truncate these planar coatings (e.g. top-centre Fig 14B). Ti-rich void linings occur as extremely thin (~1–2 μm) coatings overlying planar and/or botryoidal Fe-rich void linings. In places, Fe and Ti appear to occur as complex mixed fills. However, on inspection, these are highly altered, rounded detrital Fe-Ti-oxide grains with surrounding Fe-rich void fills. In the area of thin-section SH3B shown in Fig 14D, for example, the host sediment would have comprised quartz grains and an altered Fe-Ti-oxide grain. Euhedral quartz overgrowths developed on quartz grains while a more irregular quartz cement formed at the interface with the porous Fe-Ti grain. Fe-rich minerals then part lined and/or filled the remaining void spaces.

## 4. Geochemistry of the core

The geochemistry of the Phillips' Core from Stone 58 was assessed using three approaches. First, a handheld pXRF scanner was used to determine chemical variability along the full

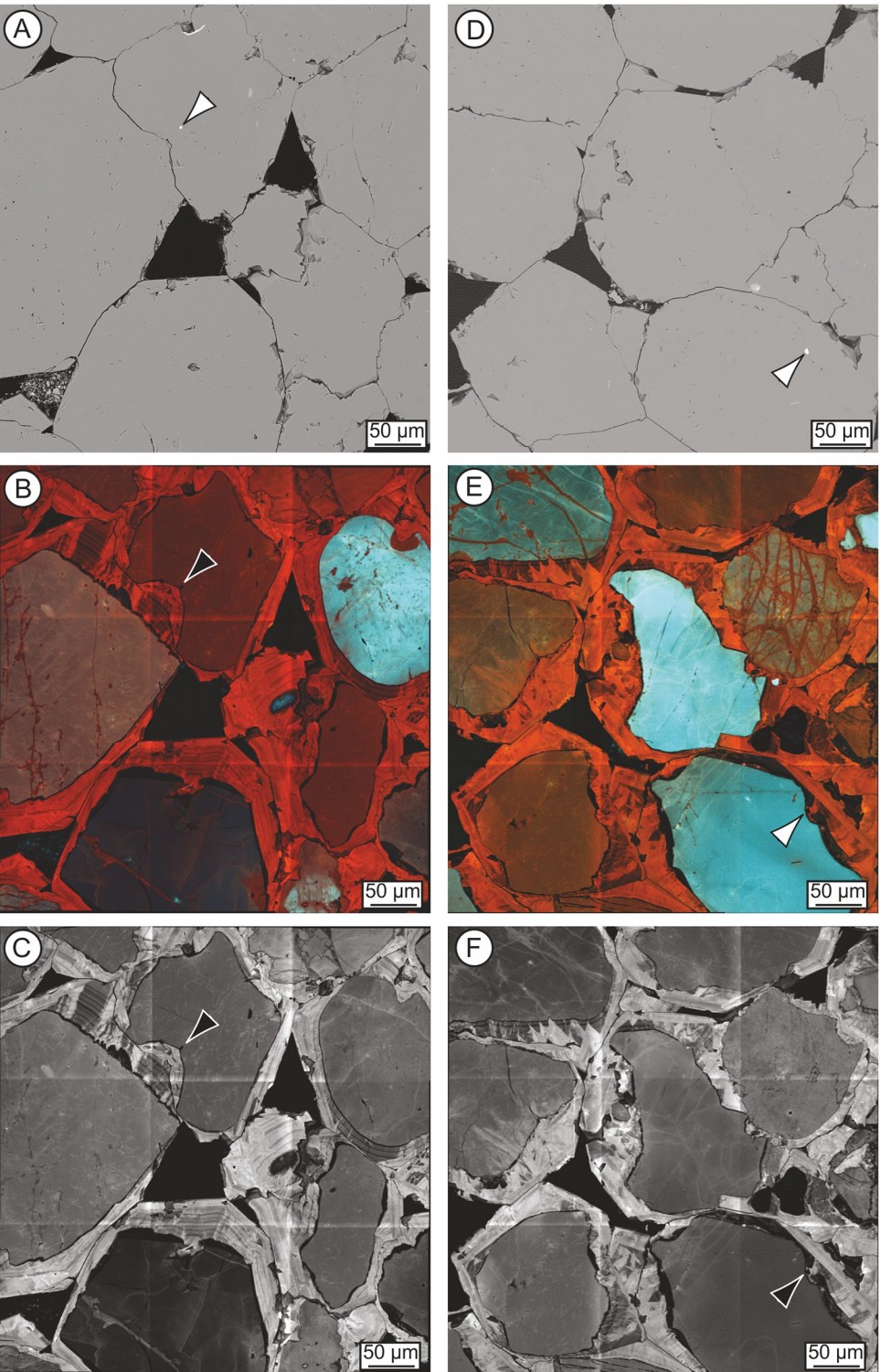

**Fig 13. Variability in the quartz-rich host sediment and quartz cement within polished thin-sections SH2B (left hand column) and SH1B (right hand column) from the Phillips' Core.** Back-scattered electron (BSE) images (A, D; 786 nm pixel size) and cathodoluminescence (SEM-CL) images of the same areas (B, E—Red-Green-Blue composite; C, F—red component; 320 nm pixel size). Arrows show ~2–6 μm zircon grains at the contact of a quartz grain and initial layer of non-luminescing quartz cement. These images are reproduced under a CC BY 4.0 license, with permission from The Trustees of the Natural History Museum, original copyright (2019).

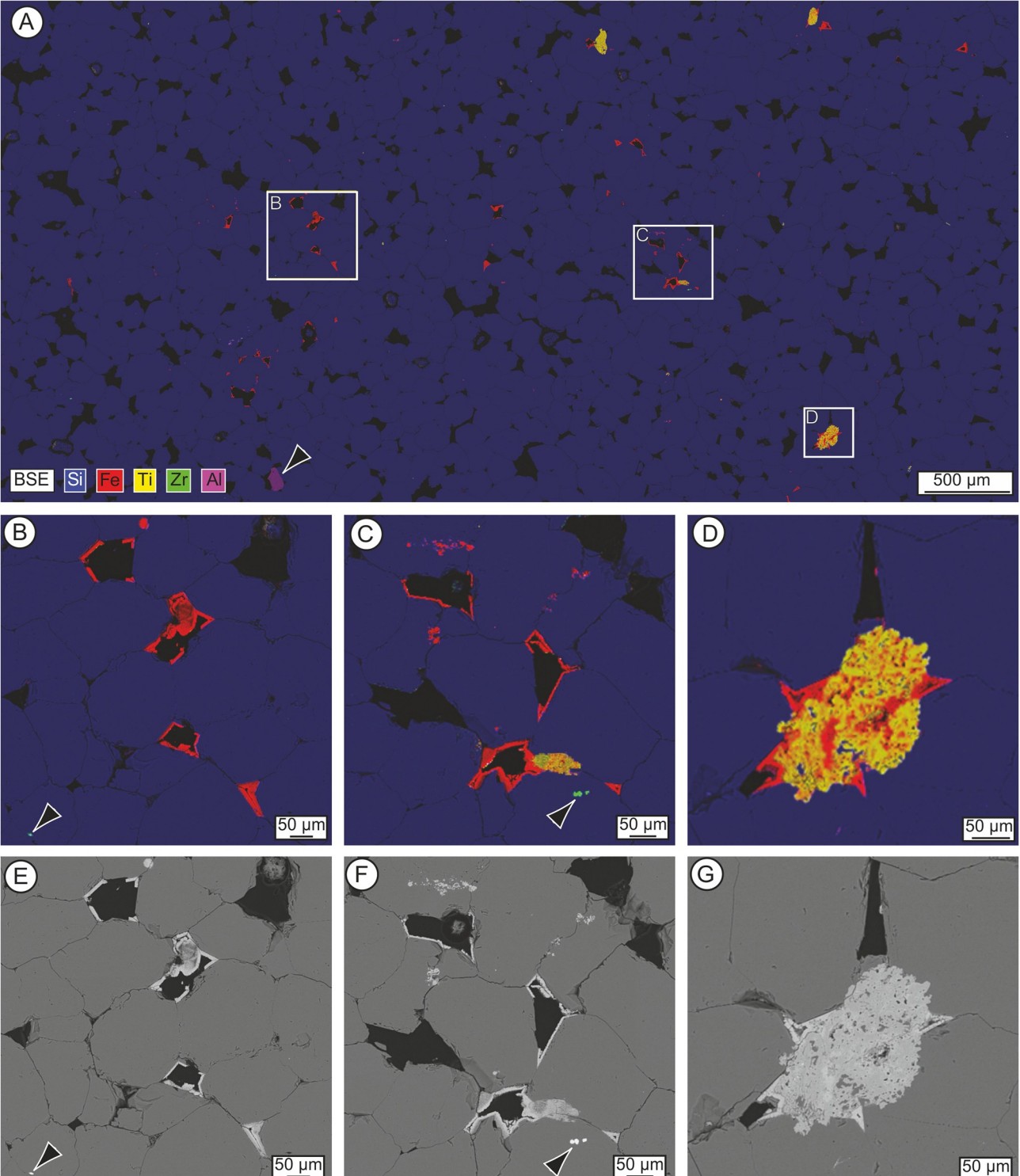

**Fig 14. Energy-dispersive spectrometry (EDS) net intensity composite elemental maps (A-D) and back-scattered electron (BSE) images (E-G) of polished thin-section SH3B from the Phillips' Core.** (A) Mosaic EDS map (3.2 μm pixel size) overlain with BSE micrograph (5100 × 2500 pixels, 1.6 μm pixel size). Quartz is represented in blue (Si), iron oxides/hydroxides in red (Fe), titanium oxides in yellow (Ti), zircon in green (Zr) and kyanite (arrow) in magenta (Al). (B-D) Image detail of the rectangles shown in (A). Arrows indicate ~5–10 μm zircon grains. (E-G) BSE images (749 nm pixel size) of the areas shown in (B-D). These images are reproduced under a CC BY 4.0 license, with permission from The Trustees of the Natural History Museum, original copyright (2019).

length of the core. Then the length of the semi-cylinder of sarsen from section 2–3 was analysed using an XRF core scanner; and set B of the polished thin-sections was analysed using a μXRF scanner. ICP-MS and ICP-AES were applied to subsamples from section 2–3 of the core (Fig 6D) to determine the whole-rock geochemistry. Whole-rock isotopic analysis was also applied to the remaining subsamples from section 2–3 (Fig 6D).

## 4.1. Whole-rock major and trace element analysis (by pXRF)

**4.1.1. Methodology.** Portable XRF analysis was used to provide a rapid geochemical assessment of the full Phillips' Core. Analyses along the length of the core were undertaken using a handheld Olympus Innov-X Delta Professional XRF spectrometer at the English Heritage Collections Store, Temple Cloud (see Fig 5 for positions of analyses). The model operates at 40 kV, is equipped with a Rh anode 4W X-Ray tube and uses a silicon drift detector. The 'Geochem' mode was used for all pXRF analyses; this captures Mg, Al, Si, P, S, K, Ca, Ti, V, Cr, Mn, Fe, Co, Ni, Cu, Zn, As, Se, Rb, Sr, Y, Zr, Nb, Mo, Ag, Cd, Sn, Sb, W, Hg, Pb, Bi, U and Th. Analyses were undertaken at approximately 5 cm intervals (Fig 5). At each point, the stone was analysed for 120 s of total exposure. The device was positioned such that the detector window was completely covered by the stone and the centre of the detection window sat on the apex of the core. The Innov-X Delta instrument has a detector window of ~20 mm in diameter, while the X-ray source excites a target circle with a 3 mm diameter. At the start and end of the analyses, a calibration check was made against a 316 Stainless Steel Calibration Check Reference Coin to ensure accuracy and consistency of the results. Data were processed in Excel.

**4.1.2. Portable XRF results.** Selected pXRF geochemical data for the Phillips' Core are shown in Table 4 and Fig 15 (the full dataset is available in the S1 Data and is archived at the Archaeology Data Service). Note that these are uncalibrated raw count % data and, while internally consistent as a dataset, should not be compared directly with the whole rock major and trace element analyses presented in sections 4.2 and 4.4; we include the data here for the benefit of future researchers seeking to explore sarsen chemistry via pXRF. The pXRF data confirm that the sarsen comprises almost pure silica, with only minor proportions of other major elements identified (Al, Ca, Fe, K, Mg, Mn, P and Ti). This remarkably high purity reflects the quartz-rich mineralogy of the rocks. The chemistry remains remarkably consistent along the length of the core, with only Fe showing magnitude variations in concentration (Fig 15A).

Throughout the core, the pXRF data show no correlation between Fe and Ti (Fig 15B). This suggests that Fe variation within the core is not entirely controlled by the variable abundances of Fe-Ti bearing oxides (e.g. rutile, ilmenite or magnetite). Sulfur was below detection limit throughout the core, and there is no correlation between Fe and the abundance of either Cu (Fig 15C) or Zn (Fig 15D). This indicates that the Fe variation is unlikely to be related to the variable abundances of metal sulfide grains (or their weathering products). Instead, cross-referencing these results with the petrographic observations in section 3.3.2 suggests that the variability in Fe is likely driven mainly by changes in the abundance of Fe-rich oxides and hydroxides such as limonite. This inference is supported by the association between peaks in Fe concentration and the occurrence of pale brown-coloured (Fe-oxide/hydroxide) zones along the core (Fig 15A). This suggests that variations in Fe abundance are controlled by variable precipitation of Fe minerals from Fe-bearing fluids percolating through the sarsen while it was still buried in the subsurface.

## 4.2. Whole-rock major and trace element analysis (by XRF core scanner)

**4.2.1. Methodology.** Higher resolution XRF analysis of section 2–3 of the Phillips' Core from Stone 58 at Stonehenge was conducted using a Cox Analytical Systems ITRAX-MC core

**Table 4. Uncalibrated pXRF data showing element concentration (count %) measured at distances along the Phillips' Core where elements were detected (blank cells = not detected).** See Fig 5 for locations of analyses.

| Distance from end of core (cm) | Si | Ti | V | Mn | Fe | Ni | Cu | Zn | Sr | Y | Zr | Nb | Mo | Pb | Bi | U | Total |
|---|---|---|---|---|---|---|---|---|---|---|---|---|---|---|---|---|---|
| 1 | 59.14 | 0.051 | 0.0150 | | 0.0962 | | | 0.0008 | 0.0003 | 0.0002 | 0.0038 | 0.0004 | | | | | 59.31 |
| 6 | 61.38 | 0.062 | 0.0137 | | 0.0422 | | | | | | 0.0035 | | | | 0.0008 | | 61.50 |
| 11 | 58.55 | 0.080 | | | 0.0441 | | 0.0012 | | | | 0.0034 | 0.0005 | | | | | 58.68 |
| 15 | 60.14 | 0.049 | 0.0120 | | 0.1138 | | 0.0008 | | | | 0.0046 | | | | | | 60.32 |
| 20 | 55.93 | 0.056 | | | 0.1215 | | | | | | 0.0035 | | | | | | 56.11 |
| 25 | 58.36 | 0.042 | 0.0133 | | 0.1341 | | | | | | 0.0031 | 0.0004 | | | | | 58.55 |
| 30 | 62.64 | 0.067 | 0.0171 | | 0.1065 | 0.0018 | 0.0010 | | | 0.0003 | 0.0023 | | 0.0005 | | | | 62.84 |
| 35 | 52.70 | 0.108 | | | 0.1412 | | 0.0013 | 0.0006 | 0.0002 | | 0.0033 | 0.0008 | 0.0011 | | | | 52.96 |
| 39 | 56.65 | 0.080 | 0.0213 | 0.0041 | 0.0531 | | | | | | 0.0031 | 0.0004 | | | | | 56.81 |
| 44 | 57.14 | 0.037 | 0.0127 | | 0.0884 | | | | | | 0.0029 | | | | | | 57.28 |
| 49 | 58.37 | 0.044 | 0.0168 | | 0.0655 | | | | 0.0002 | | 0.0032 | | | | | | 58.50 |
| 54 | 62.98 | 0.046 | | | 0.0615 | | | | 0.0002 | | 0.0028 | | | | | | 63.09 |
| 59 | 61.97 | 0.046 | | | 0.3807 | | | | | | 0.0030 | | | | | | 62.40 |
| 63 | 61.48 | 0.046 | 0.0129 | | 0.0435 | | | | | | 0.0032 | | | | | | 61.59 |
| 68 | 61.47 | 0.053 | | | 0.0960 | | | | | | 0.0038 | | | | | | 61.62 |
| 73 | 60.66 | 0.038 | 0.0120 | | 0.1414 | | 0.0010 | 0.0010 | | | 0.0034 | | | | | | 60.86 |
| 78 | 60.11 | 0.049 | | | 0.1308 | | | 0.0006 | | 0.0002 | 0.0027 | | | | | | 60.29 |
| 83 | 59.34 | 0.057 | | | 0.1518 | | | | | | 0.0033 | | 0.0004 | | | | 59.55 |
| 87 | 54.30 | 0.041 | 0.0136 | | 0.1043 | | | | | | 0.0039 | | | | | | 54.46 |
| 92 | 56.37 | 0.123 | | | 0.0413 | | | | 0.0002 | | 0.0029 | | | | | | 56.54 |
| 97 | 59.56 | 0.066 | 0.0121 | | 0.0528 | | | 0.0008 | 0.0002 | | 0.0039 | | | | | | 59.70 |
| 102 | 59.67 | 0.030 | | | 0.1259 | | 0.0011 | 0.0009 | 0.0002 | | 0.0030 | | | 0.0006 | | | 59.83 |
| 107 | 58.99 | 0.072 | | 0.0046 | 0.0497 | | 0.0017 | | 0.0004 | 0.0003 | 0.0045 | 0.0006 | 0.0008 | | | 0.0005 | 59.13 |

scanner at the British Geological Survey, Keyworth, UK. XRF profile scanning was performed along the length of the sample using an Rh-tube set to 30 kV and 1.2 mA, and a Si-drift chamber detector. The ITRAX-MC has a detector window of 8 mm (cross-core) by 1 mm (down-core). Measurements were taken using continuous scanning at 1 mm resolution and an exposure time of 30 s/mm. Replicate scans of the whole sample were performed to ensure the consistency of results. XRF spectral data were processed using the Cox Analytical Systems Q-Spec spectral analysis software version 15.7 to fit and extract the net peak intensities (counts) for specific elements. The same software can be used to quantify elemental concentrations using a fundamental parameter-based calibration and NIST610 as a reference sample.

**4.2.2. XRF core scanning results.** The process of spectral fitting was complicated by the crystalline nature of the sample material, giving rise to spectral artefacts that have been interpreted as diffraction peaks, 29 of which were used in the spectrum analysis. In addition, anomalous attenuations in the silica-rich cement impact elemental responses; for example, a small peak for Mg will be interpreted as very high concentrations due to the high inter-element effects in such a cement. To the low energy side of a dominant peak, in this case Si, the low energy tailing can significantly increase the background scatter. Even with the implementation of the diffraction peaks, the background at the tail of the spectrum (>8 keV) is poorly fitted, hence elements in this region (e.g. Zr) cannot be properly quantified. The raw XRF data are available in the S2 Data and are archived at the Archaeology Data Service. However, quantified data for elements other than Si should be treated with caution and are therefore not reported here.

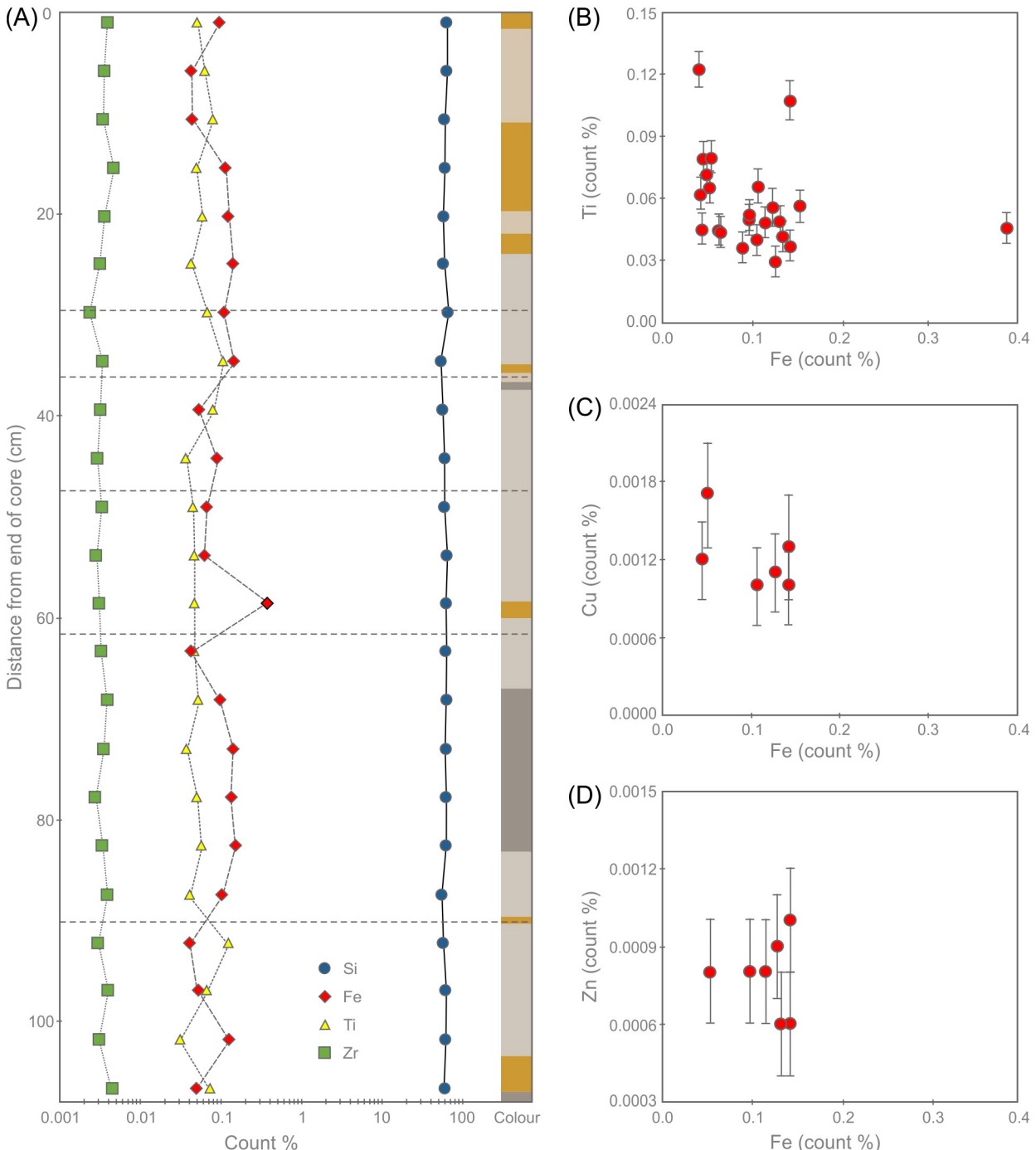

**Fig 15. Portable XRF geochemical data showing (A) the variation in count % of selected elements and indicative Munsell colour along the length of the Phillips' Core from Stone 58.** Fractures are indicated as dashed lines to allow cross-referencing with Fig 5. Panels (B) to (D) show the correlation between Fe count % and (B) Ti, (C) Cu and (D) Zn. Error bars in B-D indicate instrumental error. Note that the error for Fe is smaller than the symbol diameter so is not displayed.

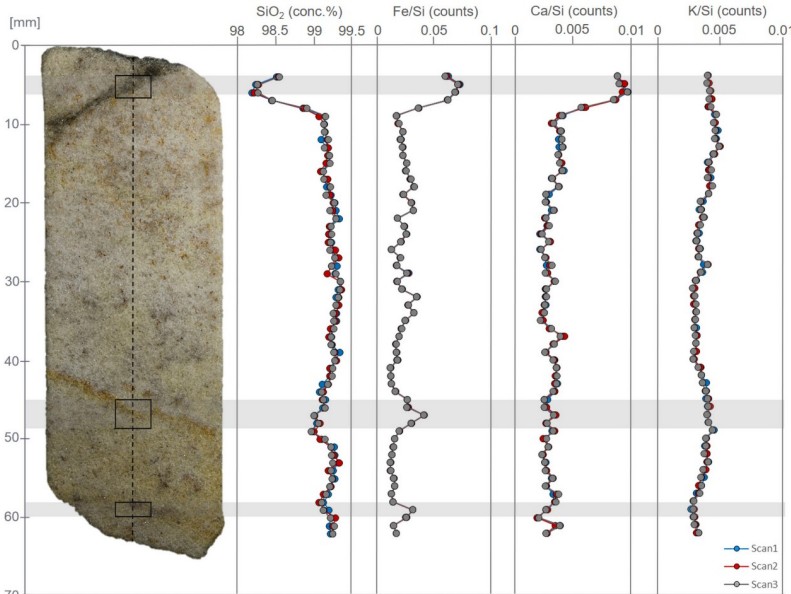

**Fig 16. Optical image and selected XRF data for section 2–3 of the Phillips' Core from Stone 58 at Stonehenge.**
Dashed line indicates path of XRF detector along the central axis length of the sample. Solid boxes indicate areas of
interest (see section 4.2.2.). Replicate XRF scans (3×) along the same axis line were performed to ensure consistency of
the results. The image and data are reproduced under a CC BY 4.0 license, with permission from the British Geological
Survey, original copyright (2019).

The sample is $SiO_2$-rich ($SiO_2$ ~99.2 wt. %), supporting the petrographic (section 3.3),
pXRF (section 4.1) and ICP-AES (section 4.4) results. The concentration of $SiO_2$ is relatively
constant along the length of the sample (Fig 16). Other element peak counts have been nor-
malised to the silica-peak to identify potential trends. Element counts of Fe and Ca show a
slight increase (Fig 16) in the first 10 mm of the scan, possibly caused by the artefact corre-
sponding to the diamond saw blade mark (Fig 7). Additionally, the elemental counts for Fe
show further variation, with peaks in counts centred upon ~47 and ~59 mm correlating with
thin bands of iron hydroxide staining. Apart from this, there is no significant chemical vari-
ability along the full length of section 2–3 of the Phillips' Core. Other elements were present at
low levels such that it has not been possible to observe patterns in their distribution; this is
compounded by uncertainty in elements such as Al, P and S due to proximity to the dominant
Si peak, and Ti, V, Ni, Cr and Ba due to their interference with spectral artefacts and poor spec-
tral fitting.

## 4.3. Whole-rock major and trace element analysis (by μXRF)

**4.3.1. Methodology.** μXRF analysis was used to produce high resolution maps of the dis-
tribution and abundance of selected elements within section 2–3 of the Phillips' Core. Polished
thin-sections SH1B, SH2B and SH3B were analysed using a Bruker Tornado M4 micro X-ray
fluorescence (μXRF) scanner at the Vrije Universiteit Brussel. μXRF mapping was carried out
under near-vacuum conditions (20 mbar), along a 2D grid with 25 μm spacing, a spot size of
25 μm and an integration time of 1 ms per pixel. The X-ray source was operated under maxi-
mum energy settings (600μA, 50kV) and no source filters were applied. Total scan time was
approximately 3.5 h to cover 1983×3143 pixels. This mapping approach by μXRF permits ele-
ment abundance distributions to be visualised.

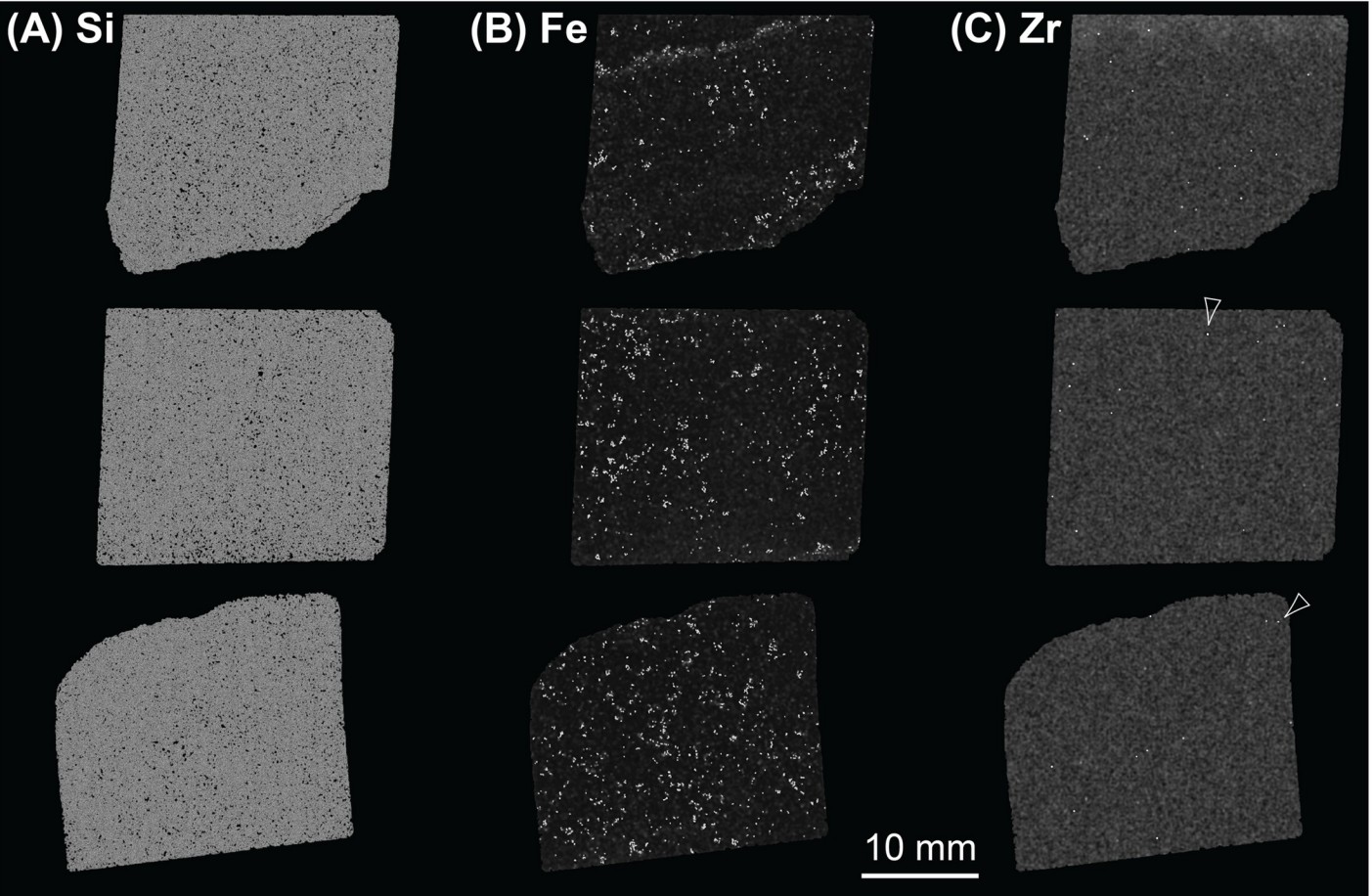

**Fig 17. Composite showing μXRF elemental maps of thin sections SH1B (bottom row), SH2B (middle row) and SH3B (top row) from the Phillips' Core (from Stone 58 at Stonehenge) for (A) Si, (B) Fe and (C) Zr.** Note that Zr is present at much lower concentrations than Si and Fe so some background noise is present in panel C—example spots of higher Zr intensity are arrowed.

**4.3.2. μXRF results.** The results of μXRF analyses complement the data generated via QEMSCAN (section 3.4), SEM-EDS (section 3.5) and the other whole-rock XRF approaches (sections 4.1 and 4.2). Due in part to the dominance of silica (Fig 17A) and in part to thin-section thickness, few elements provide clear signals. However, some patterns can be seen on elemental abundance maps for those elements present in relatively higher concentrations. Abundance maps for all elements providing clear signals and heatmaps for selected elements are archived at the Archaeology Data Service.

Consistent with the results of SEM-EDS mineralogy (Table 2), Fe is the next most abundant element after Si. Fe is distributed irregularly throughout sections SH1B and SH2B (lower two images on Fig 17B), reflecting the distribution of Fe-rich detrital mineral grains and late-stage void linings and fills (see section 3.2.2). Heterogeneity in Fe distribution can, however, be recognised in section SH3B (uppermost image on Fig 17B), where two bands of higher Fe concentration are visible running through the top and bottom of this thin-section. These bands correspond to the zones of pale brown staining seen parallel to the natural fracture plane in section 2–3 of the Phillips' Core (right hand end of Fig 7) and comprise late-stage void-lining/-filling limonite (section 3.2.2). Differences in the Fe concentration in each band can be seen from the Fe heatmap in Fig 18. The lower band on the heatmap, closest to the natural fracture

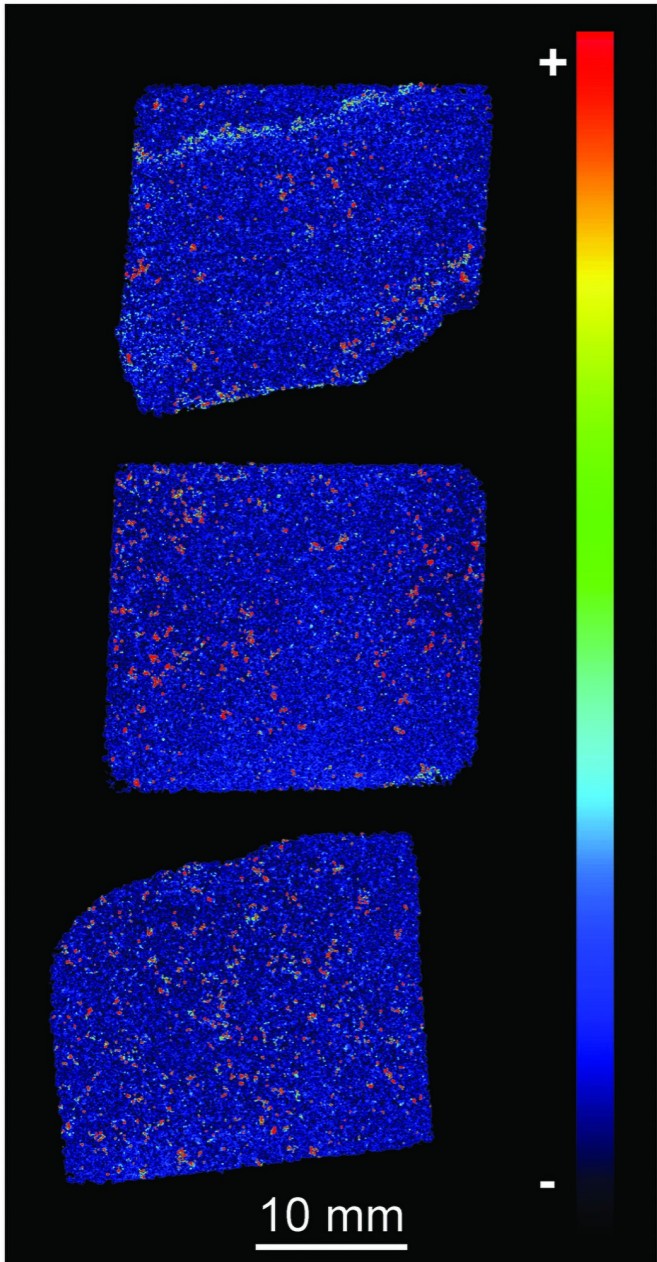

**Fig 18. µXRF heatmap of the relative intensity of Fe in thin sections SH1B (bottom), SH2B (middle) and SH3B (top) from the Phillips' Core (from Stone 58 at Stonehenge).** Red colours indicate higher and blue colours lower relative Fe concentrations.

surface, exhibits bright orange-red colours, indicating that Fe is present in higher relative concentrations than the upper band (turquoise-green colours) ~20 mm away from the fracture surface. The distribution of other elements tends to be irregular, as illustrated by the elemental abundance map for Zr (Fig 17C).

## 4.4. Whole-rock major and trace element analysis (by ICP-MS and ICP-AES)

**4.4.1. Methodology.** ICP-MS and ICP-AES analyses were conducted to generate high-resolution whole-rock geochemical data from the Phillips' Core. The three larger subsamples shown in Fig 6D were processed and analysed by ALS Minerals (Seville, Spain). These subsamples are numbered SHCORE-ICP01 to SHCORE-ICP03, with the numbers 01, 02 and 03 corresponding to the respective thin-section number. In Spain, each subsample was crushed using a hardened steel jaw crusher such that >70% of the resulting fragments passed through a 2 mm screen size (ALS Geochemistry preparation package CRU-31). The crushed samples were powdered in an agate ball mill such that >85% passed a 75 μm screen size (ALS Geochemistry package PUL-42). Major/minor oxides were analysed by lithium metaborate fusion digestion and ICP-AES (ALS Geochemistry method ME-ICP06). Trace elements, including rare earth elements, were determined using lithium metaborate fusion digestion and ICP-MS (ALS Geochemistry method ME-MS81). As, Bi, Hg, In, Re, Sb, Se and Te were determined by aqua regia digestion followed by ICP-MS (ALS Geochemistry method ME-MS42). Ag, Cd, Co, Cu, Li, Mo, Ni, Pb, Sc, and Zn were determined by four-acid digestion and ICP-AES (ALS Geochemistry method ME-4ACD81). In all cases, ICP-MS analyses were conducted using an Elan 9000 instrument and ICP-AES analyses using a Varian 700 Series instrument. Total C and S were analysed by Leco induction furnace and Leco sulfur analyser (ALS Geochemistry methods C-IR07 and S-IR08 respectively). Loss on Ignition (LOI) was calculated following ignition of sample powders at 1000°C (ALS Geochemistry method OA-GRA05).

**4.4.2. ICP-MS and ICP-AES results.** *Patterns within the whole-rock geochemical data.* Whole-rock geochemical data for the three subsamples are shown in Table 5. Full data are available in the S3 Data, including analyses of certified reference materials (S4 Data), and are archived at the Archaeology Data Service. Nash et al. [5] have described selected trace element patterns in this dataset (Ba, Ce, Dy, Er, Gd, Hf, Ho, La, Nb, Nd, Pr, Rb, Sm, Sr, Tb, Th, Ti, Tm, U, Y, Yb, Zr); here we consider the full geochemical dataset.

Supporting the petrographic data from section 3.3, QEMSCAN data from section 3.4 and XRF data from sections 4.1–4.3, the three subsamples are highly silica-rich ($SiO_2 \geq 99.7$ wt. %), with very little variation in major element chemistry (0.05–0.06 wt. % $Al_2O_3$, 0.01 wt. % CaO, 0.09–0.12 wt. % $Fe_2O_3$ and 0.06 wt. % $TiO_2$). The remaining major element oxides ($Na_2O$, MgO, $K_2O$, MnO and $P_2O_5$) are at or below detection limit (0.01 wt. %). The trace element data in Table 5 also show consistency in elemental concentrations across the three subsamples.

The average total rare earth element (REE) contents of the three subsamples range from 3.01–3.48 ppm. These extremely low concentrations are likely to be partially due to dilution by $SiO_2$ during silicification [40, 41]. On chondrite-normalised REE plots (Fig 19), the three subsamples plot as a set of parallel concave signatures characterised by enrichment in both the light (LREE: La, Ce, Pr and Nd) and heavy REE (HREE: Ho, Er, Tm, Yb and Lu) relative to the middle REE (MREE: Sm, Eu, Gd, Tb and Dy). Subsample SHCORE-ICP01 is defined by a partly 'sawtooth' pattern in the HREE. This subsample also contains highest $SiO_2$ content and the lowest REE abundances of the three samples. It is likely, therefore, that the sawtooth pattern is an instrumentation artefact caused by the extremely low abundances. This mechanism is also likely to have affected the reported Lu concentration, which was at detection limit in two of the three subsamples and below detection limit in the third.

Noting these issues, comparison of the average chondrite-normalised signature from the Phillips' Core with that of the Upper Continental Crust (UCC) shows that, though the three subsamples contain orders of magnitude lower absolute abundances of REE, both have similar

**Table 5. Whole-rock major and trace element geochemical data (by ICP-AES and ICP-MS) for three subsamples from section 2–3 of the Phillips' Core (Stone 58, Stonehenge).** Detection limit is indicated by inequality sign.

| Major elements (wt. %) | SHCORE-ICP01 | SHCORE-ICP02 | SHCORE-ICP03 |
|---|---|---|---|
| $SiO_2$ | 100 | 100 | 99.7 |
| $Al_2O_3$ | 0.05 | 0.06 | 0.06 |
| $Fe_2O_3$ | 0.09 | 0.12 | 0.09 |
| CaO | 0.01 | 0.01 | 0.01 |
| MgO | <0.01 | <0.01 | <0.01 |
| $Na_2O$ | 0.01 | <0.01 | 0.01 |
| $K_2O$ | <0.01 | <0.01 | <0.01 |
| $TiO_2$ | 0.06 | 0.06 | 0.05 |
| MnO | <0.01 | <0.01 | <0.01 |
| $P_2O_5$ | <0.01 | <0.01 | <0.01 |
| C | 0.07 | 0.1 | 0.08 |
| S | <0.01 | <0.01 | <0.01 |
| LOI | 0.53 | 0.39 | 0.34 |
| Total | 101.75 | 100.64 | 100.26 |
| **Trace elements (ppm)** | | | |
| V | <5.00 | 5.00 | 5.00 |
| Cr | 10.00 | 10.00 | 10.00 |
| Co | <1.00 | <1.00 | <1.00 |
| Ni | 2.00 | 2.00 | 1.00 |
| Ga | 0.80 | 0.80 | 0.70 |
| Rb | 0.20 | 0.20 | 0.50 |
| Sr | 1.20 | 1.00 | 1.60 |
| Y | 0.90 | 1.60 | 0.90 |
| Zr | 40.00 | 36.00 | 37.00 |
| Nb | 1.00 | 1.10 | 0.90 |
| Sn | 1.00 | 1.00 | 2.00 |
| Cs | 0.02 | 0.01 | 0.03 |
| Ba | 12.80 | 11.90 | 11.60 |
| La | 0.70 | 0.70 | 0.70 |
| Ce | 1.10 | 1.30 | 1.20 |
| Pr | 0.12 | 0.17 | 0.14 |
| Nd | 0.40 | 0.50 | 0.50 |
| Sm | 0.10 | 0.07 | 0.11 |
| Eu | <0.03 | 0.03 | <0.03 |
| Gd | 0.08 | 0.08 | 0.06 |
| Dy | 0.14 | 0.18 | 0.13 |
| Ho | 0.03 | 0.05 | 0.03 |
| Er | 0.14 | 0.18 | 0.10 |
| Tm | 0.02 | 0.04 | 0.03 |
| Yb | 0.13 | 0.14 | 0.10 |
| Lu | 0.01 | 0.01 | <0.01 |
| Hf | 1.10 | 1.00 | 0.90 |
| Ta | 0.10 | 0.10 | 0.10 |
| W | 4.00 | 3.00 | 12.00 |
| Pb | <2.00 | <2.00 | 2.00 |
| Th | 0.24 | 0.23 | 0.22 |
| U | 0.16 | 0.17 | 0.18 |

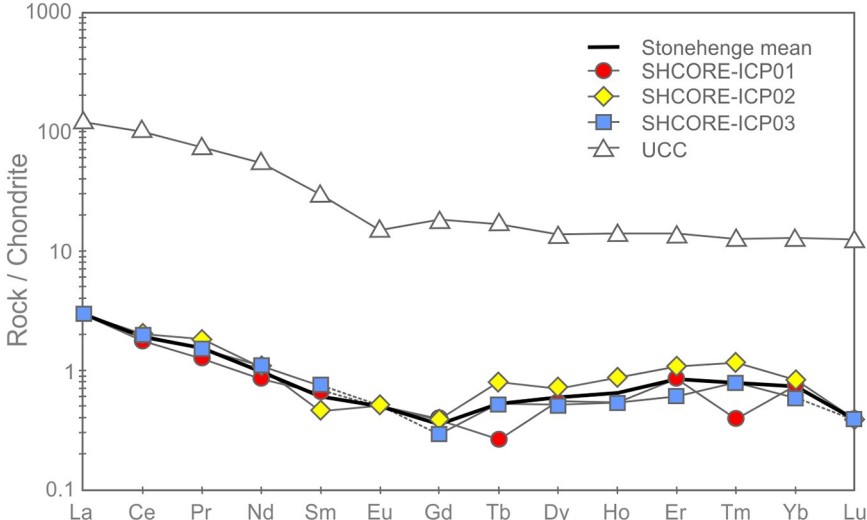

**Fig 19. Chondrite-normalised REE diagram for the three subsamples from section 2–3 of the Phillips' Core from Stonehenge (SH) showing the Upper Continental Crust (UCC) for comparison [42].** Concentrations below detection limit are plotted at detection limit and are signified by dashed lines [normalisation factors from 43].

relative enrichments in the LREE relative to the MREE. In contrast, the subsamples record inclined MREE—HREE patterns $(Gd/Yb)_N \leq 0.5$, which are distinctly different to those of the UCC $(Gd/Yb)_N = 1.4$. This may indicate that the original sands that were cemented to form the Stonehenge sarsens were derived from the sorting and deposition of mineral grains present in proportions significantly different to 'standard' crustal materials.

Fig 20 shows the complete trace element chemistry of the three subsamples normalised to the UCC. Also shown are trace element signatures for the North American Shale Composite (NASC) and average values for sandstones deposited during the Archean, Proterozoic and Phanerozoic aeons [44, 45].

The uniformly <1 normalised abundances of trace elements within the three subsamples further attest to the diluting effects of silicification on the trace element geochemistry of the original sediment [e.g. 40, 41]. Notable anomalies (i.e. significant peaks or troughs in normalised element abundance) in the signature are a function of the relative abundance of particular mineral phases, within which the specific trace elements are compatible [46]. Comparison of the signatures in Fig 20 with the QEMSCAN mineralogy (Table 2) and petrographic descriptions (section 3.3.2) allows some causative determinations to be made. The notable positive anomaly in Ti in the Stonehenge signatures is likely due to the presence of former Fe-Ti oxides in the subsamples, while the negative Sr anomaly is likely linked to the relative dearth of plagioclase (only two potential grains observed in thin-section SH1A; section 3.3.2) as compared to the UCC. The minor positive anomalies in Nb and Ta are likely due to an overabundance of $TiO_2$ phases (present in all three subsamples) relative to the UCC. The negative anomalies in Rb and Ba are most easily explained by a relatively low detrital mica content in the subsamples relative to the UCC. Minor positive anomalies in Zr and Hf are likely due to detrital zircon present in abundances relatively higher than that found in the UCC. These factors together produce a Stonehenge average trace element geochemical signature that, of the sediment compositions displayed in Fig 20, most closely resembles that recorded by Archean sandstones [44].

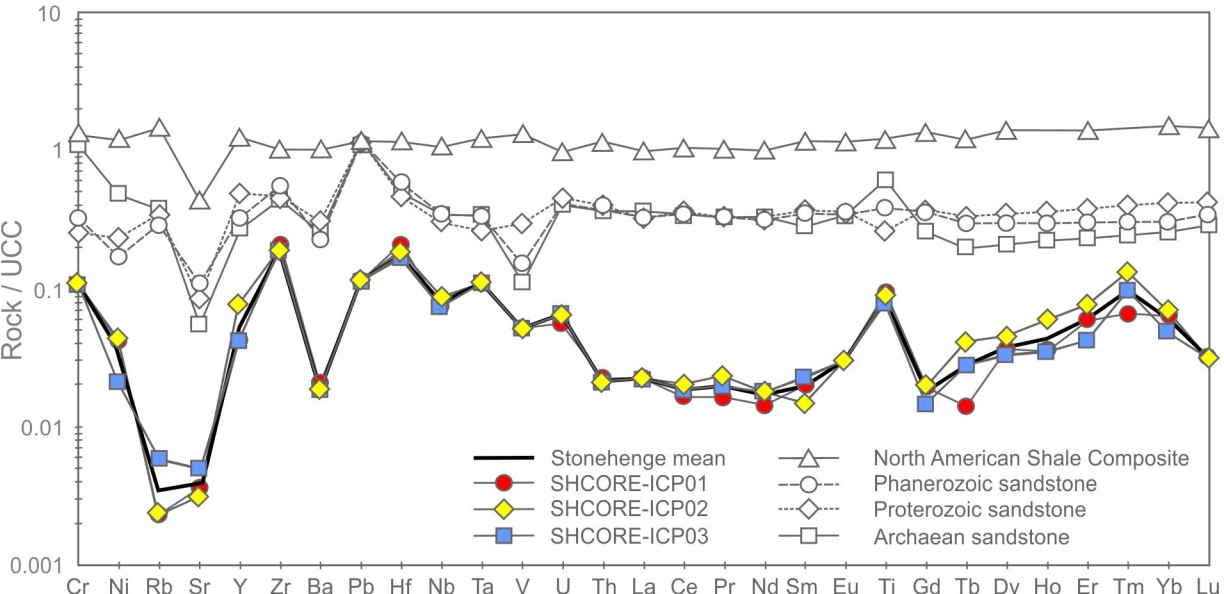

**Fig 20. UCC-normalised trace element diagram for the three subsamples from section 2–3 of the Phillips' Core from Stonehenge showing North American Shale Composite [45] and average compositions of Archean, Proterozoic and Phanerozoic sandstones (SST) [44] for comparison [normalising factors from 42].**

One perplexing feature of the UCC-normalised trace element geochemical signature for the three subsamples is the positively inclined HREE trend. Positively inclined HREE trends in sedimentary rocks are often explained by a relative abundance of detrital garnet in the sample [47, 48]; this is due to HREE becoming more compatible in garnet with increasing atomic number [49]. However, garnet was not detected optically or by QEMSCAN analysis in the subsamples (Table 2). This seeming contradiction between the geochemistry and mineralogy of the subsamples could potentially be the result of alteration of any detrital garnet originally in the sediment (to mica ± chlorite ± illite), during diagenetic or other alteration processes. Such a mechanism could have altered the garnet, thereby changing the mineralogy of the rock, but simultaneous retaining the immobile HREE within the alteration mineral assemblage. Alternatively, the HREE trend might be partly explained by the presence of relative overabundances of zircon and/or tourmaline grains, within which the HREE are increasingly compatible with increasing atomic mass [50, 51]. However, several of the HREE (e.g. Lu, Tm) are detected in only some of the subsamples at, or near detection limit (Table 5). Given that the accuracy of ICP-MS analysis for HREE becomes less good at such low concentrations [e.g. 52], there is potential for some of the interesting HREE trend observed in the Stonehenge core to be, in part, an instrumentation artefact.

*Classification.* Chemical classification of the subsamples from the Phillips' Core using standard geochemical schemes [e.g. 53, 54] is somewhat problematic. During the process of silcrete formation, silicification of the original host sediment necessarily increases the $SiO_2$ content of the resulting rock relative to the original clastic detritus, while other major elements (e.g. Al and Ti) may be remobilised [55]. Thus, classification of silcretes using schemes that rely on major element abundances likely reflect the effects of silicification rather than exclusively the composition of the original sediment.

The three subsamples from the Phillips' Core plot as a tight cluster in the quartz arenite field on major element classification diagrams (Fig 21). These major element diagrams are

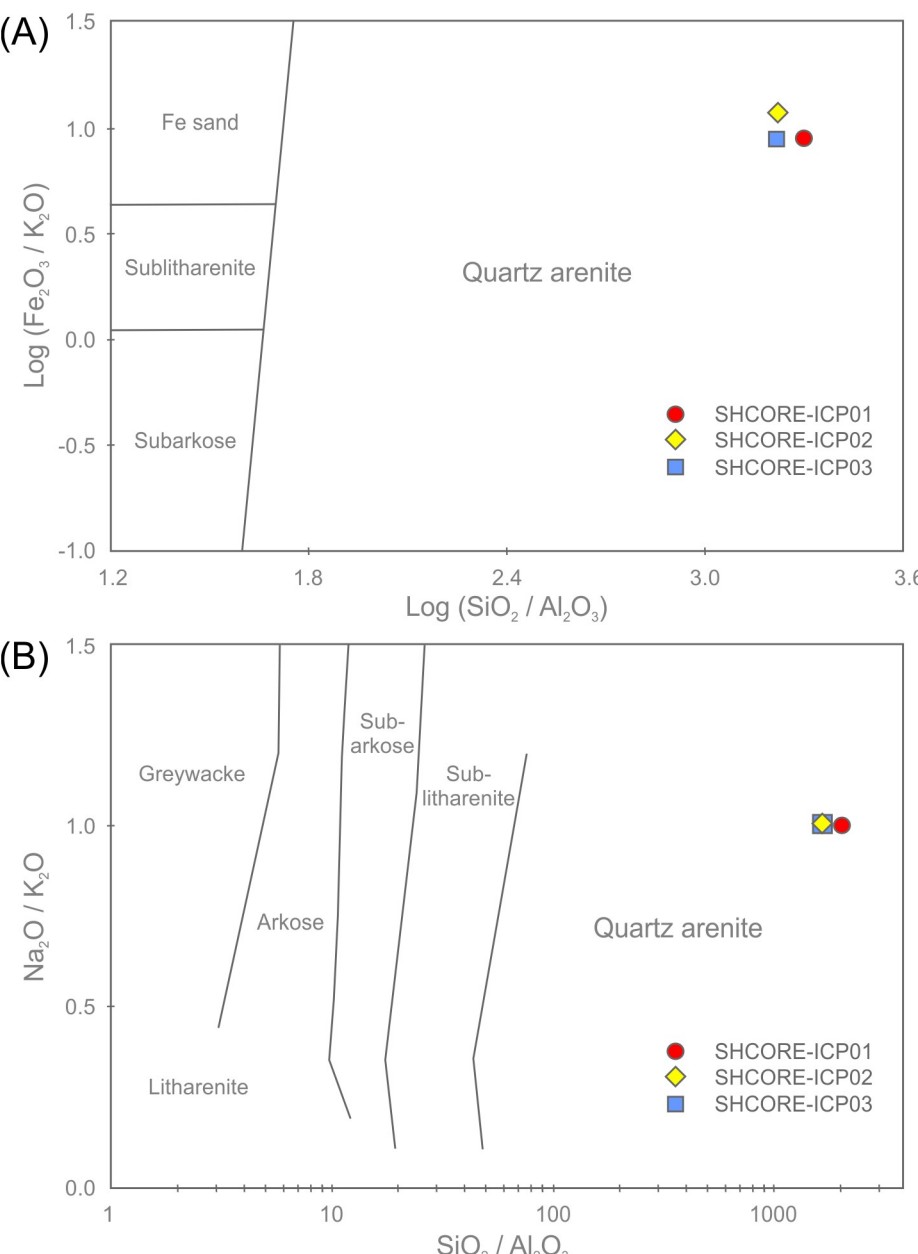

**Fig 21.** Major element classification schemes for the three subsamples from section 2–3 of the Phillips' Core from Stone 58 at Stonehenge: (A) Log ($SiO_2$ / $Al_2O_3$) vs. Log ($Fe_2O_3$ / $K_2O$), modified after [53]; (B) $SiO_2$ / $Al_2O_3$ vs. $Na_2O$ / $K_2O$, modified after [54]. $Na_2O$ was below detection limit in SHCORE-ICP02 and $K_2O$ was below detection limit in all three samples. For plotting, detection limit values (0.01 wt. %) are used for the affected analyses.

probably of limited use in understanding the petrology of the rock (due to the effects of silicification noted above). They do, however, have utility as a tool for comparison between the core and potential source regions, where the focus is on the degree of geochemical similarity rather than understanding petrological mechanisms.

In contrast to major element classification schemes, those that use trace elements may be more useful for both petrological and comparative purposes. During silicification by

groundwater [the most likely mechanism for the majority of UK silcretes; 1, 2], many of the high field strength elements (including the REE) and other trace elements are not significantly introduced to the host sediment by the percolating fluid [e.g. 56]. Simultaneously, these elements remain immobile in the host sediment and are not lost [e.g. 40] via dissolution. As such, though their absolute abundances will decrease due to dilution by silica, the magnitude of their ratios will not. The three subsamples plot as a tight cluster either within or close to the fields defined by sandstones sourced from passive continental margins or continental arcs on such classification diagrams (Fig 22).

## 4.5. Whole-rock isotope analysis

**4.5.1. Methodology.** Whole-rock isotope analyses were undertaken to constrain the potential source sediments from which the host sediments in Stone 58 were derived. Two small chips from the Phillips' Core (SHCORE-ISO01 and SHCORE-ISO02; numbers correspond to respective thin-sections) were used for whole-rock isotope analysis (Fig 6). Each subsample was crushed in an impact mortar and pestle and then reduced to a fine powder in an agate ball mill. The resulting samples were transferred to a clean cabinet (class 100, laminar flow) where they were weighed into pre-cleaned Teflon beakers. The sample was mixed with $^{84}$Sr tracer solution and $^{150}$Nd tracer solution and dissolved in cleaned Savillex Teflon beakers using 8M $HNO_3$ and Ultrapur 29M hydrofluoric acid. Samples were converted to bromide form using Ultrapur HBr. Lead was collected using Eichrom AG1 X8 anion resin. The wash from these columns was dried down and converted to chloride form using Teflon distilled 6 M CL. The samples were taken up in calibrated 2.5 M HCl and centrifuged. Strontium and the bulk REE fraction were collected using Eichrom AG50 X8 resin columns. Nd was separated using 2ml of Eichrom Ln-Spec ion exchange resin packed into 10ml Biorad Poly-Prep columns.

Sr was loaded onto a single Re filament [following 57], and the isotope composition and Sr concentrations were determined by thermal ionisation mass spectroscopy (TIMS) using a Thermo Scientific Triton multi-collector mass spectrometer. The international standard for $^{87}$Sr/$^{86}$Sr, NBS987, gave a value of 0.710273 ±.000016 (n = 21, 2σ) during the analysis of these samples and data are corrected to the accepted value for this standard of 0.710250.

Pb isotope analysis of the samples was conducted using a Thermo Fisher Neptune Plus MC-ICP-MS (multi-collector ICP-MS). This mass spectrometer is fitted with the Jet interface, in which enhanced sensitivity is achieved using a large volume interface pump (Pfeiffer On-Tool Booster 150) in combination with the Jet sampler and X-skimmer cones. Prior to analysis, each sample was appropriately diluted (using Teflon distilled 2% $HNO_3$) and spiked with a solution of thallium (Tl), which is added (in a ratio of ~1 Tl:10 Pb) to allow for the correction of instrument-induced mass bias. Samples were then introduced into the instrument via an ESI 50 μl/min PFA micro-concentric nebuliser attached to a desolvating unit (Cetac Aridus II). All isotopes of interest were simultaneously measured using the cup configuration detailed in Table 6. The acquisition consisted of 50 ratios, collected at 8.4 s integrations, following a 60 s de-focused baseline measurement made at the beginning of each analytical session.

The precision and accuracy of the method was assessed through repeat analysis of NBS 981 Pb reference solution (also spiked with Tl). Data are corrected (normalised) relative to the known values for this reference [taken from 58]: $^{206}$Pb/$^{204}$P = 16.9417; $^{207}$Pb/$^{204}$Pb = 15.4996; $^{208}$Pb/$^{204}$Pb = 36.724; $^{207}$Pb/$^{206}$Pb = 0.91488; $^{208}$Pb/$^{206}$Pb = 2.1677. The analytical errors, reported for each of the sample ratios, are propagated relative to the reproducibility of the session NBS 981, to take into account the errors associated with the normalisation process.

For Nd analysis, fractions were dissolved in 1 ml of 2% $HNO_3$ prior to analysis on a Thermo-Electron Neptune mass spectrometer, using a Cetac Aridus II desolvating nebuliser.

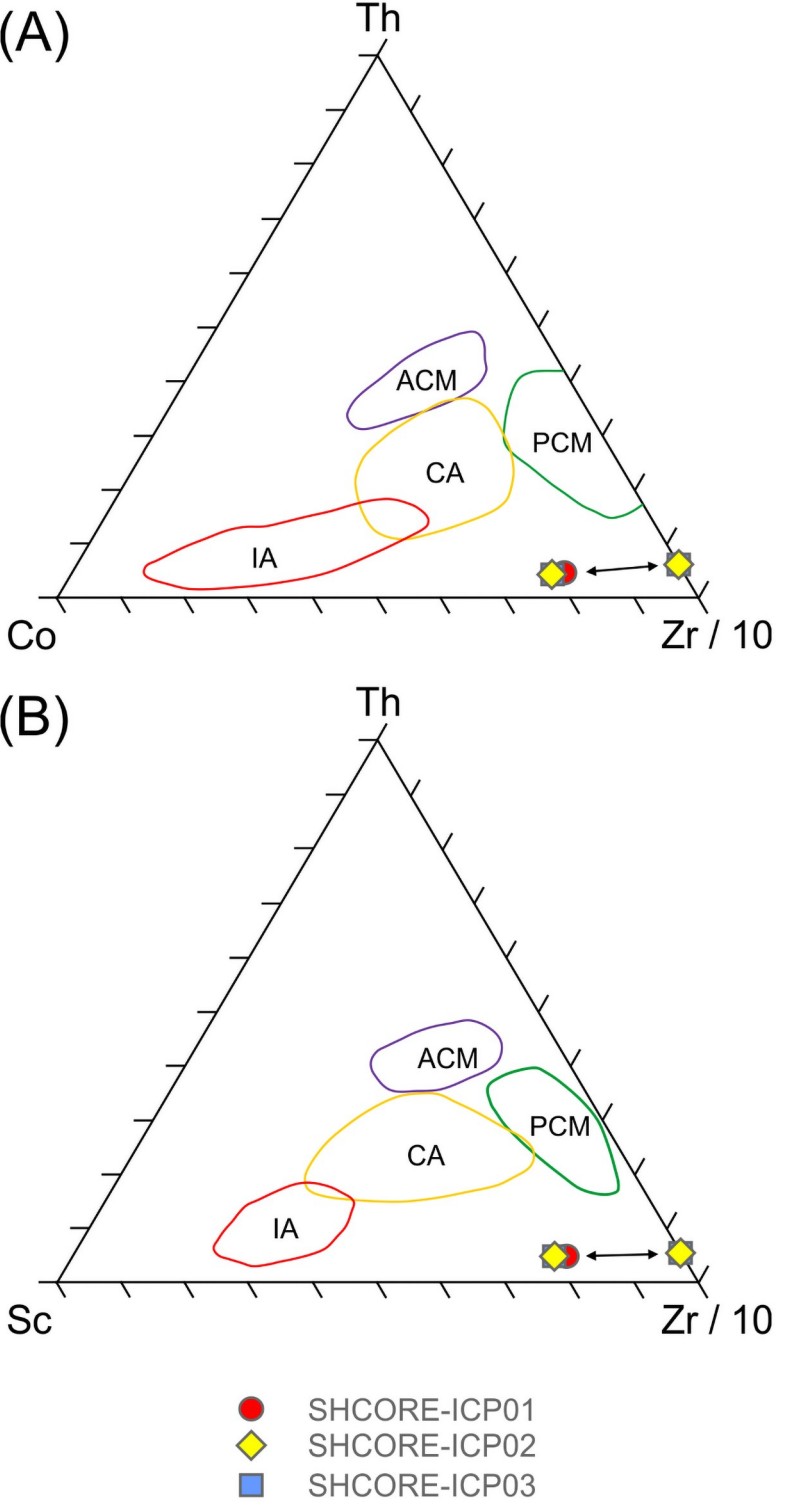

**Fig 22. (A) Th-Zr/10-Co and (B) Th-Zr/10-Sc trace element discrimination diagrams for the three subsamples from the Phillips' Core (Stone 58, Stonehenge).** Co and Sc were below detection limit in all three samples. For plotting, the samples are shown as arrays defined by Sc and Co concentrations at both detection limit (1 ppm) and 0 ppm. Field names: IA—Island Arc; CA—Continental Arc; ACM—Active Continental Margin; PCM—Passive Continental Margin.

**Table 6. Cup configuration for whole-rock isotope measurement.**

| High 4 | High 3 | High 2 | High 1 | Ax | Low 1 | Low 2 | Low 3 | Low 4 |
|--------|--------|--------|--------|------|-------------|---------|------------|--------|
|  | 208Pb | 207Pb | 206Pb | 205Tl*2 | 204Pb, 204Hg | 203Tl*2 | 202Hg*1 |  |

[1] Measured to allow for the correction of the isobaric interference of 204Hg on 204Pb.

[2] Measured to allow for the correction of instrumental mass bias.

0.010 l/min of nitrogen were introduced via the nebuliser in addition to argon in order to min-imise oxide formation. The instrument was operated in static multi-collection mode, with cups set to monitor $^{142}Ce$, $^{143}Nd$, $^{144}Nd$, $^{145}Nd$, $^{146}Nd$, $^{147}Sm$, $^{149}Sm$ and $^{150}Nd$. 1% dilutions of each sample were tested prior to analysis, and samples diluted to ~20 ppb. Jet sample cones and X-skimmer cones were used, giving a typical signal of c. 800–1000 V/ppm Nd. Correction for $^{144}Sm$ on the $^{144}Nd$ peak was made using a ratio for $^{147}Sm/^{144}Nd$ derived from multiple analyses of SpecPur samarium solution. This correction was insignificant due to the efficiency of the column separation. Data are reported relative to $^{146}Nd/^{144}Nd = 0.7219$. The Nd standard solution JND-i was analysed during each analytical session and sample $^{143}Nd/^{144}Nd$ ratios are reported relative to a value of 0.512115 for this standard.

**4.5.2. Results of whole rock isotope analysis.** The data in Tables 7 and 8 show that sam-ples SHCORE-ISO01 and -ISO02 are taken from a "clean" silcrete with few impurities; this is reflected in the low concentrations of Sr (1.6 and 1.3 ppm) and Nd (0.62 and 0.48 ppm) in each sample respectively. The isotope composition reflects the average values of the mineral components that contribute to the silcrete host sediment. The quartz-rich nature of the host sediment means there are very few Rb bearing minerals, which may explain the relatively low average $^{87}Sr/^{86}Sr$ value (0.71348) for the two samples compared with other British sedimentary rocks [59–61]. The $^{146}Nd/^{144}Nd$ ratio of 0.511946 is consistent with sediments with an average crustal residence age of Mesoproterozoic (i.e. 1.6 to 1.0 Gyr ago) [59], assuming a typical Sm/Nd ratio of c. 0.19 [62]. The Pb data are provided for reference, as no published Pb whole rock data from British sedimentary rocks were found for cross-comparison.

The Sr and Nd data are displayed in Fig 23 with published data [from 63] to provide some constraints on the possible types of source sediment. It should be noted that there is limited Sr and Nd isotope data on UK sediments, particularly sandstones. The two core samples have an Nd value that is most closely matched by Mesozoic sedimentary rocks, suggesting that the sar-sen host sediments were derived from eroded sediments of this age; Palaeozoic sedimentary rocks tend to plot with higher $^{143}Nd/^{144}Nd$ values. Aside from Sr and Nd, only one previous study has published isotope data for a British silcrete [17], but for stable oxygen isotopes only.

# 5. Discussion

Using a suite of complementary state-of-the-art methodologies, the preceding sections have provided a petrological and geochemical characterisation of samples taken from Stone 58, one of the large sarsen uprights within the trilithon horseshoe at Stonehenge (Fig 1). Here, we

**Table 7. Sr and Nd concentrations and $^{87}Sr/^{86}Sr$ and $^{143}Nd/^{144}Nd$ isotope ratios for the whole-rock samples from the Phillips' Core at Stonehenge.**

| Sample | Batch | N | Nd ppm | $^{143}Nd/^{144}Nd$ corr | Sr ppm | $^{87}Sr/^{86}Sr$ |
|--------|-------|---|--------|--------------------------|--------|-------------------|
| SHCORE-ISO01 | P894 | 1 | 0.62 | 0.511946 | 1.6 | 0.713688 |
| SHCORE-ISO02 | P894 | 2 | 0.48 | 0.511946 | 1.3 | 0.713261 |

**Table 8. Pb isotope composition and uncertainties for the whole-rock samples.**

| Sample | $^{206}Pb/^{204}Pb$ | 2s % | $^{207}Pb/^{204}Pb$ | 2s % | $^{208}Pb/^{204}Pb$ | 2s % |
|---|---|---|---|---|---|---|
| SHCORE-ISO01 | 20.4267 | 0.0107 | 15.7128 | 0.0052 | 38.8567 | 0.0064 |
| SHCORE-ISO02 | 19.9937 | 0.0108 | 15.7228 | 0.0052 | 38.9146 | 0.0063 |
| | $^{207}Pb/^{206}Pb$ | 2s % | $^{208}Pb/^{206}Pb$ | 2s % | | |
| SHCORE-ISO01 | 0.7692 | 0.0055 | 1.9023 | 0.0061 | | |
| SHCORE-ISO02 | 0.7864 | 0.0055 | 1.9464 | 0.0061 | | |

summarise the key properties of the stone and then draw comparisons with published analyses of sarsens elsewhere, both at Stonehenge and in other areas of southern Britain.

## 5.1 Synthesis of the geological properties of Stone 58

Petrographic analyses of subsamples from the Phillips' Core demonstrate that Stone 58 is a highly indurated, grain-supported, structureless and texturally-mature groundwater silcrete, comprising predominantly fine-to-medium grained detrital quartz sand cemented by optically-continuous syntaxial quartz overgrowths. Optical microscopic and QEMSCAN results indicate that, in addition to detrital quartz, silica-rich rock fragments and (in order of decreasing abundance) trace quantities of Fe-oxides/hydroxides, kaolinite, chlorite, Ti-oxides, Fe-bearing Ti-oxides, calcite, Mg-tourmaline, tourmaline, pyrite, plagioclase, kyanite, illite and illite-smectite, Fe-illite and Fe-illite-smectite, zircon, dolomite, staurolite, muscovite, ferroan dolomite, K-feldspar, biotite, chromite, apatite and baryte are also present.

Optical microscopic and SEM-CL analyses show that the quartz overgrowth cement developed in multiple phases. These phases include an initial typically <10 μm thick zone of non-luminescing quartz cement that infills irregularities in quartz grain surfaces, followed by up to 16 distinct quartz cement growth zones that culminate in crystal terminations in open pore spaces. Late-stage Fe-oxides/hydroxides and Ti-oxides line and/or infill some of these pores;

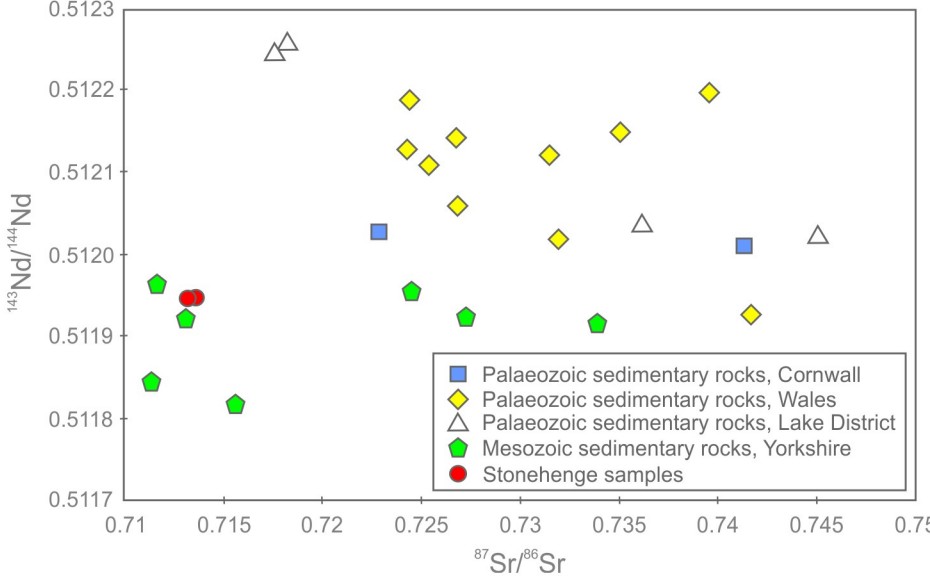

**Fig 23. Sr and Nd isotope data for the whole-rock samples from the Phillips' Core at Stonehenge plotted alongside equivalent published data for UK lithologies [data from 63].**

these are most prevalent in a zone close to an original fracture surface in the Phillips' Core and were likely introduced into Stone 58 while it was still in a sub-surface environment. As proposed for saccharoid sarsens in Sussex, this ferruginisation may have accompanied break-up of the original sarsen/silcrete body and its inclusion in Clay-with-Flints (a residual deposit formed from the dissolution, decalcification and cryoturbation of the Chalk Group and Palaeogene formations) [16]. QEMSCAN analyses indicate that the silcrete preserves 7.2 to 9.2 area % porosity as a moderately connected intergranular network.

ICP-AES and other geochemical data indicate that the silcrete is chemically very pure, comprising $\geq$ 99.7 wt. % $SiO_2$. pXRF and XRF core scanner data show that the major and trace element chemistry is also highly consistent, with the only magnitude variations being observed in Fe content. The non-quartz accessory minerals within the silcrete host sediments impart a trace element geochemical signature distinct from 'standard' sedimentary and other crustal materials. $^{143}Nd/^{144}Nd$ isotopic analyses suggest that the detritus that makes up the sarsen host sediment was likely derived from eroded Mesozoic rocks and that these Mesozoic rocks incorporated much older Mesoproterozoic material.

## 5.2 Comparison with previous analyses of sarsens at Stonehenge

The results in this study represent the first comprehensive analysis of sarsen samples taken directly from one of the Stonehenge megaliths. The petrographic and mineralogical data in section 3 can now be compared with previous analyses of sarsen debitage from the monument. As noted in section 1, the earliest detailed description of sarsens at Stonehenge was by John Wesley Judd, who worked on rock fragments excavated in 1901 by William Gowland from an area around Stone 56. The report on the excavations [8] is written mainly by Gowland, with a 13-page "Note on the Nature and Origin of the Rock-fragments found in the Excavations" provided by Judd (pp.106-118). In his preamble to the 1902 report, Gowland notes that:

"...the sarsens in their composition are sandstones, consisting of quartz sand, either fine or coarse, occasionally mixed with pebbles and angular bits of flint, all more or less firmly cemented by silica... They range in structure from a granular rock resembling loaf sugar [saccharoid sarsen] in internal appearance to one of great compactness similar to and sometimes passing into quartzite [likely hard sarsen]. The monoliths and trilithons all consist of the granular rock"

[8, p.45].

Judd's descriptions of sarsen fragments build upon this overview:

"In some cases they are coarse-grained, in others very fine-grained; the sand grains of which they are composed are sometimes well rounded, at other times angular. Other minerals present within them besides quartz are feldspars more or less altered, mica and glauconite, whilst chips of flint are in some cases not rare. The grains sometimes show only a small quantity of cement between them; at other times this siliceous cement is large in quantity and the outlines of the original grains can be traced only with difficulty, the rock being almost indistinguishable from quartzite. Occasionally the cement around the grains shows [a] radiated or spherulitic appearance"

[8, p.109-110].

It is not always easy to tell from Judd's account if he is describing hard or saccharoid sarsen (or indeed both). Further, his descriptions of mineralogy suggest that some glauconitic

sandstone fragments may have been incorporated into his 'sarsen' sample. However, his account includes a separate analysis of small chips broken from eight hard sarsen hammerstones [8, p.114]. By comparing Judd's notes on these hammerstones with his more general descriptions of sarsens, it is evident that the fragments in which the "siliceous cement is large in quantity" and the "outlines of the original grains can be traced only with difficulty" are saccharoid sarsen and are essentially the same material as described from Stone 58.

No new information on the lithology of the Stonehenge sarsens was added until the work of Hilary Howard in 1982 [pp.119-123 in 12]. Howard analysed thin-sections of 12 saccharoid and 14 hard sarsen fragments from Mike Pitts' 1979 and 1980 excavations [12–14] and conducted heavy mineral analyses of selected fragments and natural sarsens. Howard's thin-section descriptions drew upon the now widely adopted terminology for silcrete fabrics introduced by Mike Summerfield in the late 1970s and early 1980s [64], making her findings directly comparable with those in this study. We focus here on her results for saccharoid sarsen fragments only [12, p.120-123]; note that we do not know if these fragments came originally from a single dressed boulder or multiple stones or, indeed, if the stone(s) from which the fragments were struck remain at Stonehenge.

Howard's saccharoid sarsen samples all exhibited a grain-supported fabric and displayed a "remarkable macroscopical similarity" [12, p.121-122]. Quartz grains within the thin-sections were generally more rounded than in her hard sarsen samples and ranged from 0.2 to 0.5 mm diameter (fine to medium sand). Howard described the quartz cement within the saccharoid sarsens as ". . .extremely sparse, and when present is always microcrystalline in nature", noting also that the "matrix crystals are distinctly angular" [12, p.122]. Howard further observed that the quartz cement within her samples was sometimes obscured by ferruginous staining, with patches of "brown haematite" (most likely limonite) observed "clustering around individual grains" [12, p.122].

Howard's thin-section descriptions compare well in terms of grain size variability with those of the subsamples from Stone 58 (section 3.3). There are, however, significant differences in petrography. First, Howard's descriptions indicate that her saccharoid sarsen fragments contain different types and quantities of cement compared to Stone 58 –respectively microquartz versus quartz overgrowths, and sparse versus extensive cements. Second, her samples appear to contain more progressive ferruginisation by late stage Fe-minerals. The latter difference could simply be the product of the residence time in a sub-soil environment for the buried sarsen fragments. The contrasting cements are, however, less easily explained. One possibility is that Howard's samples comprise material dressed from the less well-cemented outer surface of one or more sarsen boulders. Alternatively, the contrast could reflect a fundamental petrological difference between Stone 58 and the original sarsen(s) from which Howard's samples were struck.

Howard assessed the relative abundance of heavy minerals in five saccharoid sarsen fragments using a four-point ordinal scale with the categories 'very rare', 'rare', 'present' to 'abundant' [see Table 3 in 12]. In addition to ubiquitous non-magnetic opaque minerals (likely Ti-oxides): zircon and rutile were identified in all five samples in quantities ranging from 'rare' to 'abundant'; andalusite occurred in all samples and ranged from 'very rare' to 'present'; staurolite was 'present' to 'abundant' in all samples; kyanite was 'present' in three of the samples only; tourmaline and garnet were present in two and one samples, respectively, where they were 'very rare'.

The suite of non-quartz accessory minerals described by Howard [12] is broadly similar to that from the Phillips' Core. However, the specific minerals identified, and their relative abundances differ. For example, neither garnet nor rutile were detected via optical microscopy in Stone 58, while zircon is less abundant and tourmaline much more abundant in Stone 58 than

in Howard's descriptions. The lack of garnet in Stone 58 could, in part, be diagenetic. As noted in section 4.4.2, the positively inclined HREE trend within the UCC-normalised trace element geochemical signature for the Phillips' Core (Fig 20) may suggest that garnet was once present in the sarsen host sediments but was altered prior to or during silicification. The rarity or absence of tourmaline within Howard's results is, however, surprising. Tourmaline is ubiquitous within the thin-sections from Stone 58 and in descriptions of sarsen from across southern Britain. It would appear that, either tourmaline has been under-identified in Howard's analyses or—combining her thin-section and heavy mineral data—the selected fragments from Pitts' excavation were all struck from a boulder (or boulders) of less well-cemented, 'tourmaline-absent' sarsen. If the latter is true, then Howard's samples are likely to be fragments from a mineralogically distinct sarsen to Stone 58 –possibly Stone 26 or 160, identified by Nash et al. [5] as chemically different to other sarsens at Stonehenge, or one or more of the ~28 sarsens missing from the monument today [3].

## 5.3 Comparisons with sarsens elsewhere in southern Britain

Although the literature on sarsens is plentiful, there is comparatively little detailed work concerning their micromorphological variability. Comparing the petrography of the subsamples from Stone 58 with available published accounts suggests that their essential mineralogy and textures are typical of saccharoid sarsens from across southern Britain. The majority of petrological studies of sarsens confirm a simple micromorphology consistent with an origin as a groundwater silcrete [2, 24], with samples commonly exhibiting a grain-supported fabric cemented by optically-continuous syntaxial quartz overgrowths and/or microquartz and/or 'chalcedony' and/or cryptocrystalline silica [15–17, 65–72]. Stone 58 is relatively unusual in this respect in having a cement approaching 100% quartz overgrowths on framework grains. This suggests that the original sarsen host sediments were extremely 'clean' (i.e. they had a very low initial clay content), since the presence of clay minerals inhibits the development of quartz overgrowths [64] and instead favours a microquartz-dominant or poorly crystalline silica cement [55]. QEMSCAN results (section 3.4.2) support this interpretation by showing that the trace quantities of clay minerals present within the sarsen occur mainly as late-stage void linings.

Published geochemical data for saccharoid sarsens are even more rare than detailed petrographic descriptions and, where available, include analyses of major and only selected trace elements. Comparisons of various geochemical datasets from the Phillips' Core (e.g. pXRF, XRF scanner, μXRF, ICP-AES) with equivalent published data for sarsens [16, 69, 70, 72, 73] suggest that—while at the upper end in terms of wt. % $SiO_2$ content—Stone 58 is typical of saccharoid sarsens elsewhere in southern Britain. The consistency in geochemistry across the three samples from Stone 58 is also in line with results from Ullyott and Nash [16], the only previous study to make multiple chemical assays from a single sarsen boulder. In common with other sarsens [e.g. 16, 72], the Fe content in the samples from Stone 58 increases in visible zones of ferruginisation, but still constitutes a minor component of the rock chemistry.

The only studies to include extensive trace element data for sarsens are the recent papers by Nash et al. [5] and Day [73]. Using these data as a basis for discrimination, differences between Stone 58 and sarsens elsewhere in Britain can be assessed. Nash et al. [5] focus on a selected suite of 21 immobile trace elements (section 4.4.2) to compare the composition of Stone 58 presented here with equivalent data for 20 sites across southern Britain. Sarsens from the Valley of the Stones (Dorset), Blue Bell Hill and Lenham Quarry (both Kent), Sudbury (Suffolk) and one site at Gestingthorpe (Essex) exhibit different immobile trace element signatures for all elements except Hf and Sr. Other sites in Wiltshire, Devon, Hampshire, Sussex, Essex and Norfolk show closer immobile trace element signatures to Stone 58, but still differ in at least

two element ratios. The only site with the same immobile trace element signature to Stone 58 is West Woods in Wiltshire, 25 km north of Stonehenge, suggested by Nash et al. to be the source of the sarsen.

Day [73] provides whole-rock compositional data for nine fragments of sarsen collected from the vicinity of the Medway megalith structures at Kit's Coty, north Kent. Based on visual assessment only, Day interprets these fragments as debitage left over from the dressing of the sarsens that form the megalith structures. Treating Day's whole-rock compositional data using Nash et al.'s [5] method allows a direct comparison between the data for sarsens from Kent in the two studies and the equivalent data for Stone 58. Such a comparison supports the conclusion made by Nash et al. that sarsens from Kent exhibit a different immobile trace element signature to Stone 58, with Day's sarsen samples differing in three or more of the 21 element ratios used. It also suggests that there may be greater chemical variability across the sarsen clusters remaining in Kent than was captured by Nash et al.

Day's paper [73] includes limited mineral compositional data (for detrital zircon and Ti-rich grains) and $^{207}Pb/^{206}Pb$ ages for a small number of detrital zircons within the sarsen fragments. These types of analysis remain productive avenues for future sarsen provenancing studies. However, as noted in section 1, we were unable to determine reliable zircon ages from Stone 58 owing to a paucity of large zircon grains. As such, unless technology improves, the future application of these types of mineral compositional investigations to Stonehenge would need to focus on minerals other than zircon.

## 6. Conclusions

Using a suite of state-of-the-art methods, this paper has presented an in-depth characterisation of the sedimentology, mineralogy and geochemistry of a known sarsen megalith (Stone 58) at Stonehenge, yielding the first detailed insights into the primary stone type used in the construction of the monument. The main findings of the study are summarised in section 5.1. No previous investigation has analysed a single sarsen boulder with such a range of complementary techniques, meaning that comparisons with previous studies are limited to selected rock properties. Aside from being almost exclusively cemented by optically-continuous syntaxial quartz overgrowths, the petrography of Stone 58 is otherwise unremarkable compared to other sarsens. Likewise, the major element geochemistry of the stone is similar to the small number of chemical datasets available for sarsens. Where Stone 58 differs to sarsens at other sites across southern Britain is in its immobile trace element geochemistry. Selected geochemical data from the Phillips' Core have already been used by Nash et al. [5] to identify the source provenance of Stone 58 and, by inference, all but two of the extant sarsens at Stonehenge, to the West Woods area southwest of Marlborough, Wiltshire. This paper has presented a range of significant new datasets for Stone 58 that could be used in future studies to further constrain this provenance. Of these, studies that focus on the host sediment properties, and, in particular, sediment mineralogy and chemistry of sarsen source areas, are likely to be the most productive. The novel combination of analytical approaches employed here will also be of use in the wider context of stone procurement studies, not only for sarsens within the Stonehenge landscape but anywhere in the world where silcrete was used in an archaeological context.

## Supporting information

**S1 Data. Portable XRF data at ~5 cm intervals for the full length of the Phillips' Core from Stone 58 at Stonehenge.**
(XLS)

**S2 Data. XRF core-scanner data of section 2–3 of the Phillips' Core from Stone 58 at Stonehenge.**
(XLS)

**S3 Data. ICP-AES and ICP-MS data for three samples from section 2–3 of the Phillips' Core from Stone 58 at Stonehenge.**
(XLS)

**S4 Data. Quality control data for ICP-AES and ICP-MS analyses of three samples from section 2–3 of the Phillips' Core from Stone 58 at Stonehenge.**
(XLSX)

**S1 Movie. Three-dimensional reconstruction and full simulation of X-ray computed tomography data.**
(WMV)

## Acknowledgments

**General**: The authors would like to thank the Phillips family for access to their recollections, Martin Allfrey for permission to analyse and sample the Phillips' Core, Adrian Green for permission to inspect the Salisbury Museum Core, and ALS Minerals (Sevilla, Spain) for ICP-MS and ICP-AES analyses. Core cutting was undertaken by Joe Shaw and Rich Turley (University of Bristol). Subsampling and thin-section preparation was carried out by Michelle Higgins (Open University). SGS Canada Inc. are thanked for QEMSCAN instrument time and Michael Owen / ThermoFisher for the use of iDiscover software.

## Author Contributions

**Conceptualization:** David J. Nash, T. Jake R. Ciborowski, Timothy Darvill, Mike Parker Pearson, J. Stewart Ullyott.

**Data curation:** David J. Nash.

**Formal analysis:** David J. Nash, T. Jake R. Ciborowski, Timothy Darvill, Mike Parker Pearson, J. Stewart Ullyott, Magret Damaschke, Jane A. Evans, Steven Goderis, Susan Greaney, Jennifer M. Huggett, Robert A. Ixer, Duncan Pirrie, Matthew R. Power, Tobias Salge, Neil Wilkinson.

**Funding acquisition:** David J. Nash, T. Jake R. Ciborowski, Timothy Darvill, Mike Parker Pearson, J. Stewart Ullyott, Duncan Pirrie.

**Investigation:** David J. Nash, T. Jake R. Ciborowski, J. Stewart Ullyott, Magret Damaschke, Jane A. Evans, Steven Goderis, Susan Greaney, Jennifer M. Huggett, Robert A. Ixer, Duncan Pirrie, Matthew R. Power, Tobias Salge, Neil Wilkinson.

**Methodology:** David J. Nash, T. Jake R. Ciborowski, Magret Damaschke, Jane A. Evans, Steven Goderis, Jennifer M. Huggett, Robert A. Ixer, Duncan Pirrie, Matthew R. Power, Tobias Salge, Neil Wilkinson.

**Project administration:** David J. Nash.

**Resources:** David J. Nash.

**Validation:** David J. Nash, T. Jake R. Ciborowski, Magret Damaschke, Jane A. Evans, Steven Goderis, Jennifer M. Huggett, Robert A. Ixer, Duncan Pirrie, Matthew R. Power, Tobias Salge, Neil Wilkinson.

**Visualization:** David J. Nash, T. Jake R. Ciborowski, Magret Damaschke, Jane A. Evans, Steven Goderis, Robert A. Ixer, Duncan Pirrie, Matthew R. Power, Tobias Salge, Neil Wilkinson.

**Writing – original draft:** David J. Nash, T. Jake R. Ciborowski, Timothy Darvill, Mike Parker Pearson, J. Stewart Ullyott, Magret Damaschke, Jane A. Evans, Steven Goderis, Susan Greaney, Jennifer M. Huggett, Robert A. Ixer, Duncan Pirrie, Tobias Salge.

**Writing – review & editing:** David J. Nash, T. Jake R. Ciborowski, Timothy Darvill, Mike Parker Pearson, J. Stewart Ullyott, Magret Damaschke, Jane A. Evans, Steven Goderis, Susan Greaney, Jennifer M. Huggett, Robert A. Ixer, Duncan Pirrie, Matthew R. Power, Tobias Salge, Neil Wilkinson.

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
