## [Decision Letter · Decision Letter 0]

21 Dec 2020

PONE-D-20-32409

Petrological and geochemical characterisation of the sarsen stones at Stonehenge

PLOS ONE

Dear Dr. Nash,

Thank you for submitting your manuscript to PLOS ONE. After careful consideration, we feel that it has merit but does not fully meet PLOS ONE’s publication criteria as it currently stands. Therefore, we invite you to submit a revised version of the manuscript that addresses the points raised during the review process.

I've now received three reviews of your manuscript, written by both geologists and geo-archeologists. All of them, including I, agree that the data presented are sound and the interpretations are well supported. However, Reviewer #1, who undoubtfully did a very thorough reading of the manuscript, raised two major concerns that should be attended to, for the manuscript to be improved and ready for resubmission. The first and more serious issue raised by Reviewer #1 is the rather 'technical' way of arranging the results by methods. I agree that combining results from various methods for each mineral will make the manuscript more fluent and easier to read. Following Reviewer #1's second comment will help shorten the manuscript. Reviewer #1 also had comments and suggestions for improvement in many specific places in the text, figures and figure captions. Please  refer to each of these comments. The two other  reviewers also had useful comments, including questioning the necessity of using the term 'silcrete', while 'quartz arenite' is the true rock name of the studied stone. 

We look forward to receiving your revised manuscript.

Kind regards,

Prof. Yaron Katzir

Academic Editor

PLOS ONE

2. In your manuscript, please provide additional information regarding the specimens used in your study. Ensure that you have reported specimen numbers and complete repository information, including museum name and geographic location. If permits were required, please ensure that you have provided details for all permits that were obtained, including the full name of the issuing authority, and add the following statement to both the Ethics Statement and Methods section: 'All necessary permits were obtained for the described study, which complied with all relevant regulations.' If no permits were required, please include the following statement in the aforementioned sections: 'No permits were required for the described study, which complied with all relevant regulations.' For more information on PLOS ONE's requirements for palaeontology and archaeology research, see https://journals.plos.org/plosone/s/submission-guidelines#loc-paleontology-and-archaeology-research.

4.Thank you for stating the following in the Financial Disclosure section:

"DJN, TJRC, JSU, MPP, TD - awarded British Academy / Leverhulme Trust Small Research Grant SG170610 https://www.thebritishacademy.ac.uk/

DP received additional funding support from the University of South Wales CESRIS grant (no number) https://www.southwales.ac.uk/

We note that one or more of the authors are employed by a commercial company: Gatan UK, Vidence Inc. and Petroclays Ltd

5.We note that Figure 2 and 4 includes an image of a patient / participant in the study. 

6. We note that [Figure(s) 1] in your submission contain map images which may be copyrighted. All PLOS content is published under the Creative Commons Attribution License (CC BY 4.0), which means that the manuscript, images, and Supporting Information files will be freely available online, and any third party is permitted to access, download, copy, distribute, and use these materials in any way, even commercially, with proper attribution. For these reasons, we cannot publish previously copyrighted maps or satellite images created using proprietary data, such as Google software (Google Maps, Street View, and Earth). For more information, see our copyright guidelines: http://journals.plos.org/plosone/s/licenses-and-copyright.

1.    You may seek permission from the original copyright holder of Figure(s) [1] to publish the content specifically under the CC BY 4.0 license. 

Reviewers' comments:

Reviewer's Responses to Questions

**Comments to the Author**

1. Is the manuscript technically sound, and do the data support the conclusions?

Reviewer #1: Yes

Reviewer #2: Yes

Reviewer #3: Yes

2. Has the statistical analysis been performed appropriately and rigorously? 

Reviewer #1: N/A

Reviewer #2: Yes

Reviewer #3: N/A

3. Have the authors made all data underlying the findings in their manuscript fully available?

Reviewer #1: Yes

Reviewer #2: Yes

Reviewer #3: Yes

4. Is the manuscript presented in an intelligible fashion and written in standard English?

Reviewer #1: Yes

Reviewer #2: Yes

Reviewer #3: Yes

5. Review Comments to the Author

**Reviewer #1**: Dear editor and authors,

The paper by Nash et al shows an impressive and comprehensive geochemical, mineralogical and petrographical analysis of sarsen stone 58 from Stonehenge with the aim of revealing some information about the geological history of the rock itself, and forming a reference data set for future comparison. This is important work for both geological (as these rocks aren't targeted in research too often) and heritage purposes.

I have major issues with the paper:

1. The overall structure is based on the methods. Each chapter describes a method, and what can be learned by applying the method. This leads to non-ideal flow of the manuscript. For example, the first petrography describes TiO2 oxides, but how can this be known at this early stage? Or, most sections describe that the rocks are almost pure silica. This is repetitive. Instead, I would recommend structuring the paper by describing minerals using all methods at once (after they've been properly described in the methods section). For example, a section on quartz, synthesising all data from CL, optical petrography, etc. A section on TiO2, synthesising all data from all methods. Then a section on whole rock geochemistry, saying that it's pure silica based on method A B and C. In my opinion, the paper will be more interesting, easier to read, and shorter. 

2. The other issue is the choice of methods. Some methods are superior to others - ie pxrf vs the other xrf methods. Or, QEMSCAN vs the high-resolution mapping on the XMax. Why are the inferior methods included, and what do they tell us that we don't know from the better methods? The papers states the the methods are complementary, but I am missing some statement on exactly what is the unique thing that we learn from each method.

That's mostly it. I don't think it should be too hard to address these two issues.

A disclaimer - Although I know quite a lot about the analytical methods employed here, I admit I do not know much on English geology, and sedimentary petrology. These two topics have received little attention in my review. 

Specific comments below (written as I was reading the paper):

line 60: Is there any evidence that sarsens were less interesting? The fact that we have no surviving accounts of the sarsens from that time does not mean that they were not interesting.

line 65: "modern methods" is ambiguous. Atom probe, for example, may be modern for 2020, did not exist in the 1980s, and may be obsolete in 2040.

line 74: Define what and when is "Bronze Age".

figure 1: It's unclear what "stone 58" is. I assume that it refers to the single stone on the top, but it could be misinterpreted as suggesting the entire circled region comprising of two stones is "stone 58". Adding an arrow should resolve this issue.

Also, why is 'Bluestone' is in quotes? If it's a vernacular term, how is that different to sarsen, which is not in quotes?

figure 5: If there's no coarse sand, why include it in the figure? Remove and simplify. And is it possible to add the actual sizes for the different sizes?

line 271-272: This part is a bit unclear. You are describing your work in order, so at this stage you do not know yet that those minerals are TiO2. And are the TiO2 larger or smaller than 2 um? It's also unclear. TiO2 does not have "constituent minerals". There are several minerals, for example rutile or anatase, that are composed of TiO2. I would rewrite this as "Some fine-grained minerals grains could not be optically identified, but were later revealed to be TiO2. Whether they are rutile, anatase, or brookite remains to be determined." Or something like that.

line 276: What is the scale per pixel?

line 281: How was 187 um determined? Automated petrography? Manual estimates? What's the standard deviation on that number? And remember that you are looking at a section. Unless you cut all grains exactly in their middle, you are underestimating their true size. There are methods to infer the real size from the observed size - this book has some approaches: https://doi.org/10.1017/CBO9780511535574 I'm not sure how important this is to your study, but might as well try. Shouldn't be too difficult.

figure 10: Can you annotate the figure to show where the accessory minerals and dust lines are?

line 290: I don't see polycrsytalline quartz in fig 10 A or B. Also, "metamorphic origin" is a bit broad. Do you mean mylonitic quartz? And it implies the other quartz is not "metamorphic", but you do not know that.

line 292: IMA-approved spelling is "hematite".

line 293: How did you distinguish magnetite from hematite? They are notoriously challenging to tell apart.

line 296: Well you did find feldspars in QEMSCAN, so having it here is reasonable.

line 299: Add a figure for this section?

line 306: You said earlier that your zircon was not suitable for dating. 200x60 um zircon is as good as it gets for dating, so this is contradictory.

line 312: I would recommend against identifying the spinel as chromite based on whole rock data. It may be spinel, or not. Cr2O3 can go into many of the other phases as well. (after seeing Table 2 - well if you identified it in QEMSCAN, why this phrasing? Just say it's chromite, and why do you even need then whole rock data?).

table 2: I would combine the two illites. This is a very fine grained mineral, and the QEMSCAN cannot distinguish illite from Fe-illite from an intergrowth of illite and hematite. Just call everything illite. IMA-approved spelling is "baryte". I'm also curious about the tourmalines. Mg-tourmaline (dravite) is distinguished from a tourmaline with no chemical information - is this a schorl? In my opinion, just combine all tourmalines into one. They are strongly zoned minerals, and the distinction is probably the result of the beam hitting a specific zone.

line 432: What are the wavelengths for the RGB filters that you used?

line 440: What you're describing here is essentially QEMSCAN on steroids. The XMax 80 detector has much better resolution than the Bruker detectors installed on QEMSCAN, and you have a resolution of 3.2 um per pixel, much better than the QEMSCAN 10 um. This raises the question - why was QEMSCAN conducted, if a superior method is already described here?

line 464: If that's the case, you wouldn't see blue, you'd see only red. I reckon any beam damage is minor, and you can either exclude this statement or add a caveat that it's probably minor.

line 528: Then what is the purpose of pxrf? You already have core scanner XRF data which is much better. The only reason I could think of why pxrf data would be useful is to have some reference when someone goes with a pxrf to the field and tries to see if an arbitrary sarsen is somewhat similar to those from Stonehenge. If that's the case, then say it.

figure 15 and related text: The correct spelling is "sulfide". This is not an American/British spelling issue. The British Royal Society of Chemistry, IUPAC, and other UK-based organisation has long ago adopted "sulfide". Please, let's use the correct spelling. And, the lack of correlation between Fe and Cu or Zn only means that Fe variation is not related to chalcopyrite or sphalerite, not to sulfides in general. You could have pyrite-hosted Fe. That said, your work so far has showed that Fe is overwhelmingly hosted in oxides and hydroxides. Was the correlation with chalcopyrite or sphalerite even an issue?

line 600/fig 17: Since Zr is a trace element in your rock most pixels should result in zero Zr, with very strong Zr peaks when it hits a zircon. I'd expect it to look like the Fe map - dark grey with several white pixels. Instead, it's all ~homogeneous medium grey. To me, this appears as if it did not hit a single zircon, and instead it is mapping background noise. Can you clarify this please?

fig 18: The rainbow colour scheme is not a good map to use. If you have access to the data or software and can change the colour scheme to something else, I strongly recommend to do so. Read this for the motivation: https://doi.org/10.1038/s41467-020-19160-7

line 627: Rb is one of the most mobile elements known. Sr and U (when hexavalent) are also rather mobile.

line 360-361: Calling these major elements is not accurate. They are commonly considered as major elements in many rocks, but if they're not major, then they're not major elements.

figure 19: Your data shows Eu bdl in two samples, yet your lines are continuous through Eu in fig 19. This is creating the false impression that there is no Eu anomaly (which most likely exists). One way to solve this is to make the line connecting Sm and Gd dashed or dotted. Regarding normalisation factors - the M&S paper is from 25 years ago. We have much better data today. It would probably not matter much, but I would consider it best practice to use newer values. My personal favourites are the ones available here: https://doi.org/10.1093/petrology/egw047 column CI in table 1.

Table 5: It's customary to sort trace elements by atomic number, not by name. The the REE follow a logical order.

line 651: Not necessarily non-standard. There aren't much REE in quartz, so your REE budget is dominated by the tiny amount of other minerals in your rock. Zircon is one mineral that comes into mind, and probably tourmaline. The REE pattern then reflects the sorting of those minerals during sedimentary transport, and does not necessarily imply that the material was "non-standard".

figure 20: This really shows the elements. You have high in Zr and Hf, and likewise high HREE. I'm almost certain that your HREE signal is coming from zircon. Nb and Ta are also a bit high, which is what you would expect from rutile (a titanium oxide), which is also common in your rock. Uranium is concentrated in hematite. Cr and V in magnetite and spinels.

line 682: No, the upwards HREE trend is probably real. As I said above, it's probably coming from zircon. It could also be from the clays, HREE tend to adsorb into them.

figure 22: Sc and Co are bdl, so placing points on a ternary that includes them is misleading. This can be easily misused by others, and the points plotted in subsequent papers showing a "Stonehenge" field, without actually reading your paper (happens too often, unfortunately). A better way to do it would be to plot the points twice - once at the detection limit, and another time at zero Sc/Co, and draw a line between them. Then you're showing the possible range, and better representing the limited data available.

line 761: There are few/no Rb-bearing minerals at this moment, but they were clearly there when the Sr isotope signature formed. 0.713 is anything but low - it's a reasonable continental crust value. It is all relative though - I mostly work on mantle rocks where "relatively low" is 0.702 and 0.713 would be extremely radiogenic.

line 762: Not quite - crustal residence age is not the time it has been in a sedimentary environment. It has been the time since it was separated from the mantle. It could have been in igneous rocks ever since.

line 763: Missing space? c.0.19

line 787: As said before, the distinction between Mg-tourmaline and tourmaline, the illites, dolomite and Fe-dolomite is an arbitrary divide set by the person who designed the QEMSCAN mineral identification list, and most likely has no petrological significance. For simplicity, I would group them together into just tourmaline, illite, and dolomite.

line 800: That's an example of the issue of structuring the paper according to methods. It's not only ICP, everything you have indicates the the silcrete is pure.

line 806: What is the evidence for two cycles?

**Reviewer #2**: This is an incredibly detailed and excellent study of the Sarsen Stone 58 at Stonehenge, and should be published as is!

The only minor edits/additions I would suggest are:

- p.4, para 2, line 81ff: give some brief reasons as to why all these methods were necessary to use in the analysis?

- p. 27, end of para 2, line 670: what does the resemblance to 'Archean sandstones' imply?

- p.35, para 3, line 926: what is the proximity of West Woods to Stonehenge?, and therefore the feasibility of moving sarsens from there to Stonehenge?

**Reviewer #3**: This paper reports on an exhaustive investigation of an important archeological artifact, and as such the results have potentially important scientific and cultural significance. The depth and detail of the analyses may exceed that of any previous study of a quartz sandstone, and perhaps of any other terrestrial rock sample! This makes for rather a long manuscript, but it is very clearly written, well-organized, and easy to read. The data employed all seem appropriate and reliable, and the conclusions generally appear well-supported by the data.

I do have two specific comments aimed at further improving the manuscript. First, the introduction might be improved by a more explicit statement of the problem or hypothesis being addressed. Likely this seems self-evident to the authors but it might not be so to the reader. I surmise that the main question is ‘what is the provenance of this and other sarsen stones?’ If so, it would be nice to have some information in the introduction (or discussion) regarding specific source candidates. No doubt specialists in this arena are already familiar with the alternatives, but adding a few lines for the benefit of the broader readership would be helpful.

Secondly, the manuscript correctly notes several times that the studied sample is “technically” a quartz arenite or orthoquartzite. Why then the continued use of the term “silcrete?” Silcrete carries a specific genetic implication, i.e., silica precipitation associated with soil formation. If this is indeed the preferred interpretation then it needs to be explicitly defended based on the data presented in the study. At present there is passing mention of cement precipitation from groundwater, but such precipitation is not unique to silcrete. Most (all?) cements precipitate from groundwater. Syntaxial, zoned quartz cement is common in many quartz arenites for which there is no association with soil formation. For example, similar cements in the Cambrian of the central U.S. have been attributed to advection of fluids expelled from adjacent basins. If there are cogent arguments in favor of the sarsen stones being silcrete then they should be discussed. If not then I suggest sticking with ‘quartz arenite’ in the interest of precision.

6. PLOS authors have the option to publish the peer review history of their article (what does this mean?). If published, this will include your full peer review and any attached files.

Reviewer #1: No

Reviewer #2: **Yes: **Charles French

Reviewer #3: No

---

## [Author Response · Author response to Decision Letter 0]

12 Apr 2021

This is a copy of the information provided in the document "Core 58 Response to reviewers".

Response to the Academic Editor

I've now received three reviews of your manuscript, written by both geologists and geo-archeologists. All of them, including I, agree that the data presented are sound and the interpretations are well supported. However, Reviewer #1, who undoubtfully did a very thorough reading of the manuscript, raised two major concerns that should be attended to, for the manuscript to be improved and ready for resubmission. The first and more serious issue raised by Reviewer #1 is the rather 'technical' way of arranging the results by methods. I agree that combining results from various methods for each mineral will make the manuscript more fluent and easier to read. Following Reviewer #1's second comment will help shorten the manuscript. Reviewer #1 also had comments and suggestions for improvement in many specific places in the text, figures and figure captions. Please refer to each of these comments. The two other reviewers also had useful comments, including questioning the necessity of using the term 'silcrete', while 'quartz arenite' is the true rock name of the studied stone. 

Thank you for these comments. We would be grateful if you could pass on our thanks to Reviewer #1, in particular, for their incredibly detailed insights, which have improved the manuscript considerably. We address the two major concerns raised by Reviewer #1 on pp.1-2 of this response. We then deal with the detailed comments raised by reviewers #1 (pp.3-9), #2 (p.10) and #3 (pp.11) in turn. We conclude with answers to questions concerning the journal requirements (pp.12-15). 

Response to reviewers

Reviewer #1: 

The paper by Nash et al shows an impressive and comprehensive geochemical, mineralogical and petrographical analysis of sarsen stone 58 from Stonehenge with the aim of revealing some information about the geological history of the rock itself, and forming a reference data set for future comparison. This is important work for both geological (as these rocks aren't targeted in research too often) and heritage purposes.

I have major issues with the paper:

1. The overall structure is based on the methods. Each chapter describes a method, and what can be learned by applying the method. This leads to non-ideal flow of the manuscript. For example, the first petrography describes TiO2 oxides, but how can this be known at this early stage? Or, most sections describe that the rocks are almost pure silica. This is repetitive. Instead, I would recommend structuring the paper by describing minerals using all methods at once (after they've been properly described in the methods section). For example, a section on quartz, synthesising all data from CL, optical petrography, etc. A section on TiO2, synthesising all data from all methods. Then a section on whole rock geochemistry, saying that it's pure silica based on method A B and C. In my opinion, the paper will be more interesting, easier to read, and shorter. 

Response: We thank the reviewer for their opening statements but disagree with their suggestion that section 3 of the manuscript be restructured. This suggestion seems to be predicated on the assertion that we cannot ‘know’ things in one section without having identified them in another. As we discuss on p.3 of this response, this assertion is incorrect – we can, for example, easily identify TiO2 oxides from thin-section analysis alone and the same is true for other methods. The reviewer also suggests that our approach is repetitive. This is partly true regarding descriptions of silica content, but otherwise our methods are complementary and reinforcing not repetitive. 

The reviewer is suggesting that we adopt what might be called a ‘mineral-progressive’ approach to Section 3. However, this would introduce its own repetition. We identify tens of different minerals in our samples – describing each in turn as suggested, first using optical petrography, then QEMSCAN, then SEM-CL and then SEM-EDS, would be incredibly repetitive for the reader. Significantly, it would also risk downplaying important textural and structural information about the sarsen subsamples – key information for future studies. 

Our use of a ‘method-progressive’ approach to the manuscript, discussing techniques and results in turn and then synthesising the data, was deliberate. As Reviewer #1 identifies in their opening comments, the manuscript is intended to be a paper of record and ”a reference data set for future comparison”. We consider it very unlikely that other researchers will have access to the full suite of techniques we have used, but they might be able to use some. Having the paper method-progressive rather than mineral-progressive makes it easier for other researchers to select their equivalent method and directly compare their data to ours (without the pain of disentangling synthesised findings – which would be a product of the reviewer’s suggestion). Reviewer #3 recognises the value of this approach, noting that they found the manuscript “…very clearly written, well-organized, and easy to read.”

Rather than completely restructure section 3, we have added the following text to the final paragraph of the introduction to better explain and justify the paper’s organisation: “The paper unfolds by first detailing the history of the Phillips’ and Salisbury Museum cores. We then review the results of each technique in turn before drawing the key findings together in section 5. We adopt this approach so that each set of results can be evaluated in isolation and – recognising that not all future investigations will have access to the same suite of techniques – that subcomponents of the dataset can be easily compared in follow-on studies.” We have also reviewed the descriptions of silica content in section 3 and removed unnecessary repetition. We hope that this is acceptable.

2. The other issue is the choice of methods. Some methods are superior to others - ie pxrf vs the other xrf methods. Or, QEMSCAN vs the high-resolution mapping on the XMax. Why are the inferior methods included, and what do they tell us that we don't know from the better methods? The papers states the methods are complementary, but I am missing some statement on exactly what is the unique thing that we learn from each method.

That's mostly it. I don't think it should be too hard to address these two issues.

Response: Thanks for this point. Some of the methods we use are, indeed, superior to others (but see p.6 below for our views on QEMSCAN vs. XMax). However, as noted above, we recognise that future studies may not have access to the same range of approaches as covered in our manuscript. As such we consider it important to include as many methods as possible, from relatively basic to state-of-the-art. Incorporating such a wide range of approaches can only make our study more useful to a wider range of future researchers.

To expand on this issue, we have added the text in italics to the penultimate paragraph of the introduction: “The aim of this paper is to document and provide a detailed characterisation of the cores extracted from Stone 58, the first study of its kind for a sarsen megalith at Stonehenge. We do this using a range of techniques (Table 1), selected to include both standard sedimentological approaches and state-of-the-art mineralogical and geochemical methods.”

...and to the final paragraph of the introduction: “We recognise that some techniques have offer better image, spatial, elemental or spectral resolution than others, but by including all here we provide methodological insights that future studies may want to pick up on when evaluating the relative merits of different approaches.”

We have also added brief text at the start of each methodology sub-section to explain the unique aspects that can be learnt from each approach.

A disclaimer - Although I know quite a lot about the analytical methods employed here, I admit I do not know much on English geology, and sedimentary petrology. These two topics have received little attention in my review. 

Specific comments below (written as I was reading the paper):

line 60: Is there any evidence that sarsens were less interesting? The fact that we have no surviving accounts of the sarsens from that time does not mean that they were not interesting.

Response: Thanks for this. We have amended the text to read “Seemingly, the sarsens were considered less worthy of detailed description...”

line 65: "modern methods" is ambiguous. Atom probe, for example, may be modern for 2020, did not exist in the 1980s, and may be obsolete in 2040.

Response: Thanks. We have amended the text to read “The only other 20th century study...”

line 74: Define what and when is "Bronze Age".

Response: Minor revisions to the text mean that the reference to the Bronze Age has been deleted, so there is no longer need for this definition.

figure 1: It's unclear what "stone 58" is. I assume that it refers to the single stone on the top, but it could be misinterpreted as suggesting the entire circled region comprising of two stones is "stone 58". Adding an arrow should resolve this issue.

Also, why is 'Bluestone' is in quotes? If it's a vernacular term, how is that different to sarsen, which is not in quotes?

Response: We have added an arrow to Fig 1 to indicate the position of Stone 58 and removed the quotation marks around ‘bluestone’.

figure 5: If there's no coarse sand, why include it in the figure? Remove and simplify. And is it possible to add the actual sizes for the different sizes?

Response: We have added actual sizes to the class boundaries as suggested. There is sediment at ~60cm along the core that is on the boundary between medium and coarse sand, so we need to retain this in the figure. 

line 271-272: This part is a bit unclear. You are describing your work in order, so at this stage you do not know yet that those minerals are TiO2. And are the TiO2 larger or smaller than 2 um? It's also unclear. TiO2 does not have "constituent minerals". There are several minerals, for example rutile or anatase, that are composed of TiO2. I would rewrite this as "Some fine-grained minerals grains could not be optically identified, but were later revealed to be TiO2. Whether they are rutile, anatase, or brookite remains to be determined." Or something like that.

Response: The reviewer underestimates the power of optical microscopy here. TiO2 minerals are easily identifiable from their optical properties in reflected light and, in some cases, it is possible to distinguish rutile and anatase. All the opaque minerals described in this section were identified using their optical properties alone, with no recourse to mineral chemistry. We have added the following to section 3.3.1 to explain further how mineral identification proceeded: “Mineralogical identification in transmitted and reflected light was made following standard optical mineralogy texts [e.g. 20, 21] and atlases [22], with petrographic descriptions also following standard protocols for sedimentary rocks [23] and silcretes [1].” Four new references have been added:

Deer WA, Howie RA, Zussman J. An Introduction to the Rock-Forming Minerals (Second Edition). London: The Geological Society; 2013.

Folk RL. Petrology of Sedimentary Rocks. Austin TX: Hemphill Publishing Co.; 1974.

Ixer RA. Atlas of Opaque and Ore Minerals in their Associations. Milton Keynes: Open University Press; 1990.

Kerr PF. Optical Mineralogy. London: McGraw-Hill; 1959.

We have refined the text about the size of mineral phases that could be identified, as follows: “All mineral phases greater than 2 µm diameter were identified. Note, however, that the fine-grained nature of the TiO2 phases present sometimes prevented further discrimination. Where a TiO2 phase could be identified with certainty it is given a mineral name in section 3.3.2; where not, it is simply referred to as a TiO2 mineral.”

line 276: What is the scale per pixel?

Response: Pixel size is ~2 µm - the final sentence of section 3.3.1 now reads “A ZEISS Axio Imager.M2m light microscope with motorised stage was used to obtain these images with a resolution of ~14,000x12,000 pixels and a pixel size of ~2 µm.”

line 281: How was 187 um determined? Automated petrography? Manual estimates? What's the standard deviation on that number? And remember that you are looking at a section. Unless you cut all grains exactly in their middle, you are underestimating their true size. There are methods to infer the real size from the observed size - this book has some approaches: https://doi.org/10.1017/CBO9780511535574 I'm not sure how important this is to your study, but might as well try. Shouldn't be too difficult.

Response: This is a standard particle size class division and was determined by a grain size card as described in the second sentence of section 3.3.1. 

figure 10: Can you annotate the figure to show where the accessory minerals and dust lines are?

Response: We have added arrows to indicate these features and amended the fig caption.

line 290: I don't see polycrystalline quartz in fig 10 A or B. Also, "metamorphic origin" is a bit broad. Do you mean mylonitic quartz? And it implies the other quartz is not "metamorphic", but you do not know that.

Response: Thanks for this. We have edited the caption to Fig 10 to remove the reference to clear and polycrystalline quartz and note mylonitic as opposed to metamorphic quartz in the section of 3.3.2 discussing host sediments.

line 292: IMA-approved spelling is "hematite".

Response: Many thanks. Corrected in the text except where quoted from an original study (e.g. in section 5.2).

line 293: How did you distinguish magnetite from hematite? They are notoriously challenging to tell apart.

Response: The reviewer is mistaken here. In reflected light, the optical properties of magnetite and hematite are so distinct it would be almost impossible to confuse them. Magnetite appears brown, has a lower reflectance and is isotropic. Hematite appears blue to white with higher reflectance, shows strong anisotropy and marked bireflectance and has spectacular blood red internal reflections.

line 296: Well you did find feldspars in QEMSCAN, so having it here is reasonable.

Response: Agreed, although using optical petrography it was not recognised, hence our cautious wording here.

line 299: Add a figure for this section?

Response: Rock clasts are very minor components of the sarsen so we would prefer not to draw unnecessary attention to them by including a further figure.

line 306: You said earlier that your zircon was not suitable for dating. 200x60 um zircon is as good as it gets for dating, so this is contradictory.

Response: There are indeed large zircons but in insufficient numbers to allow statistically reliable age estimates (as we note in the introduction). We have added the words “statistically reliable” to the final sentence on p.4 to provide explanatory detail and note in section 3.3.2 that all detrital accessory minerals are rare.

line 312: I would recommend against identifying the spinel as chromite based on whole rock data. It may be spinel, or not. Cr2O3 can go into many of the other phases as well. (after seeing Table 2 - well if you identified it in QEMSCAN, why this phrasing? Just say it's chromite, and why do you even need then whole rock data?).

Response: The initial identification of the spinel phase as being chromite was based purely on its optical properties in reflected light. This was confirmed by SEM-EDS and QENSCAN analyses. We have revised the sentence to read: “(iii) very rare, 20-50 μm diameter, euhedral spinel (present in thin-sections from SH1A and SH2A only), most likely chrome-rich magnetite/spinel or chromite (based on optical properties in reflected light and supported by QEMSCAN and SEM-EDS analyses; see section 3.4.2)”

table 2: I would combine the two illites. This is a very fine grained mineral, and the QEMSCAN cannot distinguish illite from Fe-illite from an intergrowth of illite and hematite. Just call everything illite. IMA-approved spelling is "baryte". I'm also curious about the tourmalines. Mg-tourmaline (dravite) is distinguished from a tourmaline with no chemical information - is this a schorl? In my opinion, just combine all tourmalines into one. They are strongly zoned minerals, and the distinction is probably the result of the beam hitting a specific zone.

Response: This comment has three components: 

(I) Combining illite and Fe illite. We would prefer not to do this. Any illite / hematite intergrowths present would be on a very small scale. By documenting both illite and Fe illite we provide the highest resolution data possible from our sections and draw attention to the fact that there are two illite compositions present in the samples. 

(ii) Barite v Baryte. Spelling changed in text and on Fig 11.

(iii) Combining the tourmalines. We would prefer not to do this. The sarsen thin-sections are so quartz-rich that obtaining any detail about the trace mineralogy is critical. QEMSCAN can easily differentiate the different tourmaline compositions. We can confirm that our results are not the product of zoned minerals – there are two distinct tourmaline compositions present in the thin-sections.

line 432: What are the wavelengths for the RGB filters that you used?

Response: We have revised the text to explain this: “Four digital images (each 540x540 pixels, 320 nm pixel size) with three energy ranges, red (600-750 nm wavelength), green (475-580 nm) and blue (375-450 nm)…"

line 440: What you're describing here is essentially QEMSCAN on steroids. The XMax 80 detector has much better resolution than the Bruker detectors installed on QEMSCAN, and you have a resolution of 3.2 um per pixel, much better than the QEMSCAN 10 um. This raises the question - why was QEMSCAN conducted, if a superior method is already described here?

Response: We disagree with the reviewer that one method is superior and point out earlier in the manuscript that these methods are instead complementary. While the ~3 times better pixel resolution for conventional Aztec EDS allows the identification of smaller features (see Fig 14), QEMSCAN provides quantitative data of the modal content for classified minerals (Table 2). The multiple EDS detectors within the QEMSCAN instrument provide better pulse throughput and impulse statistics that are necessary for a reliable mineral classification. The same data quality could only be achieved by conventional SEM-EDS with a single EDS detector by significantly extending both acquisition and evaluation time. Considering an acquisition time of 20-29 hours for the existing datasets, this would be difficult to realise.

line 464: If that's the case, you wouldn't see blue, you'd see only red. I reckon any beam damage is minor, and you can either exclude this statement or add a caveat that it's probably minor.

Response: Many thanks. The text has been modified to address this comment: “We acknowledge, however, that CL colour shifts from blue to red with increasing radiation, and that this has almost certainly happened, to a limited extent, during the scanning of the polished sections.”

line 528: Then what is the purpose of pxrf? You already have core scanner XRF data which is much better. The only reason I could think of why pxrf data would be useful is to have some reference when someone goes with a pxrf to the field and tries to see if an arbitrary sarsen is somewhat similar to those from Stonehenge. If that's the case, then say it.

Response: This relates to the reviewer’s second major concern about the manuscript – why we use a wide range of methods when some are ‘superior’ to others (see p.2). Please see our earlier response and the amendments to the final paragraph of the introduction. We have also added the following to section 4.1.2: “we include the data here for the benefit of future researchers seeking to explore sarsen chemistry via PXRF”.

figure 15 and related text: The correct spelling is "sulfide". This is not an American/British spelling issue. The British Royal Society of Chemistry, IUPAC, and other UK-based organisation has long ago adopted "sulfide". Please, let's use the correct spelling. And, the lack of correlation between Fe and Cu or Zn only means that Fe variation is not related to chalcopyrite or sphalerite, not to sulfides in general. You could have pyrite-hosted Fe. That said, your work so far has showed that Fe is overwhelmingly hosted in oxides and hydroxides. Was the correlation with chalcopyrite or sphalerite even an issue?

Response: Spelling of sulfide has been corrected throughout. S is below the PXRF detection limit in all readings – therefore we did not include a bivariate diagram of Fe vs S in our original submission. Instead, we used bivariate plots of chalcophile “sulfur-loving” elements (Cu and Zn) as a ‘best-fit’ proxy for the presence of sulphides and their weathering products. We wish to retain the panels with Cu and Zn vs Fe in Fig 15. We have amended the text in section 4.1.2 to explain that S is below detection limit throughout the core (it is only present in very rare tiny pyrite inclusions in quartz), and that this observation (along with the lack of relationship between Fe and Cu or Zn) indicates that the distribution of Fe in the core is unlikely to be primarily controlled by the presence of metal sulfide grains.

line 600/fig 17: Since Zr is a trace element in your rock most pixels should result in zero Zr, with very strong Zr peaks when it hits a zircon. I'd expect it to look like the Fe map - dark grey with several white pixels. Instead, it's all ~homogeneous medium grey. To me, this appears as if it did not hit a single zircon, and instead it is mapping background noise. Can you clarify this please?

Response: Zr is present at much lower concentrations than Si and Fe. The corresponding signal-to-noise ratio is far from ideal with some background noise introduced in panel C. We have revised the caption to Fig 17 to explain this.

fig 18: The rainbow colour scheme is not a good map to use. If you have access to the data or software and can change the colour scheme to something else, I strongly recommend to do so. Read this for the motivation: https://doi.org/10.1038/s41467-020-19160-7

Response: Thanks for this observation. We have read the suggested article and looked again at the data but would prefer to leave the figure as per the original manuscript. The false colour display (as shown) makes it much easier to discriminate minor intensity changes, such as the Fe-rich lineation present on thin-section SH3B. 

line 627: Rb is one of the most mobile elements known. Sr and U (when hexavalent) are also rather mobile.

Response: Thanks for this point. The text has been clarified.

line 360-361: Calling these major elements is not accurate. They are commonly considered as major elements in many rocks, but if they're not major, then they're not major elements.

Response: There is no agreed definition of a ‘major element’ versus a minor element or trace element. We use the term ‘major’ following the convention widely adopted in geoscience research and industry. This recognises that 95% of crustal materials are made out of Si, Al, Fe, Ti, Mn, Mg, Na, K and P oxides – hence these are the ‘major’ elements (see, for example, https://www.sgs.co.uk/en-gb/mining/analytical-services/chemical-testing/major-elements).

figure 19: Your data shows Eu bdl in two samples, yet your lines are continuous through Eu in fig 19. This is creating the false impression that there is no Eu anomaly (which most likely exists). One way to solve this is to make the line connecting Sm and Gd dashed or dotted. Regarding normalisation factors - the M&S paper is from 25 years ago. We have much better data today. It would probably not matter much, but I would consider it best practice to use newer values. My personal favourites are the ones available here: https://doi.org/10.1093/petrology/egw047 column CI in table 1.

Response: Thank you for this comment. We have done as suggested. First, we have used the latest chondrite values to redraft the figure. Second, for the Eu and Lu analyses that were below detection limit, we have added dashed trends that go through these elements at detection limit. The caption has been amended accordingly. This change requires the replacement of original reference 38 by the following: 

O’Neill, H. S. C. (2016). The smoothness and shapes of chondrite-normalized rare earth element patterns in basalts. Journal of Petrology, 57(8), 1463–1508.

Table 5: It's customary to sort trace elements by atomic number, not by name. The REE follow a logical order.

Response: We have changed the order of elements in Table 5 as the reviewer suggests.

line 651: Not necessarily non-standard. There aren't much REE in quartz, so your REE budget is dominated by the tiny amount of other minerals in your rock. Zircon is one mineral that comes into mind, and probably tourmaline. The REE pattern then reflects the sorting of those minerals during sedimentary transport, and does not necessarily imply that the material was "non-standard".

Response: We completely agree with the reviewer and make the point twice that the REE (and other trace element) abundances are controlled by the non-quartz mineralogy of the rock (e.g. lines 678 and 690 in the original manuscript). 

Thanks for pointing out zircon and tourmaline as mineral phases within which the HREE have increasing compatibility with increasing atomic number. We have amended the text to include discussion about the potential role of these minerals.

In line 653 of our original submission, we used ‘non-standard’ in the sense that the mineralogy of the sarsen must have over- (and/or) under-abundances of certain mineral phases relative to the UCC standard. We have amended the text to remove any ambiguity.

figure 20: This really shows the elements. You have high in Zr and Hf, and likewise high HREE. I'm almost certain that your HREE signal is coming from zircon. Nb and Ta are also a bit high, which is what you would expect from rutile (a titanium oxide), which is also common in your rock. Uranium is concentrated in hematite. Cr and V in magnetite and spinels.

Response: We appreciate the reviewer’s insights into compatibility. We have amended the text to give a fuller account of which mineral phases could be driving the trace element signature of the Phillips’ Core. Two references to be added are:

Bea, F., Pereira, M.D. and Stroh, A. (1994). Mineral/leucosome trace-element partitioning in a peraluminous migmatite (a laser ablation-ICP-MS study). Chemical Geology 117: 291-312.

Van Hinsberg, V. J. (2011). Preliminary experimental data on trace-element partitioning between tourmaline and silicate melt. The Canadian Mineralogist, 49(1), 153–163.

line 682: No, the upwards HREE trend is probably real. As I said above, it's probably coming from zircon. It could also be from the clays, HREE tend to adsorb into them.

Response: We have addressed this point in our response to the previous comment. 

figure 22: Sc and Co are bdl, so placing points on a ternary that includes them is misleading. This can be easily misused by others, and the points plotted in subsequent papers showing a "Stonehenge" field, without actually reading your paper (happens too often, unfortunately). A better way to do it would be to plot the points twice - once at the detection limit, and another time at zero Sc/Co, and draw a line between them. Then you're showing the possible range, and better representing the limited data available.

Response: We completely agree. We have amended Fig 22 to show the core samples as arrays defined by Co and Sc extremes of 1 and 0 ppm. The caption has been amended to explain how the arrays were derived. This approach makes the La-Sc-Th plot (Fig 22A in the original submission) obsolete as the newly defined array crosses all 4 tectonic setting fields. We have removed Fig 22A in the resubmission and renamed other panels accordingly.

line 761: There are few/no Rb-bearing minerals at this moment, but they were clearly there when the Sr isotope signature formed. 0.713 is anything but low - it's a reasonable continental crust value. It is all relative though - I mostly work on mantle rocks where "relatively low" is 0.702 and 0.713 would be extremely radiogenic.

Response: We appreciate that a 87Sr/86Sr value of 0.713 would be high for someone who works on basalt. However, this is the modern measured value of a British sediment and, as such, it is low. Welsh sediments typically have measured values of 0.72-0.74 as do the Millstone Grit and the Pennant sandstone. We have added the phrase “compared with other British sedimentary rocks” to qualify the use of ‘relatively low’ and inserted three references (see below).

line 762: Not quite - crustal residence age is not the time it has been in a sedimentary environment. It has been the time since it was separated from the mantle. It could have been in igneous rocks ever since.

Response: Accepted. We have inserted “consistent with”, cited comparison reference papers and removed the final part of this sentence to address this point.

Davies GR, Gledhill A, Hawkesworth C. Upper crustal recycling in southern Britain: evidence from Nd and Sr isotopes. Earth and Planetary Science Letters. 1985; 75:1-12.

Evans JA. Dating the transition of smectite to illite in Palaeozoic mudrocks using the Rb–Sr whole-rock technique. J Geol Soc. 1996; 153:101-108.

Johnson L, Montgomery J, Evans JA, Hamlton E. Contribution of strontium to the human diet from querns and millstones: an experiment in digestive strontium isotope uptake. Archaeometry. 2019; 61:1366-1381.

line 763: Missing space? c.0.19

Response: Corrected.

line 787: As said before, the distinction between Mg-tourmaline and tourmaline, the illites, dolomite and Fe-dolomite is an arbitrary divide set by the person who designed the QEMSCAN mineral identification list, and most likely has no petrological significance. For simplicity, I would group them together into just tourmaline, illite, and dolomite.

Response: Please see our response to this point earlier. Given the quartz content of our samples, we wish to provide as much mineralogical detail as possible and would prefer to note all the separate mineral phases we have identified.

line 800: That's an example of the issue of structuring the paper according to methods. It's not only ICP, everything you have indicates the silcrete is pure.

Response: This is true but see our response to the reviewer’s first main point above. We have added “and other geochemical data” to expand on the reviewer’s comment.

line 806: What is the evidence for two cycles?

Response: The isotope data show that the sarsen host sediments originally derived from Mesoproterozoic rocks that were eroded and the resulting sediments deposited during the Mesozoic; these Mesozoic sediments were then eroded and redeposited during the Palaeogene. However, we agree that claiming evidence for a specific number of cycles of erosion/deposition is probably over-stretching the data. We have revised the end of section 5.1 accordingly and edited the corresponding text in the abstract.

 

Reviewer #2: 

This is an incredibly detailed and excellent study of the Sarsen Stone 58 at Stonehenge, and should be published as is!

The only minor edits/additions I would suggest are:

- p.4, para 2, line 81ff: give some brief reasons as to why all these methods were necessary to use in the analysis?

Response: Please see our response to the second major issue raised by Reviewer #1 (p.2 above).

- p. 27, end of para 2, line 670: what does the resemblance to 'Archean sandstones' imply?

Response: There is no immediate implication. This statement is purely descriptive and intended to show that, geochemically, the Stonehenge sands are ‘most similar’ to Archean sandstones as compared to the other end members shown in Fig 20. Adding any further discussion at this stage about potential implications would be speculative – largely because of the limited nature of the global database of sandstone geochemistry.

- p.35, para 3, line 926: what is the proximity of West Woods to Stonehenge?, and therefore the feasibility of moving sarsens from there to Stonehenge?

Response: We have added the phrase “25 km north of Stonehenge” to this paragraph. Nash et al. (2020) discuss potential transport routes for sarsens from West Woods to Stonehenge, so we would rather not repeat that information here.

 

Reviewer #3: 

This paper reports on an exhaustive investigation of an important archeological artifact, and as such the results have potentially important scientific and cultural significance. The depth and detail of the analyses may exceed that of any previous study of a quartz sandstone, and perhaps of any other terrestrial rock sample! This makes for rather a long manuscript, but it is very clearly written, well-organized, and easy to read. The data employed all seem appropriate and reliable, and the conclusions generally appear well-supported by the data.

I do have two specific comments aimed at further improving the manuscript. First, the introduction might be improved by a more explicit statement of the problem or hypothesis being addressed. Likely this seems self-evident to the authors but it might not be so to the reader. I surmise that the main question is ‘what is the provenance of this and other sarsen stones?’ If so, it would be nice to have some information in the introduction (or discussion) regarding specific source candidates. No doubt specialists in this arena are already familiar with the alternatives, but adding a few lines for the benefit of the broader readership would be helpful.

Response: Many thanks. We have tightened up the statement of the aims of the paper in response to this suggestion.

Secondly, the manuscript correctly notes several times that the studied sample is “technically” a quartz arenite or orthoquartzite. Why then the continued use of the term “silcrete?” Silcrete carries a specific genetic implication, i.e., silica precipitation associated with soil formation. If this is indeed the preferred interpretation then it needs to be explicitly defended based on the data presented in the study. At present there is passing mention of cement precipitation from groundwater, but such precipitation is not unique to silcrete. Most (all?) cements precipitate from groundwater. Syntaxial, zoned quartz cement is common in many quartz arenites for which there is no association with soil formation. For example, similar cements in the Cambrian of the central U.S. have been attributed to advection of fluids expelled from adjacent basins. If there are cogent arguments in favor of the sarsen stones being silcrete then they should be discussed. If not then I suggest sticking with ‘quartz arenite’ in the interest of precision.

Response: We thank the reviewer for this observation. Rather than go into a lengthy discussion of why the sarsen is a groundwater silcrete formed in the near-surface, as opposed to a quartz arenite cemented at depth (this has been well-rehearsed in the literature), we have (i) added clarifying text to section 3.3.2, and (ii) removed all references to quartz arenite except for the first mention in section 3.1.2. 

The key addition to section 3.3.2 is the following: “We identify the sarsen as a groundwater silcrete (as opposed to a pedogenic silcrete) based on its simple micromorphology, textural homogeneity and lack of pedogenic features such as geopetal and colloform structures (see discriminating criteria in [24]). The tabular morphology of most sarsen boulders at Stonehenge is also consistent with this interpretation.”

 

We have edited author names and affiliations and saved all files according to the PLoS ONE style requirements.

2. In your manuscript, please provide additional information regarding the specimens used in your study. Ensure that you have reported specimen numbers and complete repository information, including museum name and geographic location. If permits were required, please ensure that you have provided details for all permits that were obtained, including the full name of the issuing authority, and add the following statement to both the Ethics Statement and Methods section: 'All necessary permits were obtained for the described study, which complied with all relevant regulations.' If no permits were required, please include the following statement in the aforementioned sections: 'No permits were required for the described study, which complied with all relevant regulations.' For more information on PLOS ONE's requirements for palaeontology and archaeology research, see https://journals.plos.org/plosone/s/submission-guidelines#loc-paleontology-and-archaeology-research.

We have added the phrase “All necessary permissions were obtained for these analyses, which complied with all relevant regulations” in section 2.3 where we describe the subsampling of the Phillips’ Core. We have amended the Ethics Statement accordingly.

The dataset associated with the manuscript is in the process of being archived with the Archaeology Data Service (University of York, UK). The early doi for the dataset is: https://doi.org/10.5284/1084808.

4. Thank you for stating the following in the Financial Disclosure section:

"DJN, TJRC, JSU, MPP, TD - awarded British Academy / Leverhulme Trust Small Research Grant SG170610 https://www.thebritishacademy.ac.uk/

DP received additional funding support from the University of South Wales CESRIS grant (no number) https://www.southwales.ac.uk/

We note that one or more of the authors are employed by a commercial company: Gatan UK, Vidence Inc. and Petroclays Ltd

We have updated the Funding Statement to read:

DJN, TJRC, JSU, MPP, TD were awarded British Academy / Leverhulme Trust Small Research Grant SG170610 https://www.thebritishacademy.ac.uk/. DP received additional funding support from the University of South Wales CESRIS grant (no number) https://www.southwales.ac.uk/. These funders had no role in study design, data collection and analysis, decision to publish, or preparation of the manuscript.

JMH, MRP and NW are employed by commercial companies (Petroclays Ltd, Vidence Inc. and Gatan UK, respectively). These companies provided support in the form of salaries but did not have any additional role in the study design, data collection and analysis, decision to publish, or preparation of the manuscript. The specific roles of these authors are articulated in the ‘author contributions’ section.

We have updated the Competing Interests Statement to read:

Three of the authors (JMH, MRP and NW) are employed by commercial companies (Petroclays Ltd, Vidence Inc. and Gatan UK, respectively). This does not alter our adherence to PLOS ONE policies on sharing data and materials.

5. We note that Figure 2 and 4 includes an image of a patient / participant in the study. 

We have obtained written permission from Lewis Phillips, Robin Phillips and Heather Sebire to use their image in Fig 4. We have also obtained written permission from Lewis Phillips to use the image of his father, Robert Phillips (deceased), in Fig 2.

We have updated the Ethics Statement of the manuscript and inserted the following text at the end of section 2.1:

“Mr Lewis Phillips (Robert Phillips’ son) has given his written informed consent (as outlined in the PLOS consent form) to publish the image of his father in Fig 2.”

…and at the end of section 2.2:

“The individuals in Fig 4 have given their written informed consent to publish their details alongside this image.” 

6. We note that [Figure(s) 1] in your submission contain map images which may be copyrighted. All PLOS content is published under the Creative Commons Attribution License (CC BY 4.0), which means that the manuscript, images, and Supporting Information files will be freely available online, and any third party is permitted to access, download, copy, distribute, and use these materials in any way, even commercially, with proper attribution. For these reasons, we cannot publish previously copyrighted maps or satellite images created using proprietary data, such as Google software (Google Maps, Street View, and Earth). For more information, see our copyright guidelines: http://journals.plos.org/plosone/s/licenses-and-copyright.

1. You may seek permission from the original copyright holder of Figure(s) [1] to publish the content specifically under the CC BY 4.0 license. 

I was advised in an email from Eoin O'Connor (PLoS Editorial Office) on 12 January 2021 to pass on the following information in this response.

Fig 1 includes components that have been redrawn from a figure originally included within the following paper: 

Nash DJ, Ciborowski TJR, Ullyott JS, Parker Pearson M, Darvill T, Greaney S, Maniatis G, Whitaker KA. Origins of the sarsen megaliths at Stonehenge. Science Advances [Internet]. 2020; 6: eabc0133. Available from: https://doi.org/10.1126/sciadv.abc0133.

Six of the authors of the current manuscript were authors of this original paper – Nash drafted the original figure. The AAAS ‘Permission for authors’ states: “If you are the author of the article that was published in a Science journal or on a Science website, you retain the rights to use your paper and its contents as permitted under AAAS's License to Publish”. This paper was originally distributed under a Creative Commons Attribution Non-Commercial License 4.0 (CC BY-NC) so we assume there are no copyright issues.

---

## [Decision Letter · Decision Letter 1]

2 Jun 2021

PONE-D-20-32409R1

Petrological and geochemical characterisation of the sarsen stones at Stonehenge

PLOS ONE

Dear Dr. Nash,

Thank you for submitting a revised version of your manuscript following the comments made by the three reviewers. The revised manuscript was reviewed by referee #1, who pointed to several minor issues that need your attention and corrections for the manuscript to be improved. This should not be difficult to perform. Please submit a corrected version of the manuscript, referring to the attached comments by reviewer #1, for it to be accepted for publication.

We look forward to receiving your revised manuscript.

Kind regards,

Yaron Katzir

Academic Editor

PLOS ONE

Journal Requirements:

Reviewers' comments:

Reviewer's Responses to Questions

**Comments to the Author**

1. If the authors have adequately addressed your comments raised in a previous round of review and you feel that this manuscript is now acceptable for publication, you may indicate that here to bypass the “Comments to the Author” section, enter your conflict of interest statement in the “Confidential to Editor” section, and submit your "Accept" recommendation.

Reviewer #1: (No Response)

2. Is the manuscript technically sound, and do the data support the conclusions?

Reviewer #1: Yes

3. Has the statistical analysis been performed appropriately and rigorously? 

Reviewer #1: N/A

4. Have the authors made all data underlying the findings in their manuscript fully available?

Reviewer #1: Yes

5. Is the manuscript presented in an intelligible fashion and written in standard English?

Reviewer #1: Yes

6. Review Comments to the Author

Reviewer #1: Looks good now. Some really minor issues, see attached file. Very good and detailed work, and well written.

7. PLOS authors have the option to publish the peer review history of their article (what does this mean?). If published, this will include your full peer review and any attached files.

Reviewer #1: No

---

## [Author Response · Author response to Decision Letter 1]

14 Jun 2021

Response to Reviewer

We thank the reviewer for his/her incredibly detailed second set of observations – even going as far as checking the supplementary data tables – that have greatly improved the manuscript. We have made changes in response to every comment, as follows:

[Reviewer] Line 25/26 “That’s a very Eurocentric statement. The Great Pyramids of Giza were built at about the same time…”

[Response] Corrected to read “…arguably the most important Late Neolithic monument in Europe”.

[Reviewer] Line 36 “omit, no need to refer to later sections in the introduction”

[Response] Corrected as suggested.

[Reviewer] Line 63/64 ““…of the Stonehenge architecture”? Reads a bit better

[Response] Corrected as suggested.

[Reviewer] Line 66 “What are hammerstones? I can guess, but be clear please”

[Response] We first use the term ‘hammerstone’ in lines 48-50. We have made minor edits to the text to expand on this information such that we do not need to repeat it later: “The hard sarsen appears to be derived from hammerstones of various size broken in the process of shaping (or dressing) the stones on site during construction.”

[Reviewer] Line 112/113 “Again, with all due respect to Stonehenge, The Great Pyramid of Khufu has already been standing for around 200 hundred years when the sarsen circles were erected…”

[Response] Corrected, as above, to read “…in Europe”

[Reviewer] Line 156-158 “Shouldn’t this be in the figure caption?” AND Line 175-176 “In the figure caption?”

[Response] We have relocated the information about informed consent into the captions for Figures 2 and 4. We trust that this is acceptable for the journal. 

[Reviewer] Line 189 “36 cm”

[Response] Corrected as suggested. We have checked through the remainder of the manuscript and inserted spaces between numbers and units wherever necessary.

[Reviewer] Line 207-208 “Does that mean it had a glass cover on top?”

[Response] We have made minor edits to the text and inserted the following sentence for clarification: “Glass cover slips were not applied to either set of thin-sections”. 

[Reviewer] Line 277 “No need to capitalise C and T here”

[Response] Corrected as suggested.

[Reviewer] Line 334-335 “Where? I don’t see anything mylonitic in figs 10A and B. And “mylonite” is a dynamic-metamorphic rock or process, which is probably not what happened in your rocks.”

[Response] In the interests of clarity for the reader, we have opted to delete the phrase ‘with minor mylonitic quartz’ from the text.

[Reviewer] Line 340-341 “Then why say it? Was there any reason to expect twinned feldspar?”

[Response] For brevity, we have deleted this sentence from the manuscript. 

[Reviewer] Line 344-349 “References to figures?”

[Response] We have added the following sentence to the end of this paragraph: “An example of rock clast type (iii) is visible bottom centre right of Fig 10D.” 

[Reviewer] Line 361 “Figures showing these things?”

[Response] We have added the following sentence to the end of this paragraph: “The characteristics of titanium mineral phases are discussed further in section 3.5.2, with examples visible in Fig 14”. 

[Reviewer] Line 387 “two spaces”

[Response] Corrected as suggested and checked throughout the manuscript.

[Reviewer] Line 399 “space between number and units: 20 kV. Here and elsewhere”

[Response] Corrected as suggested and checked throughout as noted above.

[Reviewer] Line 399 “change the letter x into the multiplication symbol: ×”

[Response] Corrected as suggested and checked throughout the manuscript.

[Reviewer] Line 465 “illite-smectite appears twice here”

[Response] We have corrected this text to make it clear that there are two populations of illite-smectite present: “…illite and illite-smectite, Fe-illite and Fe-illite-smectite…” 

[Reviewer] Line 505 “Having sub-nm beam precision is extremely difficult to achieve in porous material (like yours). I suggest to say “~750”. This is sufficiently precise, at 0.75 micrometre.”

[Response] Corrected as suggested.

[Reviewer] Line 515 “Your own radiation when analysing, or background geological radiation? If the former, then it would affect all quartz similarly, which obviously didn’t happen.”

[Response] Thanks for this observation. For clarification we have revised the sentence as follows: “We acknowledge, however, that CL colour shifts from blue to red with increasing radiation during analysis, and that this has almost certainly happened, to a limited extent, during the scanning of the polished sections.” 

[Reviewer] Line 559 ““pXRF” is much more common term than “PXRF”

[Response] Corrected as suggested and throughout the manuscript.

[Reviewer] Line 760 “no need to capitalise HFS”

[Response] Corrected as suggested.

[Reviewer] Line 784 “no need to capitalise”

[Response] Corrected as suggested.

[Reviewer] Line 819 “1.3 ppm”

[Response] Corrected as suggested and checked throughout as noted above.

[Reviewer] Line 849 “illite-smectite is repeated”

[Response] As noted above, we have corrected this text to make it clear that there are two populations of illite-smectite present: “…illite and illite-smectite, Fe-illite and Fe-illite-smectite…” 

[Reviewer] Line 859 “What is this? It’s written like it’s a term for something specific?”

[Response] We have added the following text to explain the term Clay-with-Flints for an international readership: “(a residual deposit formed from the dissolution, decalcification and cryoturbation of the Chalk Group and Palaeogene formations)”.

[Reviewer] Line 948 “tourmaline-absent”

[Response] Corrected as suggested.

[Reviewer] Line 960 “no space between number and %”

[Response] Corrected as suggested.

[Reviewer] Line 1032 “Fisher”

[Response] Corrected as suggested.

[Reviewer] Line 1230 “There’s a column called “LE” after uranium – what’s that?”

[Response] LE stands for light elements. We have added clarifying text to the bottom of the spreadsheet in S1 Data as follows: “LE indicates 'light element fraction' (atomic number <18)”.

---

## [Editor Report · Decision Letter 2]

5 Jul 2021

Petrological and geochemical characterisation of the sarsen stones at Stonehenge

PONE-D-20-32409R2

Dear Dr. Nash,

We’re pleased to inform you that your manuscript has been judged scientifically suitable for publication and will be formally accepted for publication once it meets all outstanding technical requirements.

Kind regards,

Yaron Katzir, PhD

Academic Editor

PLOS ONE
---

## [Editor Report · Acceptance letter]

13 Jul 2021

PONE-D-20-32409R2 

Petrological and geochemical characterisation of the sarsen stones at Stonehenge 

Dear Dr. Nash:

I'm pleased to inform you that your manuscript has been deemed suitable for publication in PLOS ONE. Congratulations! Your manuscript is now with our production department. 

Kind regards, 

on behalf of

Dr. Yaron Katzir 

Academic Editor

PLOS ONE